# Nonuniform Negative Sampling and Log Odds Correction with Rare Events Data

**HaiYing Wang**
Department of Statistics
University of Connecticut
haiying.wang@uconn.edu

**Aonan Zhang**
ByteDance Inc.
aonan.zhang@bytedance.com

**Chong Wang**
ByteDance Inc.
chong.wang@bytedance.com

## Abstract

We investigate the issue of parameter estimation with nonuniform negative sampling for imbalanced data. We first prove that, with imbalanced data, the available information about unknown parameters is only tied to the relatively small number of positive instances, which justifies the usage of negative sampling. However, if the negative instances are subsampled to the same level of the positive cases, there is information loss. To maintain more information, we derive the asymptotic distribution of a general inverse probability weighted (IPW) estimator and obtain the optimal sampling probability that minimizes its variance. To further improve the estimation efficiency over the IPW method, we propose a likelihood-based estimator by correcting log odds for the sampled data and prove that the improved estimator has the smallest asymptotic variance among a large class of estimators. It is also more robust to pilot misspecification. We validate our approach on simulated data as well as a real click-through rate dataset with more than 0.3 trillion instances, collected over a period of a month. Both theoretical and empirical results demonstrate the effectiveness of our method.

## 1 Introduction

Imbalanced data are ubiquitous in scientific fields and applications with binary response outputs, where the number of instances in the positive class is much smaller than that in the negative class. For example, in online search or recommendation systems where billions of impressions appear each day, non-click impressions usually dominate. Using these non-click impressions as negative instances is prevalent and proved to be useful especially in modern machine learning systems [21, 12, 4, 9, 14]. A common approach to address imbalanced data is to balance it by subsampling the negative class [6, 19] and/or up-sampling the positive class [2, 10, 20, 5], and a great deal of attentions have been drawn to this problem, see [15, 17, 3, 7, 23, 27, 1, 25, 8, 18, 30], and the references therein.

In this paper, we focus on subsampling the negative class, i.e., negative sampling, while keeping all rare positive instances. This is beneficial for modern online learning systems, where negative sampling significantly reduces data volume in distributed storage while saving training time for multiple downstream models. Rare events data and negative sampling are studied in [30], but it focused on linear logistic regression and only considered uniform sampling. We approach this problem under general binary response models from the perspective of optimal subsampling, which aims to minimize the asymptotic variance of the IPW estimator [32, 28, 31, 33]. This topic was not well investigated neither for imbalanced data nor for negative sampling. In addition, existing optimal

probability formulation minimizes the conditional asymptotic variance and the variations due to the data randomness are ignored. We fill these gaps by deriving the asymptotic distributions for both full data and subsample estimators under general binary response models, and we find the unconditional optimal probability for negative sampling. In addition, since IPW may inflate the variance [13, 22], we develop a more efficient likelihood-based estimator through nonuniform log odds correction to avoid IPW. Our main contributions are summarized as follows.

- Under a general binary response model with rare events data, we find that the difference between the full data estimator and the true parameter converges to a normal distribution at a rate that is tied to the number of rare positive instances only. This indicates the possibility to throw away the majority of negative instances without information loss, i.e, it justifies the usage of negative sampling.

- We show that there is no asymptotic information loss with aggressive negative sampling if the negative instances kept dominates the positive instances. However there is information loss expressed as a variance inflation if the negative instances are brought down to the same level of the positive instances. For this case, we obtain optimal subsampling probabilities by minimizing the unconditional asymptotic variance of a general IPW estimator.

- We develop a likelihood estimator through nonuniform log odds correction for sampled data, and prove that it has the smallest asymptotic variance among a large class of estimators.

- We apply the proposed method to a real online streaming click-through rate (CTR) dataset with more than 0.3 trillion instances and demonstrate its effectiveness.

The rest of paper is organized as follows. Section 2 defines the problem this paper focuses on and shows the full data results. Section 3 presents the optimal probability for the IPW estimator. Section 4 proposes the likelihood-based estimator and establishes its asymptotic optimality. Section 5 discusses practical implementation and the theoretical properties of the resulting estimators. Section 6 presents numerical experiments with simulated data, and applies the proposed method to a real online streaming CTR dataset with more than 0.3 trillion instances. Section 7 concludes the paper and points out limitations of our investigation. Proofs and additional experiments with misspecifications are in the supplementary materials.

## 2  Problem setup and full data results

Let $\{(\mathbf{x}_i, y_i)\}_{i=1}^N$ be training data of size $N$ that satisfies the binary response model,

$$\Pr(y = 1 \mid \mathbf{x}) = p(\mathbf{x}; \boldsymbol{\theta}), \tag{1}$$

where $y \in \{0, 1\}$ is the binary class label ($y = 1$ for the positive and $y = 0$ for the negative), $\mathbf{x}$ is the vector of features, and $\boldsymbol{\theta}$ is the unknown $d$-dimensional parameter.

Let $N_1$ be the number of positive instances ($N_1 = \sum_{i=1}^N y_i$) and $N_0$ be the number of negative instances ($N_0 = N - N_1$). We consider the scenario of massive imbalanced data, i.e., $N_1 \ll N_0$. For asymptotic investigations when $N_1$ is much smaller than $N_0$, it is more appropriate to assume that $N_1$ increases in a slower rate compared with $N_0$, i.e., $N_1/N_0 \to 0$ in probability as $N \to \infty$ [30]. This requires $\Pr(y = 1) = \mathbb{E}\{p(\mathbf{x}; \boldsymbol{\theta})\} \to 0$ as $N \to \infty$ on the model side. We assume that the parameter $\boldsymbol{\theta}$ contains two components $\boldsymbol{\theta} = (\alpha, \boldsymbol{\beta}^{\mathrm{T}})^{\mathrm{T}}$ and the log odds can be written as

$$g(\mathbf{x}; \boldsymbol{\theta}) := \log\left\{\frac{p(\mathbf{x}; \boldsymbol{\theta})}{1 - p(\mathbf{x}; \boldsymbol{\theta})}\right\} = \alpha + f(\mathbf{x}; \boldsymbol{\beta}),$$

where the true parameter, denoted as $\boldsymbol{\theta}^* = (\alpha^*, \boldsymbol{\beta}^{*\mathrm{T}})^{\mathrm{T}}$, satisfies that $\alpha^* \to -\infty$ as $N \to \infty$ and $\boldsymbol{\beta}^*$ is fixed. Here $f(\mathbf{x}; \boldsymbol{\beta})$ is a smooth function of $\boldsymbol{\beta}$, such as a linear function or a nonlinear neural network model. If it is linear, the model reduces to the logistic regression model. A diverging $\alpha^*$ and a fixed $\boldsymbol{\beta}^*$ indicate that both the marginal and conditional probabilities for a positive instance are small. Heuristically, this implies that a change in the feature value does make a rare event become a large probability event. We can also allow $\boldsymbol{\beta}^*$ to change with $N$, but as long as $\boldsymbol{\beta}^*$ has a finite limit, the situation is essentially the same as a fixed $\boldsymbol{\beta}^*$. So here we assume $\boldsymbol{\beta}^*$ to be fixed to simplify the presentation.

Throughout this paper, we use $\dot{g}(\mathbf{x}; \boldsymbol{\theta})$ and $\ddot{g}(\mathbf{x}; \boldsymbol{\theta})$ to denote the gradient and Hessian of $g(\mathbf{x}; \boldsymbol{\theta})$ with respect to (w.r.t.) $\boldsymbol{\theta}$, respectively. Let $\|\mathbf{v}\|$ be the Frobenius norm and $\mathbf{v}^{\otimes 2} = \mathbf{v}\mathbf{v}^{\mathrm{T}}$ for a vector or a

matrix $\mathbf{v}$. We denote $\pi(\mathbf{x}, y)$ as the sampling probability for an instance $(\mathbf{x}, y)$. For negative cases, we sometimes use a shorthand notation $\pi(\mathbf{x})$ to denote its sampling probability. We use $o_P(1)$ to represent a random quantity that converges to 0 in probability as $N \to \infty$.

## 2.1 It is the number of positive instances that matters

We show that for rare events data, the estimation error rate for $\boldsymbol{\theta}$ is related to $N_1$ instead of $N$. Using the full data, a commonly used estimator of $\boldsymbol{\theta}$ is the maximum likelihood estimator (MLE)

$$\hat{\boldsymbol{\theta}}_{\mathrm{f}} := \arg\max_{\boldsymbol{\theta}} \sum_{i=1}^{N} \left[ y_i g(\mathbf{x}_i; \boldsymbol{\theta}) - \log\left\{ 1 + e^{g(\mathbf{x}_i; \boldsymbol{\theta})} \right\} \right].$$

To investigate the theoretical properties of the estimator $\hat{\boldsymbol{\theta}}_{\mathrm{f}}$, we make the following assumptions.

**Assumption 1.** The first, second, and third derivatives of $f(\mathbf{x}; \boldsymbol{\beta})$ and $e^{f(\mathbf{x}; \boldsymbol{\beta})} f(\mathbf{x}; \boldsymbol{\beta})$ w.r.t. any components of $\boldsymbol{\beta}$ are bounded by a square intergrable random variable $B(\mathbf{x})$.

**Assumption 2.** The matrix $\mathbb{E}\{\dot{g}^{\otimes 2}(\mathbf{x}; \boldsymbol{\theta}^*)\}$ is finite and positive definite.

Assumption 1 imposes constraints on the smoothness of $f(\mathbf{x}; \boldsymbol{\theta})$ w.r.t. $\boldsymbol{\theta}$ and on the tightness of the distribution of $\mathbf{x}$. Assumption 2 ensures that all components of $\boldsymbol{\theta}$ are estimable.

**Theorem 1.** *Let* $\mathbf{V}_{\mathrm{f}} = \mathbb{E}\{e^{f(\mathbf{x}; \boldsymbol{\beta}^*)}\} \mathbf{M}_{\mathrm{f}}^{-1}$ *where* $\mathbf{M}_{\mathrm{f}} = \mathbb{E}\{e^{f(\mathbf{x}; \boldsymbol{\beta}^*)} \dot{g}^{\otimes 2}(\mathbf{x}; \boldsymbol{\theta}^*)\}$. *Under Assumptions 1 and 2, as* $N \to \infty$,

$$\sqrt{N_1}(\hat{\boldsymbol{\theta}}_{\mathrm{f}} - \boldsymbol{\theta}^*) \longrightarrow \mathbb{N}(\mathbf{0}, \mathbf{V}_{\mathrm{f}}), \quad \textit{in distribution.}$$

This result shows that even if all the $N$ instances are used for model training, the resulting estimator $\hat{\boldsymbol{\theta}}_{\mathrm{f}}$ converges to the true parameter $\boldsymbol{\theta}^*$ at the rate of $N_1^{-1/2}$, which is much slower than the usual rate of $N^{-1/2}$ for regular cases. This indicates that with rare events data, the available information about unknown parameters is at the scale of $N_1$. Although $N \to \infty$ much faster than $N_1 \to \infty$ (in terms of $N_1/N \to 0$), $N$ does not explicitly show in the asymptotic normality result of Theorem 1. For the specific case of linear logistic regression with $g(\mathbf{x}; \boldsymbol{\theta}) = \alpha + \mathbf{x}^{\mathrm{T}} \boldsymbol{\beta}$, our Theorem 1 reduces to Theorem 1 of [30].

## 2.2 Negative sampling

Since the available information ties to the number of positive instances instead of the full data size, one can keep all the positive instances and significantly subsmaple the negative instances to reduce the computational cost. For better estimation efficiency, we consider nonuniform sampling. Let $\varphi(\mathbf{x}) > 0$ be an integrable function and $\rho$ be the sampling rate on the negative class. Without loss of generality, assume $\mathbb{E}\{\varphi(\mathbf{x})\} = 1$ so that $\pi(\mathbf{x}_i) := \rho\varphi(\mathbf{x}_i)$ is the sampling probability for the $i$-th data point if $y_i = 1$. We present the nonuniform negative sampling procedure in Algorithm 1.

---

**Algorithm 1** Negative sampling

For $i = 1, ..., N$:   if $y_i = 1$,   include $\{\mathbf{x}_i, y_i, \pi(\mathbf{x}_i, y_i) = 1\}$ in the sample;

              if $y_i = 0$,   calculate $\varphi(\mathbf{x}_i)$ and generate $u_i \sim \mathbb{U}(0, 1)$;

                      if $u_i \le \rho\varphi(\mathbf{x}_i)$, include $\{\mathbf{x}_i, y_i, \pi(\mathbf{x}_i, y_i) = \rho\varphi(\mathbf{x}_i)\}$ in the sample.

---

## 3 Weighted estimation and its optimal negative sampling probability

Let $\delta_i = 1$ if the $i$-th data point is selected and $\delta_i = 0$ otherwise. Given a subsample taken according to $\pi(\mathbf{x}_i, y_i) = y_i + (1 - y_i)\rho\varphi(\mathbf{x}_i)$, $i = 1, ..., N$, the IPW estimator of $\boldsymbol{\theta}$ is

$$\hat{\boldsymbol{\theta}}_w = \arg\max_{\boldsymbol{\theta}} \sum_{i=1}^{N} \delta_i \pi^{-1}(\mathbf{x}_i, y_i) \left[ y_i g(\mathbf{x}_i; \boldsymbol{\theta}) - \log\left\{ 1 + e^{g(\mathbf{x}_i; \boldsymbol{\theta})} \right\} \right]. \tag{2}$$

We need an assumption on the subsampling rate to investigate the subsample estimators.

**Assumption 3.** The subsampling rate $\rho$ satisfies that $c_N := e^{\alpha^*}/\rho \to c$ for a constant $c$ such that $0 \le c < \infty$.

Note that $c_N$ cannot be zero but its limit $c$ can be exactly zero. It can be shown that $c_N \mathbb{E}\{e^{f(\mathbf{x};\boldsymbol{\beta}^*)}\} = N_1(N_0\rho)^{-1}\{1 + o_P(1)\}$, so $c\mathbb{E}\{e^{f(\mathbf{x};\boldsymbol{\beta}^*)}\}$ is the asymptotic positive/negative ratio for the sample. The theorem below shows the asymptotic distribution of $\hat{\boldsymbol{\theta}}_w$.

**Theorem 2.** *Under Assumptions 1-3, if* $\mathbb{E}[\{\varphi(\mathbf{x}) + \varphi^{-1}(\mathbf{x})\}B^2(\mathbf{x})] < \infty$ *then as* $N \to \infty$,

$$\sqrt{N_1}(\hat{\boldsymbol{\theta}}_w - \boldsymbol{\theta}^*) \longrightarrow \mathbb{N}(\mathbf{0}, \, \mathbf{V}_w), \quad \text{in distribution,}$$

*where* $\mathbf{V}_w = \mathbf{V}_f + \mathbf{V}_{\mathrm{sub}}$ *and* $\mathbf{V}_{\mathrm{sub}} = c\mathbb{E}\{e^{f(\mathbf{x};\boldsymbol{\beta}^*)}\}\mathbf{M}_f^{-1}\mathbb{E}\{\varphi^{-1}(\mathbf{x})e^{2f(\mathbf{x};\boldsymbol{\beta}^*)}\dot{g}^{\otimes 2}(\mathbf{x};\boldsymbol{\theta}^*)\}\mathbf{M}_f^{-1}$.

The subsampled estimator $\hat{\boldsymbol{\theta}}_w$ have the same convergence rate as the full data estimator. However, the asymptotic variance $\mathbf{V}_w$ may be inflated by the term $\mathbf{V}_{\mathrm{sub}}$ from subsampling the negative cases. Here $\mathbf{V}_{\mathrm{sub}}$ is zero if $c = 0$. If we keep much more negative instances than the positive instances in the subsample ($c = 0$), then there is no asymptotic information loss due to subsampling ($\mathbf{V}_w = \mathbf{V}_f$). In this scenario, the number of negative instances can still be aggressively reduced ($\rho \to 0$) so that the computational efficiency is significantly improved, and the sampling function $\varphi(\mathbf{x})$ does not play an significant role. If we have to reduce the negative instances to the same level of the positive instances ($0 < c < \infty$), then the variance inflation $\mathbf{V}_{\mathrm{sub}}$ is not negligible. In this scenario, a well designed sampling function $\varphi(\mathbf{x})$ it is more relevant and critical.

Theorem 2 shows that the distribution of the estimation error $err := \sqrt{N_1}(\hat{\boldsymbol{\theta}}_w - \boldsymbol{\theta}^*)$ is approximated by a normal distribution $\mathbb{N}(\mathbf{0}, \mathbf{V}_w)$. Thus for any $\epsilon > 0$ the probability of excess error $\Pr(err > \epsilon)$ is approximated by $\Pr\{\|\mathbb{N}(\mathbf{0}, \mathbf{V}_w)\| > \epsilon\} = \Pr\left(\sum_{j=1}^d \lambda_j z_j^2 > \epsilon\right)$, where $\lambda_j$'s are eigenvalues of $\mathbf{V}_w$ and $z_j$'s are independent standard normal random variables. Therefore, a smaller trace of $\mathbf{V}_w$ means a smaller probability of excess error of the same level since $\mathrm{tr}(\mathbf{V}_w) = \sum_{j=1}^d \lambda_j$.

The following theorem gives the optimal negative sampling function for the IPW estimator.

**Theorem 3.** *For a given sampling rate* $\rho$, *the asymptotically optimal* $\varphi(\mathbf{x})$ *that minimizes* $\mathrm{tr}(\mathbf{V}_w)$ *is*

$$\varphi_{\mathrm{os}}(\mathbf{x}) = \frac{\min\{t(\mathbf{x};\boldsymbol{\theta}^*), T\}}{\mathbb{E}[\min\{t(\mathbf{x};\boldsymbol{\theta}^*), T\}]}, \tag{3}$$

*where* $t(\mathbf{x};\boldsymbol{\theta}) = p(\mathbf{x};\boldsymbol{\theta})\|\mathbf{M}_f^{-1}\dot{g}(\mathbf{x};\boldsymbol{\theta})\|$ *and* $T$ *is the maximum number so that* $\rho[\min\{t(\mathbf{x};\boldsymbol{\theta}^*), T\}] \le \mathbb{E}[\min\{t(\mathbf{x};\boldsymbol{\theta}^*), T\}]$ *with probability one.*

**Remark 1.** If $\rho \to 0$ then $\rho t(\mathbf{x};\boldsymbol{\theta}^*) \le \mathbb{E}\{t(\mathbf{x};\boldsymbol{\theta}^*)\}$ almost surely, so $T$ can be dropped (i.e., $T = \infty$). If $\lim_{N\to\infty} \rho > 0$ then $c = 0$, so the variance inflation due to subsampling is negligible. Thus, the truncation term $T$ can be ignored in practice with imbalanced data; it only plays an role for not very imbalanced data when the sampling ratio is not very small.

In $\varphi_{\mathrm{os}}(\mathbf{x})$, $t(\mathbf{x};\boldsymbol{\theta}^*)$ consists of two components, $p(\mathbf{x};\boldsymbol{\theta}^*)$ and $\|\mathbf{M}_f^{-1}\dot{g}(\mathbf{x};\boldsymbol{\theta}^*)\|$. The term $p(\mathbf{x};\boldsymbol{\theta}^*)$ is from the binary response model structure, and it gives a higher preference for a data point with larger $p(\mathbf{x};\boldsymbol{\theta}^*)$ in the negative class. The term $\|\mathbf{M}_f^{-1}\dot{g}(\mathbf{x};\boldsymbol{\theta}^*)\|$ corresponds to the A optimality criterion [24]. For computational benefit in optimal sampling, the A optimality is often replaced by the L optimality [e.g., 32, 28]. Under the L optimality, $\|\mathbf{M}_f^{-1}\dot{g}(\mathbf{x};\boldsymbol{\theta}^*)\|$ is replaced by $\|\dot{g}(\mathbf{x};\boldsymbol{\theta}^*)\|$, and this minimizes the asymptotic mean squared error (MSE) of $\mathbf{M}_f\hat{\boldsymbol{\theta}}_w$. In case the gradient is difficult to obtain, one could ignore the gradient term and simply use $p(\mathbf{x};\boldsymbol{\theta}^*)$ to replace $t(\mathbf{x};\boldsymbol{\theta}^*)$. Although this does not give an optimal sampling probability, it outperforms the uniform subsampling because it takes into account the information of the model structure like the local case-control (LCC) sampling.

## 4 More efficient estimation based on the likelihood with log odds correction

### 4.1 Nonuniform log odds correction

The optimal function $\varphi_{\mathrm{os}}(\mathbf{x})$ assigns larger probabilities to more informative instances. However, the IPW estimator in (2) assigns smaller weights to more informative instances in estimation, so the resulting estimator can be improved to have higher estimation efficiency. A naive unweighted

estimator is biased, and the bias correction approach in [8, 29] does not work for the sampling procedure in Algorithm 1 even when the underlying model is the logistic regression. We adopted the idea of [11] and seek to define a more efficient estimator through finding the corrected likelihood of the sampled data based on any negative sampling probability $\pi(\mathbf{x})$. For data included in the subsample (where $\delta = 1$), the conditional probability of $y = 1$ is

$$p_\pi(\mathbf{x}; \boldsymbol{\theta}) := \Pr(y = 1 \mid \mathbf{x}, \delta = 1) = \frac{1}{1 + e^{-g(\mathbf{x}; \boldsymbol{\theta}) - l}}, \tag{4}$$

where $l = -\log\{\pi(\mathbf{x})\}$. Please see the detailed derivation of (4) in the supplement. To avoid the IPW, we propose the estimator based on the log odds corrected likelihood in (4), namely,

$$\hat{\boldsymbol{\theta}}_{\text{lik}} = \arg\max_{\boldsymbol{\theta}} \ell_{\text{lik}}(\boldsymbol{\theta}), \quad \text{where} \quad \ell_{\text{lik}}(\boldsymbol{\theta}) = \sum_{i=1}^{N} \delta_i \big[ y_i g(\mathbf{x}_i; \boldsymbol{\theta}) - \log\big\{1 + e^{g(\mathbf{x}_i; \boldsymbol{\theta}) + l_i}\big\}\big]. \tag{5}$$

With $\hat{\boldsymbol{\theta}}_w$ in (2), $\pi(\mathbf{x}_i)$ is in the inverse. If an instance with much smaller $\pi(\mathbf{x}_i)$ is selected in the sample, it dominates the objective function, making the resulting estimator unstable. With $\hat{\boldsymbol{\theta}}_{\text{lik}}$, this problem is ameliorated because $\pi(\mathbf{x}_i)$ is in the logarithm in the log-likelihood of (5) and $\log(v)$ goes to infinity much slower than $v^{-1}$ as $v \downarrow 0$.

## 4.2 Theoretical analysis of the likelihood-based estimator

The following shows the asymptotic distribution of $\hat{\boldsymbol{\theta}}_{\text{lik}}$.

**Theorem 4.** *Under Assumptions 1-3, if $\mathbb{E}\{e^{f(\mathbf{x}; \boldsymbol{\beta}^*)}\varphi^{-1}(\mathbf{x})B(\mathbf{x})\} < \infty$, then as $N \to \infty$,*

$$\sqrt{N_1}(\hat{\boldsymbol{\theta}}_{\text{lik}} - \boldsymbol{\theta}) \longrightarrow \mathbb{N}(\mathbf{0}, \mathbf{V}_{\text{lik}}) \quad \text{in distribution,}$$

*where $\mathbf{V}_{\text{lik}} = \mathbb{E}\{e^{f(\mathbf{x}; \boldsymbol{\beta}^*)}\}\boldsymbol{\Lambda}_{\text{lik}}^{-1}$ and $\boldsymbol{\Lambda}_{\text{lik}} = \mathbb{E}\left[\frac{e^{f(\mathbf{x}; \boldsymbol{\beta}^*)}\dot{g}^{\otimes 2}(\mathbf{x}; \boldsymbol{\theta}^*)}{1 + c\varphi^{-1}(\mathbf{x})e^{f(\mathbf{x}; \boldsymbol{\beta}^*)}}\right]$.*

Theorem 3 shows that the proposed estimator $\hat{\boldsymbol{\theta}}_{\text{lik}}$ is asymptotic normal with variance $\mathbf{V}_{\text{lik}}$. We compare $\mathbf{V}_{\text{lik}}$ with $\mathbf{V}_w$ to show that $\hat{\boldsymbol{\theta}}_{\text{lik}}$ has a higher estimation efficiency than $\hat{\boldsymbol{\theta}}_w$.

**Theorem 5.** *If $\mathbf{V}_w$ and $\mathbf{V}_{\text{lik}}$ are finite and positive definite, then $\mathbf{V}_{\text{lik}} \leq \mathbf{V}_w$, i.e., $\mathbf{V}_w - \mathbf{V}_{\text{lik}}$ is non-negative definite. The equality holds when $c = 0$ and in this case $\mathbf{V}_{\text{lik}} = \mathbf{V}_w = \mathbf{V}_f$.*

Since the estimator $\hat{\boldsymbol{\theta}}_{\text{lik}}$ in Eg. (5) is based on the conditional log-likelihood of the sampled data, it actually has the highest estimation efficiency among a class of asymptotically unbiased estimators.

**Theorem 6.** *Let $\mathbf{X} = (\mathbf{x}_1, ..., \mathbf{x}_N)^{\mathrm{T}}$ be the feature matrix and $\mathcal{D}_\delta$ denote the sampled data. Let $\hat{\boldsymbol{\theta}}_U$ be a subsample estimator with the following asymptotic representation:*

$$\hat{\boldsymbol{\theta}}_U = \mathbf{U}(\boldsymbol{\theta}^*; \mathcal{D}_\delta) + N_1^{-1/2} o_P(1), \tag{6}$$

*where the variance $\mathbb{V}\{\mathbf{U}(\boldsymbol{\theta}^*; \mathcal{D}_\delta)\}$ exists, and $\mathbf{U}(\boldsymbol{\theta}^*; \mathcal{D}_\delta)$ satisfies that $\mathbb{E}\{\mathbf{U}(\boldsymbol{\theta}^*; \mathcal{D}_\delta) \mid \mathbf{X}\} = \boldsymbol{\theta}^*$ and $\mathbb{E}\{\dot{\mathbf{U}}(\boldsymbol{\theta}^*; \mathcal{D}_\delta) \mid \mathbf{X}\} = \mathbf{0}$ with $\dot{\mathbf{U}}(\boldsymbol{\theta}^*; \mathcal{D}_\delta) = \partial \mathbf{U}(\boldsymbol{\theta}^*; \mathcal{D}_\delta)/\partial \boldsymbol{\theta}^{*\mathrm{T}}$. If $N_1 \mathbb{V}\{\mathbf{U}(\boldsymbol{\theta}^*; \mathcal{D}_\delta)\} \to \mathbf{V}_U$ in probability, then $\mathbf{V}_U \geq \mathbf{V}_{\text{lik}}$.*

**Remark 2.** Theorem 6 tells us that $\hat{\boldsymbol{\theta}}_{\text{lik}}$ is statistically the most efficient among a class of estimators, which includes both $\hat{\boldsymbol{\theta}}_w$ and $\hat{\boldsymbol{\theta}}_{\text{lik}}$ as special cases. The condition $\mathbb{E}\{\mathbf{U}(\boldsymbol{\theta}^*; \mathcal{D}_\delta) \mid \mathbf{X}\} = \boldsymbol{\theta}^*$ ensures that the estimator is asymptotically unbiased. The condition $\mathbb{E}\{\dot{\mathbf{U}}(\boldsymbol{\theta}^*; \mathcal{D}_\delta) \mid \mathbf{X}\} = \mathbf{0}$ can be intuitively interpreted as the requirement that the derivative of the $o_P(1)$ term in (6) is also negligible. Clearly, this is satisfied by all unbiased estimators for which the $o_P(1)$ term in (6) is $\mathbf{0}$.

## 5 Practical considerations

In this section, we first show how we design a more practical estimator based on the previous results and then present the theoretical analysis behind these improvements.

## 5.1 Making estimators more practical

Like the LCC sampling [8] and the A-optimal sampling [29], the $\varphi_{os}(\mathbf{x})$ in (3) depends on $\boldsymbol{\theta}$, and thus a pilot value of $\boldsymbol{\theta}$, say $\tilde{\boldsymbol{\theta}}$, is required in practice. Here, $\tilde{\boldsymbol{\theta}}$ can be constructed from a pilot sample, say $(\tilde{\mathbf{x}}_i, \tilde{y}_i)$, $i = 1, ..., \tilde{n}$, taken by uniform sampling from both classes with equal expected sample sizes. With $\tilde{\boldsymbol{\theta}}$ obtained, calculate $\tilde{t}_i = p(\tilde{\mathbf{x}}_i; \tilde{\boldsymbol{\theta}})\|\tilde{\mathbf{M}}_f^{-1}\dot{g}(\tilde{\mathbf{x}}_i; \tilde{\boldsymbol{\theta}})\|$ for $i = 1, ..., \tilde{n}$, where $\tilde{\mathbf{M}}_f$ is the Hessian matrix of the pilot sample objective function. In case that the Hessian matrix and gradients are hard to record, the gradient term can be dropped, i.e., use $\tilde{t}_i = p(\tilde{\mathbf{x}}; \tilde{\boldsymbol{\theta}})$. As mentioned in Remark 1, $T$ can be ignored for very imbalanced data. The expectation in the denominator of (3) can be approximated by mimicking the method of moment estimator, namely by

$$\tilde{\omega} = 2N_1(\tilde{n}N)^{-1}\sum_{\tilde{y}_i=1}\tilde{t}_i + 2N_0(\tilde{n}N)^{-1}\sum_{\tilde{y}_i=0}\tilde{t}_i. \tag{7}$$

In practice, it is common to set a lower threshold, say $\varrho$, on sampling probabilities, to ensure that all instances have positive probabilities to be selected. This is more critical for the IPW estimator while the likelihood estimator is not very sensitive to very small sampling probabilities.

Denote the pilot value as $\tilde{\boldsymbol{\vartheta}} = (\tilde{\boldsymbol{\theta}}, \tilde{\omega})$. The following probabilities can be practically implemented

$$\pi_\varrho^{os}(\mathbf{x}; \tilde{\boldsymbol{\vartheta}}) = \min[\max\{\rho\tilde{\varphi}_{os}(\mathbf{x}), \varrho\}, 1], \quad \text{where} \quad \tilde{\varphi}_{os}(\mathbf{x}) = \tilde{\omega}^{-1}t(\mathbf{x}; \tilde{\boldsymbol{\theta}}), \tag{8}$$

and we use this in our experiments in Section 6. The $\tilde{\omega}$ in (7) is essentially a normalizing term so that the expected subsample size is $\rho$. For some practical systems such as online models, $\tilde{\omega}$ is often treated as a tuning parameter. We will illustrated this in our real data example in Section 6.2.

## 5.2 Theoretical analysis of practical estimators

Denote $\hat{\boldsymbol{\theta}}_w^{\tilde{\vartheta}}$ and $\hat{\boldsymbol{\theta}}_{lik}^{\tilde{\vartheta}}$ the IPW estimator and the likelihood estimator, respectively, with practically estimated optimal probability in (8). We have the following results.

**Theorem 7.** *Assume that the pilot estimator $\tilde{\boldsymbol{\vartheta}}$ is independent of the data, $\tilde{\varphi}_{os}(\mathbf{x}) \to \varphi_{plt}(\mathbf{x})$ in probability, and $\rho^{-1}\varrho \to c_l > 0$. Under Assumptions 1-3, as $N \to \infty$, the $\hat{\boldsymbol{\theta}}_w^{\tilde{\vartheta}}$ satisfies that,*

$$\sqrt{N_1}(\hat{\boldsymbol{\theta}}_w^{\tilde{\vartheta}} - \boldsymbol{\theta}) \to \mathbb{N}(\mathbf{0}, \mathbf{V}_w^{plt}) \quad \text{in distribution,}$$

*where $\mathbf{V}_w^{plt}$ has the same expression of $\mathbf{V}_w$ except that $\varphi(\mathbf{x})$ is replaced by $\max\{\varphi_{plt}(\mathbf{x}), c_l\}$. If $\Pr\{\varphi_{plt}(\mathbf{x}) \geq c_l\} = 1$ and the pilot estimator is consistent, then $\mathbf{V}_w^{plt}$ achieves the optimal variance.*

**Theorem 8.** *Assume that the pilot estimator $\tilde{\boldsymbol{\vartheta}}$ is independent of the data, $\tilde{\varphi}_{os}(\mathbf{x}) \to \varphi_{plt}(\mathbf{x})$ in probability, and $\varrho = o(\rho)$. Under Assumptions 1-3, as $N \to \infty$, $\hat{\boldsymbol{\theta}}_{lik}^{\tilde{\vartheta}}$ satisfies that,*

$$\sqrt{N_1}(\hat{\boldsymbol{\theta}}_{lik}^{\tilde{\vartheta}} - \boldsymbol{\theta}) \to \mathbb{N}(\mathbf{0}, \mathbf{V}_{lik}^{plt}) \quad \text{in distribution,}$$

*where $\mathbf{V}_{lik}^{plt}$ has the same expression of $\mathbf{V}_{lik}$ except that $\varphi(\mathbf{x})$ is replaced by $\varphi_{plt}(\mathbf{x})$. If the pilot estimator is consistent, then $\mathbf{V}_{lik}^{plt}$ achieves the variance with the optimal probability.*

**Remark 3.** Theorems 7 and 8 do not require $\tilde{\boldsymbol{\vartheta}}$ to be consistent, i.e., the pilot estimator can be misspecified. However $\tilde{\boldsymbol{\vartheta}}$ has to be consistent in order to achieve the asymptotic variances with the optimal probability. Compared with the likelihood estimator $\hat{\boldsymbol{\theta}}_{lik}^{\tilde{\vartheta}}$, the IPW estimator $\hat{\boldsymbol{\theta}}_w^{\tilde{\vartheta}}$ requires a stronger condition on $\varrho$ to have asymptotic normality, because $\pi_\varrho^{os}(\mathbf{x}; \tilde{\boldsymbol{\vartheta}})$ is in the denominator of the objective function for $\hat{\boldsymbol{\theta}}_w^{\tilde{\vartheta}}$ so the turbulence of $\tilde{\boldsymbol{\vartheta}}$ is amplified. The requirement $c_l > 0$ means that $\varrho$ cannot be too small in practice. The likelihood estimator does not have this constraint.

## 6 Numerical experiments

We present simulated results and report the performance on a CTR dataset with more than 0.3 trillion instances. More experiments with model and pilot misspecifications are available in the supplement.

## 6.1 Experiments on simulated data

We implemented the following negative sampling methods for logistic regression. 1) uniW: uniform sampling with the IPW estimator. 2) uniLik: uniform sampling with the likelihood estimator. 3) optW: optimal sampling with the IPW estimator. 4) optLik: optimal sampling with the likelihood estimator. We also implement the full data MLE (Full) and the LCC for comparisons.

In each repetition of the simulation, we generate full data of size $N = 5 \times 10^5$ from the logistic regression with $g(\mathbf{x}; \boldsymbol{\theta}) = \alpha + \mathbf{x}^{\mathrm{T}}\boldsymbol{\beta}$. We set the true $\boldsymbol{\beta}^*$ to be a $6 \times 1$ vector of ones and set different values for $\alpha$ for different feature distributions so that the positive/negative ratio is close to 1:400. We consider the following feature distributions: (a) Normal distribution which is symmetric with light tails; the true intercept is $\alpha = -7.65$. (b) Log-normal distribution which is asymmetric and positively skewed; the true intercept is $\alpha = -0.5$. (c) $t_3$ distribution which is symmetric with heavier tails; the true intercept is $\alpha = -7$.

We set the sampling rate as $\rho = 0.002, 0.004, 0.006, 0.01$, and $0.02$ for all sampling methods. In each repetition, uniform samples of average size 100 are selected from each class to calculate the pilot estimates, so the uncertainty due to the pilot estimates are taken into account. We also consider pilot misspecification by adding a uniform random number from $\mathbb{U}(0, 1.5)$ to the pilot so that the pilot is systematically different from the true parameter. We repeat the simulation for $R = 1000$ times to calculate the MSE as $R^{-1}\sum_{r=1}^{R} \|\hat{\boldsymbol{\theta}}^{(r)} - \boldsymbol{\theta}\|^2$, where $\boldsymbol{\theta}^{(r)}$ is the estimate at the $r$-th repetition.

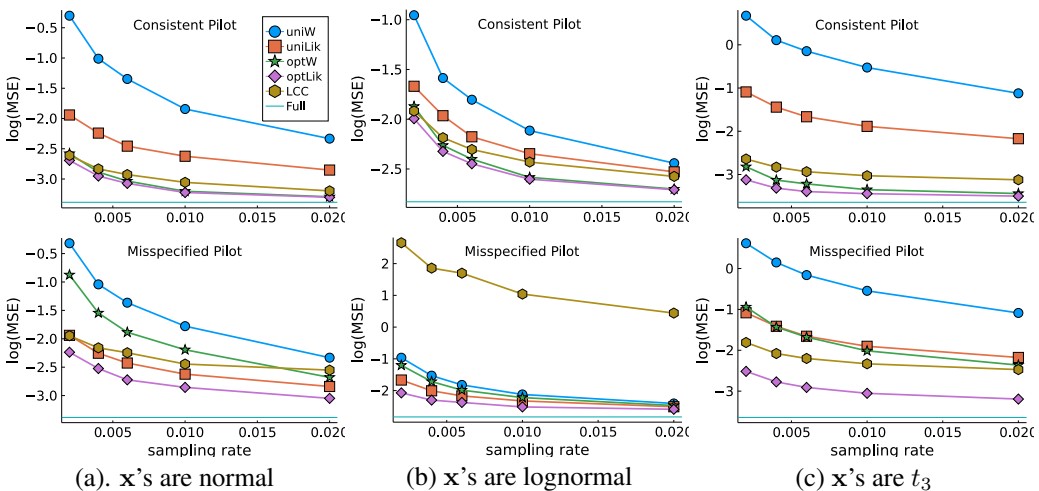

Figure 1: Log(MSEs) for different estimators (the smaller the better). The top row uses consistent pilot estimators; the bottom uses misspecified pilot estimators.

Results are presented in Figure 1. The MSEs of all estimators decrease as the sampling rate $\rho$ increases. The optLik outperforms other sampling methods in general, and its performance is close to the full data MLE when $\rho = 0.02$, especially if the pilot is consistent. The advantage of optLik over optW is more evident for smaller $\rho$. If the pilot is misspecified, optLik is quite robust but optLik and LCC significantly deteriorate. Since uniLik and uniW do not require a pilot, they are not affected by pilot misspecification, but they have much larger MSE than optLik and optW when the pilot is consistent, especially uniW. Different feature distributions also affect the performances of different sampling methods.

To verify Theorem 1 numerically, we let the full data size $N$ increase at a faster rate than the average number of positive instances $N_1^a = \mathbb{E}(N_1)$ so that the probability $\Pr(y = 1)$ decreases towards zero. As $N$ increases, we fixed the value of $\boldsymbol{\beta}^*$ and set decreasing values to $\alpha^*$ to mimic our scaling regime. We considered two covariate distributions: 1) a multivariate normal distribution for which a logistic regression model is a correct model; and 2) a multivariate log-normal distribution for which a logistic regression model is misspecified. We simulated for 100 times to calculate the empirical variance $\hat{\mathbf{V}}_f$ of the full data MLE, and report the results in Table 1. It is seen that $\mathrm{tr}(\hat{\mathbf{V}}_f)$ is decreasing towards zero. According to Theorem 1, $\mathrm{tr}(\hat{\mathbf{V}}_f)$ should converge in a rate of $1/N_1$, so $N_1^a\mathrm{tr}(\hat{\mathbf{V}}_f)$ should be relatively stable as $N$ increases. This is indeed the case as seen in the third and sixth

columns of Table 1. On the other hand, $N\mathrm{tr}(\hat{\mathbf{V}}_f)$ gets large dramatically as $N$ increases. The aforementioned observations confirm the theoretical result in Theorem 1. Furthermore, Table 1 shows that the convergence rate in Theorem 1 may also be true for some misspecified models.

Table 1: Empirical variances of the full data MLE for different full data size and average number of positive instances combinations.

| $(N, N_1^a)$ | Correct model | | | Mis-sprcified model | | |
| --- | --- | --- | --- | --- | --- | --- |
| | $\mathrm{tr}(\hat{\mathbf{V}}_f)$ | $N_1^a\mathrm{tr}(\hat{\mathbf{V}}_f)$ | $N\mathrm{tr}(\hat{\mathbf{V}}_f)$ | $\mathrm{tr}(\hat{\mathbf{V}}_f)$ | $N_1^a\mathrm{tr}(\hat{\mathbf{V}}_f)$ | $N\mathrm{tr}(\hat{\mathbf{V}}_f)$ |
| $(10^3,\ 32)$ | 0.169 | 5.41 | 169.17 | 0.969 | 30.99 | 968.70 |
| $(10^4,\ 64)$ | 0.097 | 6.20 | 969.29 | 0.322 | 20.59 | 3217.12 |
| $(10^5, 128)$ | 0.045 | 5.76 | 4497.24 | 0.135 | 17.32 | 13527.60 |
| $(10^6, 256)$ | 0.018 | 4.62 | 18048.40 | 0.046 | 11.74 | 45847.40 |

## 6.2 Experiments on real data

We conduct experiments on an internal CTR dataset from our products with more than 0.3 trillion instances. There are 299 input features, including user and ad profiles together with rich context features. We concatenate all features as a single high-dimensional vector $\mathbf{x}$, whose length is greater than $5,000$. We use a 3-layer fully connected deep neural network as a feature extractor that maps $\mathbf{x}$ into a 256-dimensional dense features $h(\mathbf{x})$. The downstream model is a linear logistic regression model taking $h(\mathbf{x})$ as input to predict the binary output whether the user clicks the ad or not.

The positive/negative ratio is around 1:80. Due to limited storage, we first uniformly drop 80% negative instances and try various negative sampling algorithms on the rest 60 billion instances. On this pre-processed dataset, we then split out 1 percent of the data for testing, and do negative sampling on the rest data. The entire training procedure scans the subsampled data set from begin to end in one-pass according to the timestamp. The testing is done all the way through training. We calculate the area under the ROC curve (AUC) for testing instances. The entire model is trained on a distributed machine learning platform using 1,200 CPU cores and can finish within two days.

We adopt the sampling probability $\pi_\varrho^{\mathrm{os}}(\mathbf{x}; \tilde{\boldsymbol{\vartheta}})$ as demonstrated in (8) and use the empirical sampling rate to calibrate the normalizing term $\tilde{\omega}$. As mentioned before, we remove the gradient term for implementation tractability so $t(\mathbf{x}; \tilde{\boldsymbol{\theta}}) = p(\mathbf{x}; \tilde{\boldsymbol{\theta}})$. As the result will show, this approximate score produces consistent results with the simulation experiments. We compare four methods: 1) uniW and 2) uniLik are the same as in simulated experiments; and 3) optW and 4) optLik represent nonuniform sampling probability using $\pi_\varrho^{\mathrm{os}}(\mathbf{x}; \tilde{\boldsymbol{\vartheta}})$ with the IPW estimator and with the likelihood estimator. We use "opt" to refer to both optW and optLik when focusing on the sampling.

We study two sets of negative sample rates (w.r.t. the original 0.3 trillion data): (i) $[0.01, 0.05]$. In this case, negative instances still outnumber positive instances. (ii) $[0.001, 0.008]$. In this case, negative subsampling is too aggressive that positive instances dominates negative instances. We demonstrate the result in Figure 2. When the sample rate is moderate (Figure 2.(b)), opt is nearly optimal and can significantly outperform uniform subsampling. There is still a small gap between the IPW estimator and the likelihood estimator. When the sample rate is extremely small (Figure 2.(a)), opt becomes sub-optimal and its gap w.r.t. uniform sampling is closer. Still we see a clear difference between the IPW estimator and the likelihood estimator. This is due to a larger asymptotic variance for uniform sampling when the sampling rate goes to zero. Note that on this huge data set, a small relative AUC gain (e.g., 0.0005) usually could bring about significant revenue gain in products.

## 6.3 Sensitivity analysis

In practice, we find it necessary to use cross-validation to find the optimal $\varrho$ in (8) given a fixed sample rate. As demonstrated in Figure 3, for each sampling rate we tune $\varrho$ from a very small value (in this case, $10^{-6}$) all the way to its largest value. When $\varrho$ achieves its largest value, $\rho\tilde{\omega}^{-1} = 0$ and $\pi_\varrho^{\mathrm{os}}(\mathbf{x}; \tilde{\boldsymbol{\vartheta}})$ reduces to uniform negative sampling. We observe that $\pi_\varrho^{\mathrm{os}}(\mathbf{x}; \tilde{\boldsymbol{\vartheta}})$ with a moderate $\varrho$ achieves the best results for various sampling rates. Note that $\varrho$ is the lower truncation level for negative sampling probabilities. Thus, a larger $\varrho$ means more "simple" negative instances ($\rho\tilde{\omega}^{-1}p(\mathbf{x}_i; \tilde{\boldsymbol{\theta}})$ is smaller than

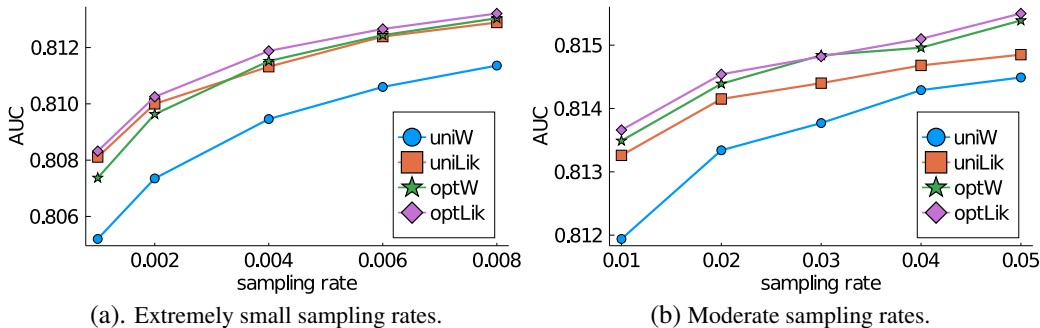

(a). Extremely small sampling rates.  (b) Moderate sampling rates.

Figure 2: Empirical testing AUC of subsample estimators for different sample sizes (the larger the better).

$\varrho$) will have better chances to be selected. This coincides with recent empirical results for negative subsampling in large search systems [14], where they mix "simple" (smaller $p(\mathbf{x}_i; \tilde{\boldsymbol{\theta}})$) and "hard" (larger $p(\mathbf{x}_i; \tilde{\boldsymbol{\theta}})$) negative instances. We also observe an empirical variance of around $0.0001$ for each setup, demonstrating that the relative improvement is consistent.

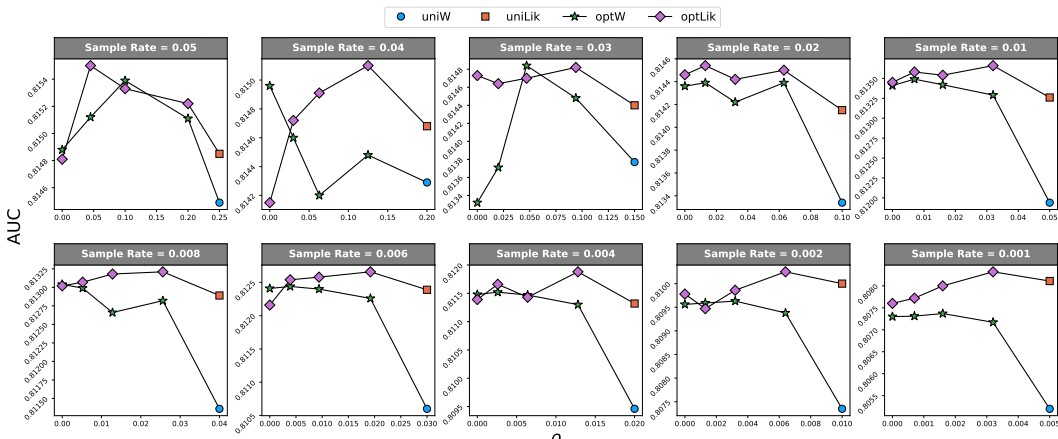

Figure 3: AUC for IPW and likelihood estimators when tuning the truncation lower bound $\varrho$.

# 7 Conclusions, discussion, and limitations

In this paper, we have derived asymptotic distributions for full data and subsample estimators with rare events data. We have also found the optimal sampling probability that minimizes the unconditional asymptotic variance of the IPW estimator, and proposed the likelihood estimator through nonuniform log odds correction to further improve the estimation efficiency. The optimal probability depends on unknown parameters so we have discussed practical implementations and investigated the convergences of the resultant estimators. Experiments on both simulated and real big online streaming data confirm our theoretical findings. Heuristically, our nonuniform negative sampling method gives preference to data points that are harder to observe, so it would give preference to sub-populations that are difficult to observe. We do not think this would bring any negative societal impacts. We have also examined our real data application and did not notice discrimination against any sub-populations.

This paper has the following limitations: 1) We assume that the model is correctly specified. Theoretical properties with model misspecification is not considered, and this important question requires future studies. 2) Oversampling the rare positive instances is another common practice to deal with imbalanced data. Its application together with negative sampling is not considered. 3) The assumption of a fixed $\boldsymbol{\beta}^*$ means that both the marginal and conditional probabilities for a positive instance to occur are small. If the proportion of the positive instances for a given feature in a dataset is large,

this assumption may not be appropriate. Further investigation is need to see if our results is still applicable. In the following, we present three scenarios that the scaling regime may or may not fit in:

Scenario 1 (phone call while driving) Car crashes occur with small probabilities, and making phone calls while driving significantly increases the probability of a car crash. However, the probability of car crashes among people making phone calls while driving is still small, so for these types of features, our scaling regime is appropriate to model the rare events.

Scenario 2 (anti-causal learning) Anti-causal learning [26, 16] assumes that label ($y$) causes observations ($x$). Thus $\Pr(x|y)$ represents the causal mechanism and is fixed. One standard example is that diseases ($y$) cause symptoms ($x$). Our scaling regime fits the framework of anti-causal learning. To see this, using Bayes Theorem we write the log odds as

$$\alpha + f(\mathbf{x}; \boldsymbol{\beta}) = \log \frac{\Pr(y = 1)}{\Pr(y = 0)} + \log \frac{\Pr(\mathbf{x} \mid y = 1)}{\Pr(\mathbf{x} \mid y = 0)}.$$

In anti-causal learning, the marginal distribution of $y$ changes while the conditional distribution of $\mathbf{x}$ given $y$ is fixed. Thus only the scale factor $\alpha$ changes, and $f(\mathbf{x}; \boldsymbol{\beta})$ is fixed.

Scenario 3 (local transmission of the COVID-19) Our scaling regime has some limitations and does not apply to all types of rare events data. For example, although the COVID-19 infection rate for the whole population is low, the infection rate for people whose family members are in the same house with positive test results is high. This means the change of a family member's test result converts a small-probability-event to a large-probability-event, and our scaling regime would not be appropriate.

## Acknowledgments and Disclosure of Funding

HaiYing Wang's research was partially supported by NSF grant CCF 2105571.

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
