# Proofs and Additional Numerical Experiments for "Nonuniform Negative Sampling and Log Odds Correction with Rare Events Data"

Before presenting the proof, we point out that Assumption 2 implies that $\mathbb{E}\{v(\mathbf{x})\dot{g}^{\otimes 2}(\mathbf{x};\boldsymbol{\theta})\}$ is positive definite for a positive function $v(\mathbf{x}) > 0$ almost surely. This is because Assumption 2 means that $\Pr\{l^{\mathrm{T}}\dot{g}(\mathbf{x};\boldsymbol{\theta}) \neq 0\} > 0$ for any $l \neq \mathbf{0}$, and therefore $\Pr\{v^{1/2}(\mathbf{x})l^{\mathrm{T}}\dot{g}(\mathbf{x};\boldsymbol{\theta}) \neq 0\} > 0$ for any $l \neq \mathbf{0}$, implying that $\mathbb{E}\{v(\mathbf{x})\dot{g}^{\otimes 2}(\mathbf{x};\boldsymbol{\theta})\}$ is positive definite. We also point out that if a sequence is $o_P(1)$ conditionally then it is also $o_P(1)$ unconditionally and vice versa [1, 2].

To facilitate the presentation, define $a_N = \sqrt{Ne^{\alpha^*}}$. We notice that

$$N_1 = N\mathbb{E}\left\{\frac{e^{\alpha^*+f(\mathbf{x};\boldsymbol{\beta}^*)}}{1+e^{\alpha^*+f(\mathbf{x};\boldsymbol{\beta}^*)}}\right\}\{1+o_P(1)\} = a_N^2\mathbb{E}\{e^{f(\mathbf{x};\boldsymbol{\beta}^*)}\}\{1+o_P(1)\}, \qquad \text{(S.1)}$$

from the dominated convergence theorem. Thus the normalizing term $\sqrt{N_1}$ for the asymptotic normality in the paper can be replaced by $a_N\mathbb{E}^{1/2}\{e^{f(\mathbf{x};\boldsymbol{\beta}^*)}\}$.

## S.1  Proof of Theorem 1

The estimator $\hat{\boldsymbol{\theta}}_{\mathrm{f}}$ is the maximizer of

$$\ell(\boldsymbol{\theta}) = \sum_{i=1}^{N}\left[y_i g(\mathbf{x}_i;\boldsymbol{\theta}) - \log\{1+e^{g(\mathbf{x}_i;\boldsymbol{\theta})}\}\right],$$

so $a_N(\hat{\boldsymbol{\theta}} - \boldsymbol{\theta}^*)$ is the maximizer of

$$\gamma(\mathbf{u}) = \ell(\boldsymbol{\theta}^* + a_N^{-1}\mathbf{u}) - \ell(\boldsymbol{\theta}^*).$$

By Taylor's expansion,

$$\gamma(\mathbf{u}) = a_N^{-1}\mathbf{u}^{\mathrm{T}}\dot{\ell}(\boldsymbol{\theta}^*) + 0.5a_N^{-2}\sum_{i=1}^{N}\phi(\mathbf{x}_i;\boldsymbol{\theta}^*)\{\mathbf{u}^{\mathrm{T}}\dot{g}(\mathbf{x}_i;\boldsymbol{\theta}^*)\}^2 + \Delta_\ell + R,$$

where

$$\dot{\ell}(\boldsymbol{\theta}) = \frac{\partial\ell(\boldsymbol{\theta})}{\partial\boldsymbol{\theta}} = \sum_{i=1}^{N}\{y_i - p(\mathbf{x}_i;\boldsymbol{\theta})\}\dot{g}(\mathbf{x}_i;\boldsymbol{\theta}),$$

$$\Delta_\ell = \frac{1}{2a_N^2}\sum_{i=1}^{N}\{y_i - p(\mathbf{x}_i;\boldsymbol{\theta}^*)\}\mathbf{u}^{\mathrm{T}}\ddot{g}(\mathbf{x}_i;\boldsymbol{\theta}^*)\mathbf{u},$$

and $R$ is the remainder term. By direct calculation,

$$R = \frac{1}{6a_N^3}\sum_{i=1}^{N}\phi(\mathbf{x}_i;\boldsymbol{\theta}^* + a_N^{-1}\acute{\mathbf{u}})\{1 - 2p(\mathbf{x}_i;\boldsymbol{\theta}^* + a_N^{-1}\acute{\mathbf{u}})\}\{\mathbf{u}^{\mathrm{T}}\dot{g}(\mathbf{x}_i;\boldsymbol{\theta}^* + a_N^{-1}\acute{\mathbf{u}})\}^3$$

35th Conference on Neural Information Processing Systems (NeurIPS 2021).

$$+\frac{1}{6a_N^3}\sum_{i=1}^{N}\phi(\mathbf{x}_i;\boldsymbol{\theta}^*+a_N^{-1}\acute{\mathbf{u}})\{\mathbf{u}^{\mathrm{T}}\dot{g}(\mathbf{x}_i;\boldsymbol{\theta}^*+a_N^{-1}\acute{\mathbf{u}})\}\{\mathbf{u}^{\mathrm{T}}\ddot{g}(\mathbf{x}_i;\boldsymbol{\theta}^*+a_N^{-1}\acute{\mathbf{u}})\mathbf{u}\}$$

$$+\frac{1}{6a_N^3}\sum_{i=1}^{N}\left[\{y_i-p(\mathbf{x}_i;\boldsymbol{\theta}^*+a_N^{-1}\acute{\mathbf{u}})\}\sum_{j_1,j_2=1}^{d}u_{j_1}u_{j_2}\,\dddot{g}_{j_1j_2}(\mathbf{x}_i;\boldsymbol{\theta}^*+a_N^{-1}\acute{\mathbf{u}})\mathbf{u}\right],$$

where $\dddot{g}_{j_1j_2}(\mathbf{x};\boldsymbol{\theta})$ is the gradient of $\ddot{g}_{j_1j_2}(\mathbf{x},\boldsymbol{\theta})$ and $\acute{\mathbf{u}}$ lies between $\mathbf{0}$ and $\mathbf{u}$. We see that

$$|R|\leq\frac{\|\mathbf{u}\|^3}{6a_N^3}\sum_{i=1}^{N}p(\mathbf{x}_i;\boldsymbol{\theta}^*+a_N^{-1}\acute{\mathbf{u}})C(\mathbf{x}_i,\boldsymbol{\theta}^*+a_N^{-1}\acute{\mathbf{u}})+\frac{d\|\mathbf{u}\|^3}{6a_N^3}\sum_{i=1}^{N}y_iB(\mathbf{x}_i)$$

$$\leq\frac{\|\mathbf{u}\|^3 e^{a_N^{-1}\|\mathbf{u}\|}}{6Na_N}\sum_{i=1}^{N}\exp\{f(\mathbf{x}_i;\boldsymbol{\beta}+a_N^{-1}\acute{\mathbf{u}}_{(-1)})\}C(\mathbf{x}_i,\boldsymbol{\theta}^*+a_N^{-1}\acute{\mathbf{u}})+\frac{d\|\mathbf{u}\|^3}{6a_N^3}\sum_{i=1}^{N}y_iB(\mathbf{x}_i)$$

$$\leq\frac{d^3\|\mathbf{u}\|^3 e^{a_N^{-1}\|\mathbf{u}\|}}{6Na_N}\sum_{i=1}^{N}B(\mathbf{x}_i)+\frac{d\|\mathbf{u}\|^3}{6a_N^3}\sum_{i=1}^{N}y_iB(\mathbf{x}_i)\equiv\Delta_{R1}+\Delta_{R2},\tag{S.2}$$

where $C(\mathbf{x},\boldsymbol{\theta})=\|\dot{g}(\mathbf{x},\boldsymbol{\theta})\|^3+\|\dot{g}(\mathbf{x},\boldsymbol{\theta})\|\|\ddot{g}(\mathbf{x},\boldsymbol{\theta})\|+\sum_{j_1j_2}^{d}\|\dddot{g}_{j_1j_2}(\mathbf{x},\boldsymbol{\theta})\|$ and $\acute{\mathbf{u}}_{(-1)}$ is $\acute{\mathbf{u}}$ with the first element removed. From Assumption 1, $\mathbb{E}(\Delta_{R1})\to 0$ and

$$\mathbb{E}(\Delta_{R2})\leq\frac{d\|\mathbf{u}\|^3}{6Na_N}\sum_{i=1}^{N}\mathbb{E}\{e^{f(\mathbf{x}_i;\boldsymbol{\beta}^*)}B(\mathbf{x}_i)\}=\frac{d\|\mathbf{u}\|^3}{6a_N}\mathbb{E}\{e^{f(\mathbf{x};\boldsymbol{\beta}^*)}B(\mathbf{x})\}\to 0.$$

Since $\Delta_{R1}$ and $\Delta_{R2}$ are both positive, Markov's inequality shows that they are both $o_P(1)$ so $R=o_P(1)$. For $\Delta_\ell$, the mean $\mathbb{E}(\Delta_\ell)=\mathbf{0}$ and the variance satisfies that

$$\mathbb{V}(\Delta_\ell)\leq\frac{\|\mathbf{u}\|^4}{4a_N^4}\sum_{i=1}^{N}\mathbb{E}\{p(\mathbf{x}_i;\boldsymbol{\theta}^*)\|\ddot{g}(\mathbf{x}_i;\boldsymbol{\theta}^*)\|^2\}\leq\frac{\|\mathbf{u}\|^4}{4Na_N^2}\sum_{i=1}^{N}\mathbb{E}[e^{f(\mathbf{x}_i;\boldsymbol{\beta}^*)}\|\ddot{g}(\mathbf{x}_i;\boldsymbol{\theta}^*)\|^2]$$

$$=\frac{\|\mathbf{u}\|^4}{4a_N^2}\mathbb{E}[e^{f(\mathbf{x};\boldsymbol{\beta}^*)}\|\ddot{g}(\mathbf{x},\boldsymbol{\theta}^*)\|^2]\to 0,$$

so $\Delta_\ell=o_P(1)$.

If we can show that

$$a_N^{-1}\dot{\ell}(\boldsymbol{\theta}^*)\longrightarrow\mathbb{N}(\mathbf{0},\,\mathbf{M}_{\mathrm{f}}),\tag{S.3}$$

in distribution, and

$$a_N^{-2}\sum_{i=1}^{N}\phi(\mathbf{x}_i;\boldsymbol{\theta}^*)\dot{g}^{\otimes 2}(\mathbf{x}_i;\boldsymbol{\theta}^*)\longrightarrow\mathbf{M}_{\mathrm{f}},\tag{S.4}$$

in probability, then from the Basic Corollary in page 2 of [3], we know that $a_N(\hat{\boldsymbol{\theta}}-\boldsymbol{\theta}^*)$, the maximizer of $\gamma(\mathbf{u})$, satisfies that

$$a_N(\hat{\boldsymbol{\theta}}-\boldsymbol{\theta}^*)=\mathbf{M}_{\mathrm{f}}^{-1}\times a_N^{-1}\dot{\ell}(\boldsymbol{\theta}^*)+o_P(1).\tag{S.5}$$

Slutsky's theorem together with (S.3) and (S.5) implies the result in Theorem 1. We prove (S.3) and (S.4) in the following.

Note that

$$\dot{\ell}(\boldsymbol{\theta}^*)=\sum_{i=1}^{N}\{y_i-p(\mathbf{x}_i;\boldsymbol{\theta}^*)\}\dot{g}(\mathbf{x}_i;\boldsymbol{\theta}^*)$$

is a summation of i.i.d. quantities. Since the distribution of $\{y-p(\mathbf{x};\boldsymbol{\theta}^*)\}\dot{g}(\mathbf{x};\boldsymbol{\theta}^*)$ depends on $N$ because $\alpha^*\to-\infty$ as $N\to\infty$, we need to use the Lindeberg-Feller central limit theorem for triangular arrays [see, Section *2.8 of 4].

We examine the mean and variance of $a_N^{-1}\dot{\ell}(\boldsymbol{\theta}^*)$. For the mean, from the fact that

$$\mathbb{E}[\{y_i-p(\mathbf{x}_i;\boldsymbol{\theta}^*)\}\dot{g}(\mathbf{x}_i;\boldsymbol{\theta}^*)]=\mathbb{E}\Big[\mathbb{E}\{y_i-p(\mathbf{x}_i;\boldsymbol{\theta}^*)\mid\mathbf{x}_i\}\dot{g}(\mathbf{x}_i;\boldsymbol{\theta}^*)\mathbf{x}_i\Big]=\mathbf{0},$$

we know that $\mathbb{E}\{a_N^{-1}\dot{\ell}(\boldsymbol{\theta}^*)\} = \mathbf{0}$.

For the variance,

$$\mathbb{V}\{a_N^{-1}\dot{\ell}(\boldsymbol{\theta}^*)\} = a_N^{-2}\sum_{i=1}^{N}\mathbb{V}[\{y - p(\mathbf{x};\boldsymbol{\theta}^*)\}\dot{g}(\mathbf{x}_i;\boldsymbol{\theta}^*)] = a_N^{-2}N\mathbb{E}\{\phi(\boldsymbol{\theta}^*)\dot{g}^{\otimes}(\mathbf{x};\boldsymbol{\theta}^*)\}$$

$$= a_N^{-2}N\mathbb{E}\left[\frac{e^{\alpha^*+f(\mathbf{x};\boldsymbol{\beta}^*)}}{\{1+e^{\alpha^*+f(\mathbf{x};\boldsymbol{\beta}^*)}\}^2}\dot{g}^{\otimes}(\mathbf{x};\boldsymbol{\theta}^*)\right] = \mathbb{E}\left[\frac{e^{f(\mathbf{x};\boldsymbol{\beta}^*)}}{\{1+e^{\alpha^*+f(\mathbf{x};\boldsymbol{\beta}^*)}\}^2}\dot{g}^{\otimes}(\mathbf{x};\boldsymbol{\theta}^*)\right].$$

We have

$$\frac{e^{f(\mathbf{x};\boldsymbol{\beta}^*)}}{\{1+e^{\alpha^*+f(\mathbf{x};\boldsymbol{\beta}^*)}\}^2}\dot{g}^{\otimes}(\mathbf{x};\boldsymbol{\theta}^*) \longrightarrow e^{f(\mathbf{x};\boldsymbol{\beta}^*)}\dot{g}^{\otimes}(\mathbf{x};\boldsymbol{\theta}^*) \quad \text{almost surely,}$$

and

$$\frac{e^{f(\mathbf{x};\boldsymbol{\beta}^*)}}{\{1+e^{\alpha^*+f(\mathbf{x};\boldsymbol{\beta}^*)}\}^2}\|\dot{g}(\mathbf{x};\boldsymbol{\theta}^*)\|^2 \le e^{f(\mathbf{x};\boldsymbol{\beta}^*)}\|\dot{g}(\mathbf{x};\boldsymbol{\theta}^*)\|^2 \quad \text{with} \quad \mathbb{E}\{e^{f(\mathbf{x};\boldsymbol{\beta}^*)}\|\dot{g}(\mathbf{x};\boldsymbol{\theta}^*)\|^2\} \le \infty.$$

Note that $\dot{g}(\mathbf{x};\boldsymbol{\theta}^*) = \{1, \dot{f}^{\mathrm{T}}(\mathbf{x}_i;\boldsymbol{\beta}^*)\}^{\mathrm{T}}$ does not depend on $N$. Thus, from the dominated convergence theorem,

$$\mathbb{V}\{a_N^{-1}\dot{\ell}(\boldsymbol{\theta}^*)\} \longrightarrow \mathbb{E}\{e^{f(\mathbf{x};\boldsymbol{\beta}^*)}\dot{g}^{\otimes}(\mathbf{x};\boldsymbol{\theta}^*)\}.$$

Now we check the Lindeberg-Feller condition. For any $\epsilon > 0$,

$$\sum_{i=1}^{N}\mathbb{E}\left[\|\{y_i - p(\mathbf{x}_i;\boldsymbol{\theta}^*)\}\dot{g}(\mathbf{x}_i;\boldsymbol{\theta}^*)\|^2 I(\|\{y_i - p(\mathbf{x}_i;\boldsymbol{\theta}^*)\}\dot{g}(\mathbf{x}_i;\boldsymbol{\theta}^*)\| > a_N\epsilon)\right]$$

$$= N\mathbb{E}\left[\|\{y - p(\mathbf{x};\boldsymbol{\theta}^*)\}\dot{g}(\mathbf{x};\boldsymbol{\theta}^*)\|^2 I(\|\{y - p(\mathbf{x};\boldsymbol{\theta}^*)\}\dot{g}(\mathbf{x};\boldsymbol{\theta}^*)\| > a_N\epsilon)\right]$$

$$= N\mathbb{E}\left[p(\mathbf{x};\boldsymbol{\theta}^*)\{1 - p(\mathbf{x};\boldsymbol{\theta}^*)\}^2\|\dot{g}(\mathbf{x};\boldsymbol{\theta}^*)\|^2 I(\|\{1 - p(\mathbf{x};\boldsymbol{\theta}^*)\}\dot{g}(\mathbf{x};\boldsymbol{\theta}^*)\| > a_N\epsilon)\right]$$

$$\quad + N\mathbb{E}\left[\{1 - p(\mathbf{x};\boldsymbol{\theta}^*)\}\{p(\mathbf{x};\boldsymbol{\theta}^*)\}^2\|\dot{g}(\mathbf{x};\boldsymbol{\theta}^*)\|^2 I(\|p(\mathbf{x};\boldsymbol{\theta}^*)\dot{g}(\mathbf{x};\boldsymbol{\theta}^*)\| > a_N\epsilon)\right]$$

$$\le N\mathbb{E}\left[p(\mathbf{x};\boldsymbol{\theta}^*)\|\dot{g}(\mathbf{x};\boldsymbol{\theta}^*)\|^2 I(\|\dot{g}(\mathbf{x};\boldsymbol{\theta}^*)\| > a_N\epsilon)\right]$$

$$\quad + N\mathbb{E}\left[\{p(\mathbf{x};\boldsymbol{\theta}^*)\}^2\|\dot{g}(\mathbf{x};\boldsymbol{\theta}^*)\|^2 I(\|p(\mathbf{x};\boldsymbol{\theta}^*)\dot{g}(\mathbf{x};\boldsymbol{\theta}^*)\| > a_N\epsilon)\right]$$

$$\le a_N^2\mathbb{E}\{e^{f(\mathbf{x};\boldsymbol{\beta}^*)}\|\dot{g}(\mathbf{x};\boldsymbol{\theta}^*)\|^2 I(\|\dot{g}(\mathbf{x};\boldsymbol{\theta}^*)\| > a_N\epsilon)\}$$

$$\quad + a_N^2\mathbb{E}\{e^{f(\mathbf{x};\boldsymbol{\beta}^*)}\|\dot{g}(\mathbf{x};\boldsymbol{\theta}^*)\|^2 I(\|\dot{g}(\mathbf{x};\boldsymbol{\theta}^*)\| > a_N\epsilon)\}$$

$$= o(a_N^2),$$

where the last step is from the dominated convergence theorem. Thus, applying the Lindeberg-Feller central limit theorem [Section *2.8 of 4], we finish the proof of (S.3).

Now we prove (S.4). This is done by noting that

$$a_N^{-2}\sum_{i=1}^{N}\phi(\mathbf{x}_i;\boldsymbol{\theta}^*)\dot{g}^{\otimes 2}(\mathbf{x}_i;\boldsymbol{\theta}^*)$$

$$= \frac{1}{Ne^{\alpha^*}}\sum_{i=1}^{N}\frac{e^{\alpha^*+f(\mathbf{x}_i;\boldsymbol{\beta}^*)}}{(1+e^{\alpha^*+f(\mathbf{x}_i;\boldsymbol{\beta}^*)})^2}\dot{g}^{\otimes 2}(\mathbf{x}_i;\boldsymbol{\theta}^*)$$

$$= \frac{1}{N}\sum_{i=1}^{N}\frac{e^{f(\mathbf{x}_i;\boldsymbol{\beta}^*)}}{(1+e^{\alpha^*+f(\mathbf{x}_i;\boldsymbol{\beta}^*)})^2}\dot{g}^{\otimes 2}(\mathbf{x}_i;\boldsymbol{\theta}^*) = \mathbb{E}\{e^{f(\mathbf{x};\boldsymbol{\beta}^*)}\dot{g}^{\otimes 2}(\mathbf{x};\boldsymbol{\theta}^*)\} + o_P(1),$$

where the last step is from Lemma 28 of [5].

## S.2 Proof of Theorem 2

First, note that $\mathbf{V}_w$ can be written as

$$\mathbf{V}_w = \mathbb{E}(e^{f(\mathbf{x};\boldsymbol{\beta}^*)})\mathbf{M}_{\mathrm{f}}^{-1}\mathbf{M}_w\mathbf{M}_{\mathrm{f}}^{-1},$$

where

$$\mathbf{M}_w = \mathbb{E}\left[\left\{1 + \frac{ce^{f(\mathbf{x};\boldsymbol{\beta}^*)}}{\varphi(\mathbf{x})}\right\}e^{f(\mathbf{x};\boldsymbol{\beta}^*)}\dot{g}^{\otimes 2}(\mathbf{x};\boldsymbol{\theta}^*)\right].$$

We then point out that under assumptions 1 and 2 the condition $\mathbb{E}[\{\varphi(\mathbf{x}) + \varphi^{-1}(\mathbf{x})\}B^2(\mathbf{x})] < \infty$ implies the following:

$$\mathbb{E}\left[\{\varphi(\mathbf{x})e^{f(\mathbf{x};\boldsymbol{\beta}^*)}\}\|\dot{g}(\mathbf{x};\boldsymbol{\theta}^*)\|^2\right] < \infty, \tag{S.6}$$

$$\mathbb{E}\left[\{1 + \varphi^{-1}(\mathbf{x})\}e^{2f(\mathbf{x};\boldsymbol{\beta}^*)}\|\dot{g}(\mathbf{x};\boldsymbol{\theta}^*)\|^4\right] < \infty, \tag{S.7}$$

$$\mathbb{E}\left\{\varphi^{-1}(\mathbf{x})e^{2f(\mathbf{x};\boldsymbol{\beta}^*)}\|\ddot{g}(\mathbf{x};\boldsymbol{\theta}^*)\|^2\right\} < \infty. \tag{S.8}$$

The estimator $\hat{\boldsymbol{\theta}}_w$ is the maximizer of

$$\ell_w(\boldsymbol{\theta}) = \sum_{i=1}^{N}\frac{\delta_i}{\pi(\mathbf{x}_i, y_i)}\left[y_i g(\mathbf{x}_i;\boldsymbol{\theta}) - \log\{1 + e^{g(\mathbf{x}_i;\boldsymbol{\theta})}\}\right],$$

so $a_N(\hat{\boldsymbol{\theta}}_w - \boldsymbol{\theta}^*)$ is the maximizer of $\gamma_w(\mathbf{u}) = \ell_w(\boldsymbol{\theta}^* + a_N^{-1}\mathbf{u}) - \ell_w(\boldsymbol{\theta}^*)$. By Taylor's expansion,

$$\gamma_w(\mathbf{u}) = \frac{1}{a_N}\mathbf{u}^{\mathrm{T}}\dot{\ell}_w(\boldsymbol{\theta}^*) + \frac{1}{2a_N^2}\sum_{i=1}^{N}\frac{\delta_i}{\pi(\mathbf{x}_i, y_i)}\phi(\mathbf{x}_i;\boldsymbol{\theta}^*)(\mathbf{z}_i^{\mathrm{T}}\mathbf{u})^2 + \Delta_{\ell_w} + R_w,$$

where

$$\dot{\ell}_w(\boldsymbol{\theta}) = \frac{\partial\ell_w(\boldsymbol{\theta})}{\partial\boldsymbol{\theta}} = \sum_{i=1}^{N}\frac{\delta_i}{\pi(\mathbf{x}_i, y_i)}\{y_i - p(\mathbf{x}_i;\boldsymbol{\theta})\}\dot{g}(\mathbf{x}_i, \boldsymbol{\theta}),$$

$$\Delta_{\ell_w} = \frac{1}{2a_N^2}\sum_{i=1}^{N}\frac{\delta_i}{\pi(\mathbf{x}_i, y_i)}\{y_i - p(\mathbf{x}_i;\boldsymbol{\theta}^*)\}\mathbf{u}^{\mathrm{T}}\ddot{g}(\mathbf{x}_i, \boldsymbol{\theta}^*)\mathbf{u},$$

and $R_w$ is the remainder. Similarly to the proof of Theorem 1, we only need to show that

$$a_N^{-1}\dot{\ell}_w(\boldsymbol{\theta}^*) \longrightarrow \mathbb{N}(\mathbf{0}, \ \mathbf{M}_w), \tag{S.9}$$

in distribution,

$$a_N^{-2}\sum_{i=1}^{N}\frac{\delta_i}{\pi(\mathbf{x}_i, y_i)}\phi(\mathbf{x}_i;\boldsymbol{\theta}^*)\dot{g}^{\otimes 2}(\mathbf{x}_i;\boldsymbol{\theta}^*) \longrightarrow \mathbf{M}_{\mathrm{f}}, \tag{S.10}$$

in probability for any $\mathbf{u}$, and $\Delta_{\ell_w} = o_P(1)$ and $R_w = o_P(1)$.

We prove (S.9) first. Let $\eta_i = \frac{\delta_i}{\pi(\mathbf{x}_i, y_i)}\{y_i - p(\mathbf{x}_i;\boldsymbol{\theta}^*)\}\dot{g}(\mathbf{x}_i;\boldsymbol{\theta}^*)$, we know that $\eta_i$, $i = 1, ..., N$, are i.i.d., with the underlying distribution of $\eta_i$ being dependent on $N$. From direct calculation, we have that $\mathbb{E}(\eta_i \mid \mathbf{x}_i) = \mathbf{0}$ and

$$\mathbb{V}(\eta_i \mid \mathbf{x}_i) = \mathbb{E}\left[\frac{\{y_i - p(\mathbf{x}_i;\boldsymbol{\theta}^*)\}^2}{\{y_i + (1 - y_i)\rho\varphi(\mathbf{x}_i)\}} \,\bigg|\, \mathbf{x}_i\right]\dot{g}^{\otimes 2}(\mathbf{x}_i;\boldsymbol{\theta}^*)$$

$$= p(\mathbf{x}_i;\boldsymbol{\theta}^*)\{1 - p(\mathbf{x}_i;\boldsymbol{\theta}^*)\}^2\dot{g}^{\otimes 2}(\mathbf{x}_i;\boldsymbol{\theta}^*) + \{1 - p(\mathbf{x}_i;\boldsymbol{\theta}^*)\}\frac{p^2(\mathbf{x}_i;\boldsymbol{\theta}^*)}{\rho\varphi(\mathbf{x}_i)}\dot{g}^{\otimes 2}(\mathbf{x}_i;\boldsymbol{\theta}^*)$$

$$\leq e^{\alpha^*}e^{f(\mathbf{x}_i;\boldsymbol{\beta}^*)}\dot{g}^{\otimes 2}(\mathbf{x}_i;\boldsymbol{\theta}^*) + \rho^{-1}e^{2\alpha^*}\frac{e^{2f(\mathbf{x}_i;\boldsymbol{\beta}^*)}}{\varphi(\mathbf{x}_i)}\dot{g}^{\otimes 2}(\mathbf{x}_i;\boldsymbol{\theta}^*).$$

Thus, by the dominated convergence theorem, we obtain that

$$\mathbb{V}(\eta_i) = \mathbb{E}\{\mathbb{V}(\eta_i \mid \mathbf{x}_i)\} = e^{\alpha^*}\mathbb{E}\left[\left\{1 + \frac{ce^{f(\mathbf{x}_i;\boldsymbol{\beta}^*)}}{\varphi(\mathbf{x}_i)}\right\}e^{f(\mathbf{x}_i;\boldsymbol{\beta}^*)}\dot{g}^{\otimes 2}(\mathbf{x}_i;\boldsymbol{\theta}^*)\right]\{1 + o(1)\}.$$

Now we check the Lindeberg-Feller condition [Section *2.8 of 4]. Denote $\delta = y + (1-y)I\{u \leq \rho\varphi(\mathbf{x})\}$ where $u \sim \mathbb{U}(0,1)$. For any $\epsilon > 0$,

$$\sum_{i=1}^{N}\mathbb{E}\big\{\|\eta_i\|^2 I(\|\eta_i\| > a_N\epsilon)\big\}$$

$$= N\mathbb{E}\big[\|\pi^{-1}(\mathbf{x},y)\delta\{y - p(\mathbf{x};\boldsymbol{\theta}^*)\}\dot{g}(\mathbf{x};\boldsymbol{\theta}^*)\|^2 I(\|\pi^{-1}(\mathbf{x},y)\delta\{y - p(\mathbf{x};\boldsymbol{\theta}^*)\}\dot{g}(\mathbf{x};\boldsymbol{\theta}^*)\| > a_N\epsilon)\big]$$

$$= \rho N\mathbb{E}\Big[\varphi(\mathbf{x})\|\pi^{-1}(\mathbf{x},y)\{y - p(\mathbf{x};\boldsymbol{\theta}^*)\}\dot{g}(\mathbf{x};\boldsymbol{\theta}^*)\|^2 I(\|\pi^{-1}(\mathbf{x},y)\{y - p(\mathbf{x};\boldsymbol{\theta}^*)\}\dot{g}(\mathbf{x};\boldsymbol{\theta}^*)\| > a_N\epsilon)\Big]$$

$$+ N\mathbb{E}\Big[\{1 - \rho\varphi(\mathbf{x})\}\|\pi^{-1}(\mathbf{x},y)y\{y - p(\mathbf{x};\boldsymbol{\theta}^*)\}\dot{g}(\mathbf{x};\boldsymbol{\theta}^*)\|^2$$

$$\times I(\|\pi^{-1}(\mathbf{x},y)y\{y - p(\mathbf{x};\boldsymbol{\theta}^*)\}\dot{g}(\mathbf{x};\boldsymbol{\theta}^*)\| > a_N\epsilon)\Big]$$

$$= \rho N\mathbb{E}\Big[\varphi(\mathbf{x})p(\mathbf{x};\boldsymbol{\theta}^*)\|\{1 - p(\mathbf{x};\boldsymbol{\theta}^*)\}\dot{g}(\mathbf{x};\boldsymbol{\theta}^*)\|^2 I(\|\{1 - p(\mathbf{x};\boldsymbol{\theta}^*)\}\dot{g}(\mathbf{x};\boldsymbol{\theta}^*)\| > a_N\epsilon)\Big]$$

$$+ N\mathbb{E}\Big[\{1 - p(\mathbf{x};\boldsymbol{\theta}^*)\}\rho^{-1}\varphi^{-1}(\mathbf{x})\|p(\mathbf{x};\boldsymbol{\theta}^*)\dot{g}(\mathbf{x};\boldsymbol{\theta}^*)\|^2$$

$$\times I(\|\rho^{-1}\varphi^{-1}(\mathbf{x})\{y - p(\mathbf{x};\boldsymbol{\theta}^*)\}\dot{g}(\mathbf{x};\boldsymbol{\theta}^*)\| > a_N\epsilon)\Big]$$

$$+ N\mathbb{E}\Big[\{1 - \rho\varphi(\mathbf{x})\}p(\mathbf{x};\boldsymbol{\theta}^*)\|\{1 - p(\mathbf{x};\boldsymbol{\theta}^*)\}\dot{g}(\mathbf{x};\boldsymbol{\theta}^*)\|^2 I(\|\{1 - p(\mathbf{x};\boldsymbol{\theta}^*)\}\dot{g}(\mathbf{x};\boldsymbol{\theta}^*)\| > a_N\epsilon)\Big]$$

$$\leq \rho Ne^{\alpha^*}\mathbb{E}\Big[\varphi(\mathbf{x})e^{f(\mathbf{x};\boldsymbol{\beta}^*)}\|\dot{g}(\mathbf{x};\boldsymbol{\theta}^*)\|^2 I(\|\dot{g}(\mathbf{x};\boldsymbol{\theta}^*)\| > a_N\epsilon)\Big]$$

$$+ Ne^{2\alpha^*}\rho^{-1}\mathbb{E}\Big[\varphi^{-1}(\mathbf{x})\|e^{f(\mathbf{x};\boldsymbol{\beta}^*)}\dot{g}(\mathbf{x};\boldsymbol{\theta}^*)\|^2 I(\|\rho^{-1}\varphi^{-1}(\mathbf{x})\{y - p(\mathbf{x};\boldsymbol{\theta}^*)\}\dot{g}(\mathbf{x};\boldsymbol{\theta}^*)\| > a_N\epsilon)\Big]$$

$$+ Ne^{\alpha^*}\mathbb{E}\Big[e^{f(\mathbf{x};\boldsymbol{\beta}^*)}\|\dot{g}(\mathbf{x};\boldsymbol{\theta}^*)\|^2 I(\|\dot{g}(\mathbf{x};\boldsymbol{\theta}^*)\| > a_N\epsilon)\Big]$$

$$= o(Ne^{\alpha^*}),$$

where the last step is due to Assumptions 1, (S.6)-(S.8), the dominated convergence theorem, and the facts that $a_N \to \infty$ and $\lim_{N\to\infty} e^{\alpha^*}/\rho = c < \infty$. Thus, applying the Lindeberg-Feller central limit theorem [Section *2.8 of 4] finishes the proof of (S.9).

Now we prove (S.10). Let

$$H_w \equiv a_N^{-2}\sum_{i=1}^{N}\frac{\delta_i}{\pi(\mathbf{x}_i, y_i)}\phi(\mathbf{x}_i;\boldsymbol{\theta}^*)\dot{g}^{\otimes 2}(\mathbf{x}_i;\boldsymbol{\theta}^*)$$

$$= \frac{1}{n}\sum_{i=1}^{N}\frac{\delta_i}{\pi(\mathbf{x}_i, y_i)}\frac{e^{f(\mathbf{x}_i;\boldsymbol{\beta}^*)}}{(1 + e^{\alpha^* + f(\mathbf{x}_i;\boldsymbol{\beta}^*)})^2}\dot{g}^{\otimes 2}(\mathbf{x}_i;\boldsymbol{\theta}^*)$$

$$= \frac{1}{n}\sum_{i=1}^{N}\frac{y_i + (1-y_i)I\{u_i \leq \rho\varphi(\mathbf{x}_i)\}}{y_i + (1-y_i)\rho\varphi(\mathbf{x}_i)}\frac{e^{f(\mathbf{x}_i;\boldsymbol{\beta}^*)}}{(1 + e^{\alpha^* + f(\mathbf{x}_i;\boldsymbol{\beta}^*)})^2}\dot{g}^{\otimes 2}(\mathbf{x}_i;\boldsymbol{\theta}^*).$$

We notice that

$$\mathbb{E}(H_w) = \mathbb{E}\left\{\frac{e^{f(\mathbf{x};\boldsymbol{\beta}^*)}}{(1 + e^{\alpha^* + f(\mathbf{x};\boldsymbol{\beta}^*)})^2}\dot{g}^{\otimes 2}(\mathbf{x};\boldsymbol{\theta}^*)\right\} = \mathbb{E}\{e^{f(\mathbf{x};\boldsymbol{\beta}^*)}\dot{g}^{\otimes 2}(\mathbf{x};\boldsymbol{\theta}^*)\} + o(1), \qquad \text{(S.11)}$$

where the last step is by the dominated convergence theorem. In addition, the variance of each component of $H_w$ is bounded by

$$\frac{1}{N}\mathbb{E}\left\{\frac{\delta}{\pi^2(\mathbf{x}, y)}e^{2f(\mathbf{x};\boldsymbol{\beta}^*)}\|\dot{g}(\mathbf{x};\boldsymbol{\theta}^*)\|^4\right\}$$

$$\leq \frac{1}{N}\mathbb{E}\left\{\frac{1}{\pi(\mathbf{x}, y)}e^{2f(\mathbf{x};\boldsymbol{\beta}^*)}\|\dot{g}(\mathbf{x};\boldsymbol{\theta}^*)\|^4\right\}$$

$$\leq \frac{1}{N\rho}\mathbb{E}\left[\left\{\rho + \frac{1}{\varphi(\mathbf{x})}\right\}e^{2f(\mathbf{x};\boldsymbol{\beta}^*)}\|\dot{g}(\mathbf{x};\boldsymbol{\theta}^*)\|^4\right] = o(1). \tag{S.12}$$

where the last step is because of (S.7) and the fact that $Ne^{\alpha^*} \to \infty$ and $e^{\alpha^*}/\rho \to c < \infty$ imply that $N\rho \to \infty$. From (S.11) and (S.12), Chebyshev's inequality implies that $H_w \to \mathbb{E}\{e^{f(\mathbf{x};\boldsymbol{\beta}^*)}\dot{g}^{\otimes 2}(\mathbf{x};\boldsymbol{\theta}^*)\}$ in probability.

In the following we finish the proof by showing that $\Delta_{\ell_w} = o_P(1)$ and $R_w = o_P(1)$. For $\Delta_{\ell_w}$, the mean $\mathbb{E}(\Delta_{\ell_w}) = \mathbf{0}$ and the variance satisfies that

$$
\begin{aligned}
\mathbb{V}(\Delta_{\ell_w}) &\leq \frac{\|\mathbf{u}\|^4 N}{4a_N^4}\mathbb{E}\left[\frac{\{y - p(\mathbf{x};\boldsymbol{\theta}^*)\}^2\|\ddot{g}(\mathbf{x};\boldsymbol{\theta}^*)\|^2}{y + (1-y)\rho\varphi(\mathbf{x})}\right] \\
&\leq \frac{\|\mathbf{u}\|^4 N}{4a_N^4}\mathbb{E}\left[\left\{1 + \frac{p(\mathbf{x};\boldsymbol{\theta}^*)}{\rho\varphi(\mathbf{x})}\right\}p(\mathbf{x};\boldsymbol{\theta}^*)\|\ddot{g}(\mathbf{x};\boldsymbol{\theta}^*)\|^2\right] \\
&\leq \frac{\|\mathbf{u}\|^4}{4a_N^2}\mathbb{E}\left[\left\{1 + \frac{e^{\alpha^*}e^{f(\mathbf{x};\boldsymbol{\beta}^*)}}{\rho\varphi(\mathbf{x})}\right\}e^{f(\mathbf{x};\boldsymbol{\beta}^*)}\|\ddot{g}(\mathbf{x};\boldsymbol{\theta}^*)\|^2\right] = o_P(1),
\end{aligned}
$$

so $\Delta_{\ell_w} = o_P(1)$. For the remainder term $R_w$. By direct calculation,

$$
\begin{aligned}
R_w &= \frac{1}{6a_N^3}\sum_{i=1}^N \frac{\delta_i}{\pi(\mathbf{x}_i, y_i)}\phi(\mathbf{x}_i;\boldsymbol{\theta}^* + a_N^{-1}\acute{\mathbf{u}})\{1 - 2p(\mathbf{x}_i;\boldsymbol{\theta}^* + a_N^{-1}\acute{\mathbf{u}})\}\{\mathbf{u}^{\mathrm{T}}\dot{g}(\mathbf{x}_i;\boldsymbol{\theta}^* + a_N^{-1}\acute{\mathbf{u}})\}^3 \\
&\quad + \frac{1}{6a_N^3}\sum_{i=1}^N \frac{\delta_i}{\pi(\mathbf{x}_i, y_i)}\phi(\mathbf{x}_i;\boldsymbol{\theta}^* + a_N^{-1}\acute{\mathbf{u}})\{\mathbf{u}^{\mathrm{T}}\dot{g}(\mathbf{x}_i;\boldsymbol{\theta}^* + a_N^{-1}\acute{\mathbf{u}})\}\{\mathbf{u}^{\mathrm{T}}\ddot{g}(\mathbf{x}_i;\boldsymbol{\theta}^* + a_N^{-1}\acute{\mathbf{u}})\mathbf{u}\} \\
&\quad + \frac{1}{6a_N^3}\sum_{i=1}^N \frac{\delta_i}{\pi(\mathbf{x}_i, y_i)}\left[\{y_i - p(\mathbf{x}_i;\boldsymbol{\theta}^* + a_N^{-1}\acute{\mathbf{u}})\}\sum_{j_1 j_2}^d u_{j_1}u_{j_2}\dddot{g}_{j_1 j_2}(\mathbf{x}_i;\boldsymbol{\theta}^* + a_N^{-1}\acute{\mathbf{u}})\mathbf{u}\right],
\end{aligned}
$$

where $\acute{\mathbf{u}}$ lies between $\mathbf{0}$ and $\mathbf{u}$. We see that

$$
\begin{aligned}
|R_w| &\leq \frac{\|\mathbf{u}\|^3}{6a_N^3}\sum_{i=1}^N \frac{\delta_i}{\pi(\mathbf{x}_i, y_i)}p(\mathbf{x}_i;\boldsymbol{\theta}^* + a_N^{-1}\acute{\mathbf{u}})C(\mathbf{x}_i, \boldsymbol{\theta}^* + a_N^{-1}\acute{\mathbf{u}}) + \frac{d\|\mathbf{u}\|^3}{6a_N^3}\sum_{i=1}^N \frac{\delta_i}{\pi(\mathbf{x}_i, y_i)}y_i B(\mathbf{x}_i) \\
&\leq \frac{\|\mathbf{u}\|^3 e^{a_N^{-1}\|\mathbf{u}\|}}{6Na_N}\sum_{i=1}^N \frac{\delta_i}{\pi(\mathbf{x}_i, y_i)}\exp\{f(\mathbf{x}_i;\boldsymbol{\beta} + a_N^{-1}\acute{\mathbf{u}}_{(-1)})\}C(\mathbf{x}_i, \boldsymbol{\theta}^* + a_N^{-1}\acute{\mathbf{u}}) \\
&\quad + \frac{d\|\mathbf{u}\|^3}{6a_N^3}\sum_{i=1}^N \frac{\delta_i}{\pi(\mathbf{x}_i, y_i)}y_i B(\mathbf{x}_i) \\
&\leq \frac{d^3\|\mathbf{u}\|^3 e^{a_N^{-1}\|\mathbf{u}\|}}{6Na_N}\sum_{i=1}^N \frac{\delta_i}{\pi(\mathbf{x}_i, y_i)}B(\mathbf{x}_i) + \frac{d\|\mathbf{u}\|^3}{6a_N^3}\sum_{i=1}^N \frac{\delta_i}{\pi(\mathbf{x}_i, y_i)}y_i B(\mathbf{x}_i) \\
&\equiv \Delta_{R_w 1} + \Delta_{R_w 2},
\end{aligned}
$$

where $C(\mathbf{x}, \boldsymbol{\theta})$ is defined in (S.2) and $\acute{\mathbf{u}}_{(-1)}$ is $\acute{\mathbf{u}}$ with the first element removed. From Assumption 1, $\mathbb{E}(\Delta_{R_w 1}) \to 0$ and

$$\mathbb{E}(\Delta_{R_w 2}) \leq \frac{d\|\mathbf{u}\|^3}{6a_N}\mathbb{E}\left[e^{f(\mathbf{x};\boldsymbol{\beta}^*)}B(\mathbf{x})\right] \to 0.$$

Since $\Delta_{R_w 1}$ and $\Delta_{R_w 2}$ are both positive, Markov's inequality shows that they are both $o_P(1)$ so $R_w = o_P(1)$.

## S.3  Proof of Theorem 3

Since only $\mathbf{V}_{\mathrm{sub}}$ in $\mathbf{V}_w$ is affect by subsampling, the probability that minimizes $\mathrm{tr}(\mathbf{V}_{\mathrm{sub}})$ also minimizes $\mathrm{tr}(\mathbf{V}_w)$. Let $\Lambda_{\mathrm{sub}} = \mathbb{E}\{\varphi^{-1}(\mathbf{x})e^{2f(\mathbf{x};\boldsymbol{\beta}^*)}\dot{g}^{\otimes 2}(\mathbf{x};\boldsymbol{\theta}^*)\}$. We notice that

$$\mathbf{V}_{\mathrm{sub}} = c\mathbb{E}(e^{f(\mathbf{x};\boldsymbol{\beta}^*)})\mathbf{M}_{\mathrm{f}}^{-1}\Lambda_{\mathrm{sub}}\mathbf{M}_{\mathrm{f}}^{-1}$$

$$= c\mathbb{E}(e^{f(\mathbf{x};\boldsymbol{\beta}^*)})e^{-2\alpha^*}\{1+o_P(1)\}\mathbf{M}_{\mathrm{f}}^{-1}\mathbb{E}\{\varphi^{-1}(\mathbf{x})p^2(\mathbf{x};\boldsymbol{\theta}^*)\dot{g}^{\otimes 2}(\mathbf{x};\boldsymbol{\theta}^*)\}\mathbf{M}_{\mathrm{f}}^{-1},$$

where the term $c\mathbb{E}(e^{f(\mathbf{x};\boldsymbol{\beta}^*)})e^{-2\alpha^*}$ does not depend on $\varphi(\mathbf{x})$. Thus we can focus on minimizing the term $\mathbf{M}_{\mathrm{f}}^{-1}\mathbb{E}\{\varphi^{-1}(\mathbf{x})p^2(\mathbf{x};\boldsymbol{\theta}^*)\dot{g}^{\otimes 2}(\mathbf{x};\boldsymbol{\theta}^*)\}\mathbf{M}_{\mathrm{f}}^{-1}$.

Note that

$$\mathrm{tr}[p^2(\mathbf{x};\boldsymbol{\theta}^*)\mathbf{M}_{\mathrm{f}}^{-1}\dot{g}^{\otimes 2}(\mathbf{x};\boldsymbol{\theta}^*)\mathbf{M}_{\mathrm{f}}^{-1}] = \|p(\mathbf{x};\boldsymbol{\theta})\mathbf{M}_{\mathrm{f}}^{-1}\dot{g}(\mathbf{x};\boldsymbol{\theta})\|^2 = t^2(\mathbf{x};\boldsymbol{\theta}^*).$$

Thus the problem is to find $\varphi(\mathbf{x})$ that minimizes

$$\mathbb{E}\left\{\frac{t^2(\mathbf{x};\boldsymbol{\theta}^*)}{\varphi(\mathbf{x})}\right\}, \quad \text{subject to} \quad 0 < \varphi(\mathbf{x}) \le \rho^{-1} \text{ and } \mathbb{E}\{\varphi(\mathbf{x})\} = 1.$$

To facilitate the presentation, denote

$$\zeta = \frac{1}{\mathbb{E}[\min\{t(\mathbf{x};\boldsymbol{\theta}^*), T\}]},$$

which is non-random. We notice that

$$\mathbb{E}\left\{\frac{\zeta^2 t^2(\mathbf{x};\boldsymbol{\theta}^*)}{\varphi(\mathbf{x})}\right\}$$
$$= \mathbb{E}\left(\frac{\zeta^2[\min\{t(\mathbf{x};\boldsymbol{\theta}^*), T\}]^2}{\varphi(\mathbf{x})}\right) + \mathbb{E}\left[\frac{\zeta^2\{t^2(\mathbf{x};\boldsymbol{\theta}^*) - T^2\}}{\varphi(\mathbf{x})}I\{t(\mathbf{x};\boldsymbol{\theta}^*) > T\}\right]$$
$$= \mathbb{E}\left(\left[\sqrt{\varphi(\mathbf{x})} - \frac{\zeta\min\{t(\mathbf{x};\boldsymbol{\theta}^*), T\}}{\sqrt{\varphi(\mathbf{x})}}\right]^2\right) - \mathbb{E}\{\varphi(\mathbf{x})\} + 2\mathbb{E}\left[\zeta\min\{t(\mathbf{x};\boldsymbol{\theta}^*), T\}\right]$$
$$+ \mathbb{E}\left[\frac{\zeta^2\{t^2(\mathbf{x};\boldsymbol{\theta}^*) - T^2\}}{\varphi(\mathbf{x})}I\{t(\mathbf{x};\boldsymbol{\theta}^*) > T\}\right]$$
$$= \mathbb{E}\left(\left[\sqrt{\varphi(\mathbf{x})} - \frac{\zeta\min\{t(\mathbf{x};\boldsymbol{\theta}^*), T\}}{\sqrt{\varphi(\mathbf{x})}}\right]^2\right) + \mathbb{E}\left[\frac{\zeta^2\{t^2(\mathbf{x};\boldsymbol{\theta}^*) - T^2\}}{\varphi(\mathbf{x})}I\{t(\mathbf{x};\boldsymbol{\theta}^*) > T\}\right] + 1$$
$$\equiv E_1 + E_2 + 1,$$

where $I(\cdot)$ is the indicator function. Note that $E_i$'s are non-negative and $E_1 = 0$ if and only if

$$\varphi(\mathbf{x}) = \zeta\min\{t(\mathbf{x};\boldsymbol{\theta}^*), T\} = \frac{\min\{t(\mathbf{x};\boldsymbol{\theta}^*), T\}}{\mathbb{E}[\min\{t(\mathbf{x};\boldsymbol{\theta}^*), T\}]},$$

which is attainable because

$$\rho\min\{t(\mathbf{x};\boldsymbol{\theta}^*), T\} \le \mathbb{E}[\min\{t(\mathbf{x};\boldsymbol{\theta}^*), T\}], \quad \text{almost surely.} \tag{S.13}$$

If $t(\mathbf{x};\boldsymbol{\theta}^*) \le T$ almost surely, then $E_2 = 0$ and the proof finishes. If $\mathrm{Pr}\{t(\mathbf{x};\boldsymbol{\theta}^*) > T\} > 0$ then

$$\frac{\rho T}{\mathbb{E}[\min\{t(\mathbf{x};\boldsymbol{\theta}^*), T\}]} = 1,$$

because if this is not true then we can find a larger $T$ that satisfies (S.13). This means that if $t(\mathbf{x};\boldsymbol{\theta}^*) > T$, then

$$\varphi(\mathbf{x}) = \frac{\min\{t(\mathbf{x};\boldsymbol{\theta}^*), T\}}{\mathbb{E}[\min\{t(\mathbf{x};\boldsymbol{\theta}^*), T\}]} = \rho^{-1},$$

which minimizes $E_2$ as well since $\varphi(\mathbf{x})$ is in the denominator. This finishes the proof.

## S.4  Derivation of corrected model (4)

Note that $\pi(\mathbf{x}, 1) = 1$ and $\pi(\mathbf{x}, 0) = \pi(\mathbf{x})$. By direct calculation, we have

$$\mathrm{Pr}(y = 1 \mid \mathbf{x}, \delta = 1)$$
$$= \frac{\mathrm{Pr}(y = 1, \delta = 1 \mid \mathbf{x})}{\mathrm{Pr}(\delta = 1 \mid \mathbf{x})}$$

$$
= \frac{\Pr(y = 1, \delta = 1 \mid \mathbf{x})}{\Pr(y = 1, \delta = 1 \mid \mathbf{x}) + \Pr(y = 0, \delta = 1 \mid \mathbf{x})}
$$

$$
= \frac{\Pr(y = 1 \mid \mathbf{x})\Pr(\delta = 1 \mid \mathbf{x}, y = 1)}{\Pr(y = 1 \mid \mathbf{x})\Pr(\delta = 1 \mid \mathbf{x}, y = 1) + \Pr(y = 0 \mid \mathbf{x})\Pr(\delta = 1 \mid \mathbf{x}, y = 0)}
$$

$$
= \frac{p(\mathbf{x}; \boldsymbol{\theta}^*)\pi(\mathbf{x}, 1)}{p(\mathbf{x}; \boldsymbol{\theta}^*)\pi(\mathbf{x}, 1) + \{1 - p(\mathbf{x}; \boldsymbol{\theta}^*)\}\pi(\mathbf{x}, 0)}
$$

$$
= \frac{e^{g(\mathbf{x}; \boldsymbol{\theta}^*)}}{e^{g(\mathbf{x}; \boldsymbol{\theta}^*)} + \pi(\mathbf{x})}
$$

$$
= \frac{e^{g(\mathbf{x}; \boldsymbol{\theta}^*) + l}}{1 + e^{g(\mathbf{x}; \boldsymbol{\theta}^*) + l}}.
$$

## S.5   Proof of Theorem 4

The estimator $\hat{\boldsymbol{\theta}}_{\mathrm{lik}}$ is the maximizer of $\ell_{\mathrm{lik}}(\boldsymbol{\theta})$, so $\mathbf{u}_N = a_N(\hat{\boldsymbol{\theta}}_{\mathrm{lik}} - \boldsymbol{\theta})$ is the maximizer of

$$
\gamma_{\mathrm{lik}}(\mathbf{u}) = \ell_{\mathrm{lik}}(\boldsymbol{\theta} + a_N^{-1}\mathbf{u}) - \ell_{\mathrm{lik}}(\boldsymbol{\theta}).
$$

By Taylor's expansion,

$$
\gamma_{\mathrm{lik}}(\mathbf{u}) = a_N^{-1}\mathbf{u}^{\mathrm{T}}\dot{\ell}_{\mathrm{lik}}(\boldsymbol{\theta}^*) + 0.5a_N^{-2}\sum_{i=1}^{N}\phi_\pi(\mathbf{x}_i; \boldsymbol{\theta}^*)\{\mathbf{u}^{\mathrm{T}}\dot{g}(\mathbf{x}_i; \boldsymbol{\theta}^*)\}^2 + \Delta_{\ell_{\mathrm{lik}}} + R_{\mathrm{lik}},
$$

where $\phi_\pi(\mathbf{x}; \boldsymbol{\theta}) = p_\pi(\mathbf{x}; \boldsymbol{\theta})\{1 - p_\pi(\mathbf{x}; \boldsymbol{\theta})\}$,

$$
p_\pi(\mathbf{x}; \boldsymbol{\theta}) = \frac{e^{g(\mathbf{x}; \boldsymbol{\theta}) + l}}{1 + e^{g(\mathbf{x}; \boldsymbol{\theta}) + l}} \quad \text{with} \quad l = -\log\{\rho\varphi(\mathbf{x})\}, \tag{S.14}
$$

$$
\dot{\ell}_{\mathrm{lik}}(\boldsymbol{\theta}) = \frac{\partial \ell_{\mathrm{lik}}(\boldsymbol{\theta})}{\partial \boldsymbol{\theta}} = \sum_{i=1}^{N}\delta_i\{y_i - p_\pi(\mathbf{x}_i; \boldsymbol{\theta})\}\dot{g}(\mathbf{x}_i; \boldsymbol{\theta}),
$$

$$
\Delta_{\ell_{\mathrm{lik}}} = \frac{1}{2a_N^2}\sum_{i=1}^{N}\delta_i\{y_i - p_\pi(\mathbf{x}_i; \boldsymbol{\theta}^*)\}\mathbf{u}^{\mathrm{T}}\ddot{g}(\mathbf{x}_i; \boldsymbol{\theta}^*)\mathbf{u},
$$

and $R_{\mathrm{lik}}$ is the remainder term. By direct calculation,

$$
R_{\mathrm{lik}} = \frac{1}{6a_N^3}\sum_{i=1}^{N}\delta_i\phi_\pi(\mathbf{x}_i; \boldsymbol{\theta}^* + a_N^{-1}\acute{\mathbf{u}})\{1 - 2p_\pi(\mathbf{x}_i; \boldsymbol{\theta}^* + a_N^{-1}\acute{\mathbf{u}})\}\{\mathbf{u}^{\mathrm{T}}\dot{g}(\mathbf{x}_i; \boldsymbol{\theta}^* + a_N^{-1}\acute{\mathbf{u}})\}^3
$$

$$
+ \frac{1}{6a_N^3}\sum_{i=1}^{N}\delta_i\phi_\pi(\mathbf{x}_i; \boldsymbol{\theta}^* + a_N^{-1}\acute{\mathbf{u}})\{\mathbf{u}^{\mathrm{T}}\dot{g}(\mathbf{x}_i; \boldsymbol{\theta}^* + a_N^{-1}\acute{\mathbf{u}})\}\{\mathbf{u}^{\mathrm{T}}\ddot{g}(\mathbf{x}_i; \boldsymbol{\theta}^* + a_N^{-1}\acute{\mathbf{u}})\mathbf{u}\}
$$

$$
+ \frac{1}{6a_N^3}\sum_{i=1}^{N}\delta_i\left[\{y_i - p_\pi(\mathbf{x}_i; \boldsymbol{\theta}^* + a_N^{-1}\acute{\mathbf{u}})\}\sum_{j_1, j_2=1}^{d}u_{j_1}u_{j_2}\ddot{g}_{j_1 j_2}(\mathbf{x}_i; \boldsymbol{\theta}^* + a_N^{-1}\acute{\mathbf{u}})\mathbf{u}\right],
$$

where $\acute{\mathbf{u}}$ lies between $\mathbf{0}$ and $\mathbf{u}$. We see that

$$
|R_{\mathrm{lik}}| \leq \frac{\|\mathbf{u}\|^3}{6a_N^3}\sum_{i=1}^{N}\delta_i p_\pi(\mathbf{x}_i; \boldsymbol{\theta}^* + a_N^{-1}\acute{\mathbf{u}})C(\mathbf{x}_i, \boldsymbol{\theta}^* + a_N^{-1}\acute{\mathbf{u}}) + \frac{d\|\mathbf{u}\|^3}{6a_N^3}\sum_{i=1}^{N}\delta_i y_i B(\mathbf{x}_i)
$$

$$
\leq \frac{\|\mathbf{u}\|^3 e^{a_N^{-1}\|\mathbf{u}\|}}{6Na_N}\sum_{i=1}^{N}\delta_i \exp[f(\mathbf{x}_i; \boldsymbol{\beta} + a_N^{-1}\acute{\mathbf{u}}_{(-1)}) - \log\{\rho\varphi(\mathbf{x}_i)\}]C(\mathbf{x}_i, \boldsymbol{\theta}^* + a_N^{-1}\acute{\mathbf{u}})
$$

$$
+ \frac{d\|\mathbf{u}\|^3}{6a_N^3}\sum_{i=1}^{N}\delta_i y_i B(\mathbf{x}_i)
$$

$$
= \frac{\|\mathbf{u}\|^3 e^{a_N^{-1}\|\mathbf{u}\|}}{6Na_N\rho}\sum_{i=1}^{N}\delta_i\varphi^{-1}(\mathbf{x}_i)\exp[f(\mathbf{x}_i; \boldsymbol{\beta} + a_N^{-1}\acute{\mathbf{u}}_{(-1)})]C(\mathbf{x}_i, \boldsymbol{\theta}^* + a_N^{-1}\acute{\mathbf{u}})
$$

$$+ \frac{d\|\mathbf{u}\|^3}{6a_N^3} \sum_{i=1}^{N} \delta_i y_i B(\mathbf{x}_i)$$

$$\leq \frac{d^3\|\mathbf{u}\|^3 e^{a_N^{-1}\|\mathbf{u}\|}}{6Na_N\rho} \sum_{i=1}^{N} \delta_i \varphi^{-1}(\mathbf{x}_i) B(\mathbf{x}_i) + \frac{d\|\mathbf{u}\|^3}{6a_N^3} \sum_{i=1}^{N} y_i B(\mathbf{x}_i)$$

$$\equiv \Delta_{R_{\text{lik}}1} + \Delta_{R2} = \Delta_{R_{\text{lik}}1} + o_P(1),$$

where $\acute{\mathbf{u}}_{(-1)}$ is $\acute{\mathbf{u}}$ with the first element removed, $C(\mathbf{x}, \boldsymbol{\theta})$ and $\Delta_{R2} = o_P(1)$ are defined in (S.2). For $\Delta_{R_{\text{lik}}1}$, we see that

$$\mathbb{E}\Big\{ (N\rho)^{-1} \sum_{i=1}^{N} \delta_i \varphi^{-1}(\mathbf{x}_i) B(\mathbf{x}_i) \Big\} = \rho^{-1}\mathbb{E}\Big( [\{p(\mathbf{x}_i; \boldsymbol{\theta}^*)\varphi^{-1}(\mathbf{x}) + \rho\{1 - p(\mathbf{x}_i; \boldsymbol{\theta}^*)\}]B(\mathbf{x}) \Big)$$

$$\leq e^{\alpha^*} \rho^{-1} \mathbb{E}\{e^{f(\mathbf{x}; \boldsymbol{\beta}^*)}\varphi^{-1}(\mathbf{x})B(\mathbf{x})\} + \mathbb{E}\{B(\mathbf{x})\}.$$

Therefore, $\mathbb{E}(\Delta_{R_{\text{lik}}1}) \to 0$. Since $\Delta_{R_{\text{lik}}1}$ is positive, Markov's inequality shows that $\Delta_{R_{\text{lik}}1} = o_P(1)$ and thus $R_{\text{lik}} = o_P(1)$.

For $\Delta_{\ell_{\text{lik}}}$, the mean $\mathbb{E}(\Delta_{\ell_{\text{lik}}}) = \mathbf{0}$ and the variance satisfies that

$$\mathbb{V}(\Delta_{\ell_{\text{lik}}}) \leq \frac{\|\mathbf{u}\|^4}{4a_N^4} \sum_{i=1}^{N} \mathbb{E}\{\delta_i p_\pi(\mathbf{x}_i; \boldsymbol{\theta}^*)\|\ddot{g}(\mathbf{x}_i; \boldsymbol{\theta}^*)\|^2\} \leq \frac{\|\mathbf{u}\|^4}{4a_N^2} \mathbb{E}[e^{f(\mathbf{x}; \boldsymbol{\beta}^*)}\|\ddot{g}(\mathbf{x}, \boldsymbol{\theta}^*)\|^2] \to 0,$$

where the last step is because Assumption 1 implies that $e^{f(\mathbf{x}; \boldsymbol{\beta}^*)}\|\ddot{g}(\mathbf{x}, \boldsymbol{\theta}^*)\|^2$ is integrable, so $\Delta_{\ell_{\text{lik}}} = o_P(1)$.

If we can show that

$$a_N^{-1}\dot{\ell}_{\text{lik}}(\boldsymbol{\theta}^*) \longrightarrow \mathbb{N}(\mathbf{0}, \boldsymbol{\Lambda}_{\text{lik}}), \tag{S.15}$$

in distribution, and

$$a_N^{-2} \sum_{i=1}^{N} \delta_i \phi_\pi(\mathbf{x}_i; \boldsymbol{\theta}^*)\dot{g}^{\otimes 2}(\mathbf{x}_i; \boldsymbol{\theta}^*) \longrightarrow \boldsymbol{\Lambda}_{\text{lik}}, \tag{S.16}$$

in probability, then from the Basic Corollary in page 2 of [3], we know that $a_N(\hat{\boldsymbol{\theta}}_{\text{lik}} - \boldsymbol{\theta}^*)$, the maximizer of $\gamma_{\text{lik}}(\mathbf{u})$, satisfies that

$$a_N(\hat{\boldsymbol{\theta}}_{\text{lik}} - \boldsymbol{\theta}^*) = \boldsymbol{\Lambda}_{\text{lik}}^{-1} \times a_N^{-1}\dot{\ell}_{\text{lik}}(\boldsymbol{\theta}^*) + o_P(1). \tag{S.17}$$

Slutsky's theorem together with (S.15) and (S.17) implies the result in Theorem 1. We prove (S.15) and (S.16) in the following.

Note that $\dot{\ell}_{\text{lik}}(\boldsymbol{\theta}^*)$ is a summation of i.i.d. quantities, $\delta_i\{y_i - p_\pi(\mathbf{x}_i; \boldsymbol{\theta}^*)\}\dot{g}(\mathbf{x}_i; \boldsymbol{\theta}^*)$'s, whose distribution depends on $N$. Note that

$$\mathbb{E}[\delta\{y - p_\pi(\mathbf{x}; \boldsymbol{\theta}^*)\}\dot{g}(\mathbf{x}; \boldsymbol{\theta}^*)]$$
$$= \mathbb{E}[\pi(\mathbf{x}, y)\{y - p_\pi(\mathbf{x}; \boldsymbol{\theta}^*)\}\dot{g}(\mathbf{x}; \boldsymbol{\theta}^*)]$$
$$= \mathbb{E}\Big( [p(\mathbf{x}; \boldsymbol{\theta}^*)\{1 - p_\pi(\mathbf{x}; \boldsymbol{\theta}^*)\} - \{1 - p(\mathbf{x}; \boldsymbol{\theta}^*)\}\rho\varphi(\mathbf{x})p_\pi(\mathbf{x}; \boldsymbol{\theta}^*)]\dot{g}(\mathbf{x}; \boldsymbol{\theta}^*) \Big) = \mathbf{0}, \tag{S.18}$$

where the last step is by a direct calculation from (S.14) to obtain the following equality

$$p(\mathbf{x}; \boldsymbol{\theta}^*)\{1 - p_\pi(\mathbf{x}; \boldsymbol{\theta}^*)\} = \{1 - p(\mathbf{x}; \boldsymbol{\theta}^*)\}\rho\varphi(\mathbf{x})p_\pi(\mathbf{x}; \boldsymbol{\theta}^*). \tag{S.19}$$

Thus we know that $\mathbb{E}\{a_N^{-1}\dot{\ell}_{\text{lik}}(\boldsymbol{\theta}^*)\} = \mathbf{0}$. The variance of $a_N^{-1}\dot{\ell}_{\text{lik}}(\boldsymbol{\theta}^*)$ satisfies that

$$\mathbb{V}\{a_N^{-1}\dot{\ell}_{\text{lik}}(\boldsymbol{\theta}^*)\}$$
$$= a_N^{-2}N\mathbb{E}[\delta\{y - p_\pi(\mathbf{x}; \boldsymbol{\theta}^*)\}^2\dot{g}^{\otimes 2}(\mathbf{x}; \boldsymbol{\theta}^*)]$$
$$= e^{-\alpha^*}\mathbb{E}[\{y + (1 - y)\rho\varphi(\mathbf{x})\}\{y - p_\pi(\mathbf{x}; \boldsymbol{\theta}^*)\}^2\dot{g}^{\otimes 2}(\mathbf{x}; \boldsymbol{\theta}^*)]$$
$$= e^{-\alpha^*}\mathbb{E}\Big( [p(\mathbf{x}; \boldsymbol{\theta}^*)\{1 - p_\pi(\mathbf{x}; \boldsymbol{\theta}^*)\}^2 + \{1 - p(\mathbf{x}; \boldsymbol{\theta}^*)\}\rho\varphi(\mathbf{x})p_\pi^2(\mathbf{x}; \boldsymbol{\theta}^*)]\dot{g}^{\otimes 2}(\mathbf{x}; \boldsymbol{\theta}^*) \Big) \tag{S.20}$$

$$= e^{-\alpha^*}\mathbb{E}\Big(\big[p(\mathbf{x};\boldsymbol{\theta}^*)\{1 - p_\pi(\mathbf{x};\boldsymbol{\theta}^*)\}^2 + p(\mathbf{x};\boldsymbol{\theta}^*)\{1 - p_\pi(\mathbf{x};\boldsymbol{\theta}^*)\}p_\pi(\mathbf{x};\boldsymbol{\theta}^*)\big]\dot{g}^{\otimes 2}(\mathbf{x};\boldsymbol{\theta}^*)\Big) \quad \text{(S.21)}$$

$$= e^{-\alpha^*}\mathbb{E}\Big(\big[\{1 - p_\pi(\mathbf{x};\boldsymbol{\theta}^*)\}^2 + \{1 - p_\pi(\mathbf{x};\boldsymbol{\theta}^*)\}p_\pi(\mathbf{x};\boldsymbol{\theta}^*)\big]p(\mathbf{x};\boldsymbol{\theta}^*)\dot{g}^{\otimes 2}(\mathbf{x};\boldsymbol{\theta}^*)\Big) \quad \text{(S.22)}$$

$$= e^{-\alpha^*}\mathbb{E}\Big(\big[\{1 - p_\pi(\mathbf{x};\boldsymbol{\theta}^*)\}^2 + \phi_\pi(\mathbf{x};\boldsymbol{\theta}^*)\big]p(\mathbf{x};\boldsymbol{\theta}^*)\dot{g}^{\otimes 2}(\mathbf{x};\boldsymbol{\theta}^*)\Big) \quad \text{(S.23)}$$

$$= \mathbb{E}\Big(\big[\{1 - p_\pi(\mathbf{x};\boldsymbol{\theta}^*)\}^2 + \phi_\pi(\mathbf{x};\boldsymbol{\theta}^*)\big]\{1 - p(\mathbf{x};\boldsymbol{\theta}^*)\}e^{f(\mathbf{x};\boldsymbol{\beta}^*)}\dot{g}^{\otimes 2}(\mathbf{x};\boldsymbol{\theta}^*)\Big) \quad \text{(S.24)}$$

where the forth equality uses (S.19). Note that elements of the term in the expectation of (S.24) are all bounded by integrable random variable $e^{f(\mathbf{x};\boldsymbol{\beta}^*)}\|\dot{g}(\mathbf{x};\boldsymbol{\theta}^*)\|^2$. Thus, from the dominated convergence theorem and the fact that

$$\big[\{1 - p_\pi(\mathbf{x};\boldsymbol{\theta}^*)\}^2 + \phi_\pi(\mathbf{x};\boldsymbol{\theta}^*)\big]\{1 - p(\mathbf{x};\boldsymbol{\theta}^*)\}$$
$$= 1 - p_\pi(\mathbf{x};\boldsymbol{\theta}^*) + o_P(1) = \frac{1}{1 + c\varphi^{-1}(\mathbf{x})e^{f(\mathbf{x};\boldsymbol{\beta}^*)}} + o_P(1),$$

we have

$$\mathbb{V}\{a_N^{-1}\dot{\ell}_{\mathrm{lik}}(\boldsymbol{\theta}^*)\} \longrightarrow \mathbb{E}\left[\frac{e^{f(\mathbf{x};\boldsymbol{\beta}^*)}\dot{g}^{\otimes 2}(\mathbf{x};\boldsymbol{\theta}^*)}{1 + c\varphi^{-1}(\mathbf{x})e^{f(\mathbf{x};\boldsymbol{\beta}^*)}}\right] = \boldsymbol{\Lambda}_{\mathrm{lik}}. \quad \text{(S.25)}$$

Now we check the Lindeberg-Feller condition. For any $\epsilon > 0$,

$$\sum_{i=1}^{N}\mathbb{E}\Big[\|\delta_i\{y_i - p_\pi(\mathbf{x}_i;\boldsymbol{\theta}^*)\}\dot{g}(\mathbf{x}_i;\boldsymbol{\theta}^*)\|^2 I(\|\delta_i\{y_i - p_\pi(\mathbf{x}_i;\boldsymbol{\theta}^*)\}\dot{g}(\mathbf{x}_i;\boldsymbol{\theta}^*)\| > a_N\epsilon)\Big]$$

$$= N\mathbb{E}\Big[\delta\|\{y - p_\pi(\mathbf{x};\boldsymbol{\theta}^*)\}\dot{g}(\mathbf{x};\boldsymbol{\theta}^*)\|^2 I(\delta\|\{y - p_\pi(\mathbf{x};\boldsymbol{\theta}^*)\}\dot{g}(\mathbf{x};\boldsymbol{\theta}^*)\| > a_N\epsilon)\Big]$$

$$= N\mathbb{E}\big[p(\mathbf{x};\boldsymbol{\theta}^*)\{1 - p_\pi(\mathbf{x};\boldsymbol{\theta}^*)\}^2\|\dot{g}(\mathbf{x};\boldsymbol{\theta}^*)\|^2 I(\{1 - p_\pi(\mathbf{x};\boldsymbol{\theta}^*)\}\|\dot{g}(\mathbf{x};\boldsymbol{\theta}^*)\| > a_N\epsilon)\big]$$
$$\quad + N\mathbb{E}\big[\{1 - p(\mathbf{x};\boldsymbol{\theta}^*)\}\rho\varphi(\mathbf{x})\{p_\pi(\mathbf{x};\boldsymbol{\theta}^*)\}^2\|\dot{g}(\mathbf{x};\boldsymbol{\theta}^*)\|^2 I(\rho\varphi(\mathbf{x})p_\pi(\mathbf{x};\boldsymbol{\theta}^*)\|\dot{g}(\mathbf{x};\boldsymbol{\theta}^*)\| > a_N\epsilon)\big]$$

$$\leq N\mathbb{E}\big[e^{\alpha^* + f(\mathbf{x};\boldsymbol{\beta}^*)}\|\dot{g}(\mathbf{x};\boldsymbol{\theta}^*)\|^2 I(\|\dot{g}(\mathbf{x};\boldsymbol{\theta}^*)\| > a_N\epsilon)\big]$$
$$\quad + N\mathbb{E}\big[e^{\alpha^* + f(\mathbf{x};\boldsymbol{\beta}^*)}p_\pi(\mathbf{x};\boldsymbol{\theta}^*)\|\dot{g}(\mathbf{x};\boldsymbol{\theta}^*)\|^2 I(e^{\alpha^* + f(\mathbf{x};\boldsymbol{\beta}^*)}\|\dot{g}(\mathbf{x};\boldsymbol{\theta}^*)\| > a_N\epsilon)\big]$$

$$\leq a_N^2\mathbb{E}\big[e^{f(\mathbf{x};\boldsymbol{\beta}^*)}\|\dot{g}(\mathbf{x};\boldsymbol{\theta}^*)\|^2 I(\|\dot{g}(\mathbf{x};\boldsymbol{\theta}^*)\| > a_N\epsilon)\big]$$
$$\quad + a_N^2\mathbb{E}\big[e^{f(\mathbf{x};\boldsymbol{\beta}^*)}\|\dot{g}(\mathbf{x};\boldsymbol{\theta}^*)\|^2 I(e^{\alpha^* + f(\mathbf{x};\boldsymbol{\beta}^*)}\|\dot{g}(\mathbf{x};\boldsymbol{\theta}^*)\| > a_N\epsilon)\big]$$

$$= o(a_N^2),$$

where the inequality in the third step uses the following fact derived from (S.14)

$$\rho\varphi(\mathbf{x})p_\pi(\mathbf{x};\boldsymbol{\theta}^*) = e^{\alpha^* + f(\mathbf{x};\boldsymbol{\beta}^*)}\{1 - p_\pi(\mathbf{x};\boldsymbol{\theta}^*)\} \leq e^{\alpha^* + f(\mathbf{x};\boldsymbol{\beta}^*)}, \quad \text{(S.26)}$$

and the last step is from the dominated convergence theorem. Thus, applying the Lindeberg-Feller central limit theorem [Section *2.8 of 4], we finish the proof of (S.15).

Now we prove (S.16). Denote

$$H_{\mathrm{lik}} = a_N^{-2}\sum_{i=1}^{N}\delta_i\phi_\pi(\mathbf{x}_i;\boldsymbol{\theta}^*)\dot{g}^{\otimes 2}(\mathbf{x}_i;\boldsymbol{\theta}^*).$$

We notice that

$$\mathbb{E}(H_{\mathrm{lik}}) = Na_N^{-2}\mathbb{E}[\{y + (1 - y)\rho\varphi(\mathbf{x})\}\phi_\pi(\mathbf{x};\boldsymbol{\theta}^*)\dot{g}^{\otimes 2}(\mathbf{x};\boldsymbol{\theta}^*)]$$
$$= e^{-\alpha^*}\mathbb{E}[\{1 - p(\mathbf{x};\boldsymbol{\theta}^*)\}\rho\varphi(\mathbf{x})p_\pi(\mathbf{x};\boldsymbol{\theta}^*)\{1 - p_\pi(\mathbf{x};\boldsymbol{\theta}^*)\}\dot{g}^{\otimes 2}(\mathbf{x};\boldsymbol{\theta}^*)]$$
$$\quad + e^{-\alpha^*}\mathbb{E}[p(\mathbf{x};\boldsymbol{\theta}^*)p_\pi(\mathbf{x};\boldsymbol{\theta}^*)\{1 - p_\pi(\mathbf{x};\boldsymbol{\theta}^*)\}\dot{g}^{\otimes 2}(\mathbf{x};\boldsymbol{\theta}^*)]$$
$$= e^{-\alpha^*}\mathbb{E}[p(\mathbf{x};\boldsymbol{\theta}^*)\{1 - p_\pi(\mathbf{x};\boldsymbol{\theta}^*)\}^2\dot{g}^{\otimes 2}(\mathbf{x};\boldsymbol{\theta}^*)]$$
$$\quad + e^{-\alpha^*}\mathbb{E}[\{1 - p(\mathbf{x};\boldsymbol{\theta}^*)\}\rho\varphi(\mathbf{x})p_\pi^2(\mathbf{x};\boldsymbol{\theta}^*)\dot{g}^{\otimes 2}(\mathbf{x};\boldsymbol{\theta}^*)]$$
$$= \mathbb{V}\{a_N^{-1}\dot{\ell}_{\mathrm{lik}}(\boldsymbol{\theta}^*)\} = \boldsymbol{\Lambda}_{\mathrm{lik}} + o(1), \quad \text{(S.27)}$$

where the third equality is obtained by applying (S.19) and the forth equality is from (S.20). In addition, the variance of each component of $H_{\text{lik}}$ is bounded by

$$a_N^{-4} N \mathbb{E}\{\delta\phi_\pi^2(\mathbf{x}_i;\boldsymbol{\theta}^*)\|\dot{g}(\mathbf{x}_i;\boldsymbol{\theta}^*)\|^4\}$$
$$\leq a_N^{-4} N \mathbb{E}[\{y+(1-y)\rho\varphi(\mathbf{x})\}p_\pi(\mathbf{x}_i;\boldsymbol{\theta}^*)\|\dot{g}(\mathbf{x}_i;\boldsymbol{\theta}^*)\|^4]$$
$$\leq a_N^{-4} N \mathbb{E}[\{p(\mathbf{x};\boldsymbol{\theta}^*)+\rho\varphi(\mathbf{x})\}p_\pi(\mathbf{x}_i;\boldsymbol{\theta}^*)\|\dot{g}(\mathbf{x}_i;\boldsymbol{\theta}^*)\|^4]$$
$$\leq 2a_N^{-4} N e^{\alpha^*} \mathbb{E}\{e^{f(\mathbf{x};\boldsymbol{\beta}^*)}\|\dot{g}(\mathbf{x}_i;\boldsymbol{\theta}^*)\|^4\} = o(1), \tag{S.28}$$

where the last inequality is because of (S.26) and the last step if because Assumption 1 implies that $e^{f(\mathbf{x};\boldsymbol{\beta}^*)}\|\dot{g}(\mathbf{x}_i;\boldsymbol{\theta}^*)\|^4$ is integrable. From (S.27) and (S.28), Chebyshev's inequality implies that $H_{\text{lik}} \to \boldsymbol{\Lambda}_{\text{lik}}$ in probability.

## S.6  Proof of Theorem 5

Let $\kappa(\mathbf{x}) = 1 + c\varphi^{-1}(\mathbf{x})e^{f(\mathbf{x};\boldsymbol{\beta}^*)}$. We first notice that

$$[\mathbb{E}\{e^{f(\mathbf{x};\boldsymbol{\beta}^*)}\}]^{-1}\mathbf{V}_w = \mathbf{M}_{\text{f}}^{-1}\mathbf{M}_{\text{f}}\mathbf{M}_{\text{f}}^{-1} + \mathbf{M}_{\text{f}}^{-1}c\boldsymbol{\Lambda}_{\text{sub}}\mathbf{M}_{\text{f}}^{-1}$$
$$= \mathbf{M}_{\text{f}}^{-1}\mathbb{E}[\kappa(\mathbf{x})e^{f(\mathbf{x};\boldsymbol{\beta}^*)}\dot{g}^{\otimes 2}(\mathbf{x};\boldsymbol{\theta}^*)]\mathbf{M}_{\text{f}}^{-1},$$

and

$$[\mathbb{E}\{e^{f(\mathbf{x};\boldsymbol{\beta}^*)}\}]^{-1}\mathbf{V}_{\text{lik}} = \boldsymbol{\Lambda}_{\text{lik}}^{-1} = \left[\mathbb{E}\{\kappa^{-1}(\mathbf{x})e^{f(\mathbf{x};\boldsymbol{\beta}^*)}\dot{g}^{\otimes 2}(\mathbf{x};\boldsymbol{\theta}^*)\}\right]^{-1}$$

We only need to show that

$$[\mathbb{E}\{e^{f(\mathbf{x};\boldsymbol{\beta}^*)}\}]^{-1}\mathbf{V}_w \geq \boldsymbol{\Lambda}_{\text{lik}}^{-1}.$$

This is proved by the following calculation

$$\mathbf{0} \leq \mathbb{E}\left(\left[\{\kappa^{1/2}(\mathbf{x})\mathbf{M}_{\text{f}}^{-1} - \kappa^{-1/2}(\mathbf{x})\boldsymbol{\Lambda}_{\text{lik}}^{-1}\}e^{f(\mathbf{x};\boldsymbol{\beta}^*)/2}\dot{g}(\mathbf{x};\boldsymbol{\theta}^*)\right]^{\otimes 2}\right)$$
$$= \mathbb{E}\left\{\mathbf{M}_{\text{f}}^{-1}\kappa(\mathbf{x})e^{f(\mathbf{x};\boldsymbol{\beta}^*)}\dot{g}^{\otimes 2}(\mathbf{x};\boldsymbol{\theta}^*)\mathbf{M}_{\text{f}}^{-1}\right\} + \mathbb{E}\left\{\boldsymbol{\Lambda}_{\text{lik}}^{-1}\kappa^{-1}(\mathbf{x})e^{f(\mathbf{x};\boldsymbol{\beta}^*)}\dot{g}^{\otimes 2}(\mathbf{x};\boldsymbol{\theta}^*)\boldsymbol{\Lambda}_{\text{lik}}^{-1}\right\}$$
$$\quad - 2\mathbb{E}\left\{\mathbf{M}_{\text{f}}^{-1}e^{f(\mathbf{x};\boldsymbol{\beta}^*)}\dot{g}^{\otimes 2}(\mathbf{x};\boldsymbol{\theta}^*)\boldsymbol{\Lambda}_{\text{lik}}^{-1}\right\}$$
$$= [\mathbb{E}\{e^{f(\mathbf{x};\boldsymbol{\beta}^*)}\}]^{-1}\mathbf{V}_w + \boldsymbol{\Lambda}_{\text{lik}}^{-1} - 2\boldsymbol{\Lambda}_{\text{lik}}^{-1} = [\mathbb{E}\{e^{f(\mathbf{x};\boldsymbol{\beta}^*)}\}]^{-1}\mathbf{V}_w - \boldsymbol{\Lambda}_{\text{lik}}^{-1}.$$

If $c = 0$ then $\kappa(\mathbf{x}) = 1$, and we can directly verify that

$$[\mathbb{E}\{e^{f(\mathbf{x};\boldsymbol{\beta}^*)}\}]^{-1}\mathbf{V}_w = \boldsymbol{\Lambda}_{\text{lik}}^{-1} = \mathbf{M}_{\text{f}}^{-1}. \tag{S.29}$$

## S.7  Proof of Theorem 6

For sampled data, (5) tell us that the joint density w.r.t. the product counting measure of the responses given the features is $\exp\left\{\ell_{\text{lik}}(\boldsymbol{\theta}) + \sum_{i=1}^N \delta_i y_i l_i\right\}$, whose support has a finite number of possible values. Thus the expectation $\mathbb{E}\{\mathbf{U}(\boldsymbol{\theta};\mathcal{D}_\delta) \mid \mathbf{X}\}$ is a sum of finite number of elements, and therefore the partial derivatives w.r.t. $\boldsymbol{\theta}$ can be passed under the integration sign in $\mathbb{E}\{\mathbf{U}(\boldsymbol{\theta};\mathcal{D}_\delta) \mid \mathbf{X}\}$. Therefore we have

$$\mathbf{I} = \frac{\partial}{\partial\boldsymbol{\theta}^{\mathrm{T}}}\mathbb{E}\{\mathbf{U}(\boldsymbol{\theta};\mathcal{D}_\delta) \mid \mathbf{X}\}$$

$$= \frac{\partial}{\partial\boldsymbol{\theta}^{\mathrm{T}}}\int \mathbf{U}(\boldsymbol{\theta};\mathcal{D}_\delta)\exp\left\{\ell_{\text{lik}}(\boldsymbol{\theta}) + \sum_{i=1}^N \delta_i y_i l_i\right\}d\mathbf{y}$$

$$= \int \dot{\mathbf{U}}(\boldsymbol{\theta};\mathcal{D}_\delta)\exp\left\{\ell_{\text{lik}}(\boldsymbol{\theta}) + \sum_{i=1}^N \delta_i y_i l_i\right\}d\mathbf{y}$$

$$\quad + \int \mathbf{U}(\boldsymbol{\theta};\mathcal{D}_\delta)\exp\left\{\ell_{\text{lik}}(\boldsymbol{\theta}) + \sum_{i=1}^N \delta_i y_i l_i\right\}\dot{\ell}_{\text{lik}}^{\mathrm{T}}(\boldsymbol{\theta})d\mathbf{y}$$

$$= \mathbb{E}\{\dot{\mathbf{U}}(\boldsymbol{\theta};\mathcal{D}_\delta) \mid \mathbf{X}\} + \mathbb{E}\{\mathbf{U}(\boldsymbol{\theta};\mathcal{D}_\delta)\dot{\ell}_{\text{lik}}^{\mathrm{T}}(\boldsymbol{\theta}) \mid \mathbf{X}\},$$

where $d\mathbf{y} = dy \times dy \times ...dy$ is a product measure and $dy$ is the counting measure. This gives us that

$$\mathbb{E}\{\mathbf{U}(\boldsymbol{\theta}; \mathcal{D}_\delta)\dot{\ell}_{\mathrm{lik}}^{\mathrm{T}}(\boldsymbol{\theta}) \mid \mathbf{X}\} = \mathbf{I} \tag{S.30}$$

Hence we have

$$\begin{aligned}
&\mathbb{V}\{\mathbf{U}(\boldsymbol{\theta}; \mathcal{D}_\delta) - a_N^{-2}\boldsymbol{\Lambda}_{\mathrm{lik}}^{-1}\dot{\ell}_{\mathrm{lik}}(\boldsymbol{\theta}) \mid \mathbf{X}\} \geq \mathbf{0} \\
&= \mathbb{V}\{\mathbf{U}(\boldsymbol{\theta}; \mathcal{D}_\delta) \mid \mathbf{X}\} + \mathbb{V}\{a_N^{-2}\boldsymbol{\Lambda}_{\mathrm{lik}}^{-1}\dot{\ell}_{\mathrm{lik}}(\boldsymbol{\theta}) \mid \mathbf{X}\} - 2a_N^{-2}\mathbb{E}\{\mathbf{U}(\boldsymbol{\theta}; \mathcal{D}_\delta)\dot{\ell}_{\mathrm{lik}}^{\mathrm{T}}(\boldsymbol{\theta}) \mid \mathbf{X}\}\boldsymbol{\Lambda}_{\mathrm{lik}}^{-1} \\
&= \mathbb{V}\{\mathbf{U}(\boldsymbol{\theta}; \mathcal{D}_\delta) \mid \mathbf{X}\} + a_N^{-2}\boldsymbol{\Lambda}_{\mathrm{lik}}^{-1}\mathbb{V}\{a_N^{-1}\dot{\ell}_{\mathrm{lik}}(\boldsymbol{\theta}) \mid \mathbf{X}\}\boldsymbol{\Lambda}_{\mathrm{lik}}^{-1} - 2a_N^{-2}\boldsymbol{\Lambda}_{\mathrm{lik}}^{-1}.
\end{aligned}$$

Taking expectation of the above conditional variance and using (S.25), we have

$$\begin{aligned}
&\mathbb{E}[\mathbb{V}\{\mathbf{U}(\boldsymbol{\theta}; \mathcal{D}_\delta) \mid \mathbf{X}\}] + a_N^{-2}\boldsymbol{\Lambda}_{\mathrm{lik}}^{-1}\mathbb{E}[a_N^{-1}\mathbb{V}\{\dot{\ell}_{\mathrm{lik}}(\boldsymbol{\theta}) \mid \mathbf{X}\}]\boldsymbol{\Lambda}_{\mathrm{lik}}^{-1} - 2a_N^{-2}\boldsymbol{\Lambda}_{\mathrm{lik}}^{-1} \\
&= \mathbb{V}\{\mathbf{U}(\boldsymbol{\theta}; \mathcal{D}_\delta)\} + a_N^{-2}\boldsymbol{\Lambda}_{\mathrm{lik}}^{-1}\{1 + o_P(1)\} - 2a_N^{-2}\boldsymbol{\Lambda}_{\mathrm{lik}}^{-1} \\
&= \mathbb{V}\{\mathbf{U}(\boldsymbol{\theta}; \mathcal{D}_\delta)\} - a_N^{-2}\boldsymbol{\Lambda}_{\mathrm{lik}}^{-1}\{1 + o_P(1)\} \\
&= \mathbb{V}\{\mathbf{U}(\boldsymbol{\theta}; \mathcal{D}_\delta)\} - N_1^{-1}\mathbf{V}_{\mathrm{lik}}\{1 + o_P(1)\} \geq \mathbf{0},
\end{aligned}$$

where the last equality is because (S.1) implies that $a_N^{-2} = N_1^{-1}\mathbb{E}\{e^{f(\mathbf{x}; \boldsymbol{\beta}^*)}\}\{1 + o_P(1)\}$. This finishes the proof.

## S.8  Proof of Theorem 7

The outline of the proof is similar to that of the proof of Theorem 2. Write $\pi_\varrho^{\mathrm{os}}(\mathbf{x}, y; \tilde{\boldsymbol{\vartheta}}) = y + (1 - y)\pi_\varrho^{\mathrm{os}}(\mathbf{x}; \tilde{\boldsymbol{\vartheta}})$ and $\delta^{\tilde{\boldsymbol{\vartheta}}} = y + (1 - y)I\{u \leq \pi_\varrho^{\mathrm{os}}(\mathbf{x}; \tilde{\boldsymbol{\vartheta}})\}$ where $u \sim \mathbb{U}(0, 1)$. The estimator $\hat{\boldsymbol{\theta}}_w^{\tilde{\boldsymbol{\vartheta}}}$ is the maximizer of

$$\ell_w^{\tilde{\boldsymbol{\vartheta}}}(\boldsymbol{\theta}) = \sum_{i=1}^N \frac{\delta_i^{\tilde{\boldsymbol{\vartheta}}}}{\pi_\varrho^{\mathrm{os}}(\mathbf{x}, y; \tilde{\boldsymbol{\vartheta}})}\left[y_i g(\mathbf{x}_i; \boldsymbol{\theta}) - \log\{1 + e^{g(\mathbf{x}_i; \boldsymbol{\theta})}\}\right],$$

so $a_N(\hat{\boldsymbol{\theta}}_w^{\tilde{\boldsymbol{\vartheta}}} - \boldsymbol{\theta}^*)$ is the maximizer of $\gamma_w^{\tilde{\boldsymbol{\vartheta}}}(\mathbf{u}) = \ell_w^{\tilde{\boldsymbol{\vartheta}}}(\boldsymbol{\theta}^* + a_N^{-1}\mathbf{u}) - \ell_w^{\tilde{\boldsymbol{\vartheta}}}(\boldsymbol{\theta}^*)$. By Taylor's expansion,

$$\gamma_w^{\tilde{\boldsymbol{\vartheta}}}(\mathbf{u}) = \frac{1}{a_N}\mathbf{u}^{\mathrm{T}}\dot{\ell}_w^{\tilde{\boldsymbol{\vartheta}}}(\boldsymbol{\theta}^*) + \frac{1}{2a_N^2}\sum_{i=1}^N \frac{\delta_i^{\tilde{\boldsymbol{\vartheta}}}}{\pi_\varrho^{\mathrm{os}}(\mathbf{x}, y; \tilde{\boldsymbol{\vartheta}})}\phi(\mathbf{x}_i; \boldsymbol{\theta}^*)(\mathbf{z}_i^{\mathrm{T}}\mathbf{u})^2 + \Delta_{\ell_w}^{\tilde{\boldsymbol{\vartheta}}} + R_w^{\tilde{\boldsymbol{\vartheta}}},$$

where

$$\dot{\ell}_w^{\tilde{\boldsymbol{\vartheta}}}(\boldsymbol{\theta}) = \sum_{i=1}^N \frac{\delta_i^{\tilde{\boldsymbol{\vartheta}}}}{\pi_\varrho^{\mathrm{os}}(\mathbf{x}, y; \tilde{\boldsymbol{\vartheta}})}\{y_i - p(\mathbf{x}_i; \boldsymbol{\theta})\}\dot{g}(\mathbf{x}_i, \boldsymbol{\theta}),$$

$$\Delta_{\ell_w}^{\tilde{\boldsymbol{\vartheta}}} = \frac{1}{2a_N^2}\sum_{i=1}^N \frac{\delta_i^{\tilde{\boldsymbol{\vartheta}}}}{\pi_\varrho^{\mathrm{os}}(\mathbf{x}, y; \tilde{\boldsymbol{\vartheta}})}\{y_i - p(\mathbf{x}_i; \boldsymbol{\theta}^*)\}\mathbf{u}^{\mathrm{T}}\ddot{g}(\mathbf{x}_i, \boldsymbol{\theta}^*)\mathbf{u},$$

and $R_w^{\tilde{\boldsymbol{\vartheta}}}$ is the remainder. Similarly to the proof of Theorem 2, we only need to show that

$$a_N^{-1}\dot{\ell}_w^{\tilde{\boldsymbol{\vartheta}}}(\boldsymbol{\theta}^*) \longrightarrow \mathbb{N}(\mathbf{0}, \Lambda_w^{\mathrm{plt}}), \tag{S.31}$$

in distribution with

$$\Lambda_w^{\mathrm{plt}} = \mathbb{E}\left[e^{f(\mathbf{x}_i; \boldsymbol{\beta}^*)}\dot{g}^{\otimes 2}(\mathbf{x}_i; \boldsymbol{\theta}^*) + \frac{ce^{2f(\mathbf{x}_i; \boldsymbol{\beta}^*)}}{\max\{\varphi_{\mathrm{plt}}(\mathbf{x}), c_l\}}\dot{g}^{\otimes 2}(\mathbf{x}_i; \boldsymbol{\theta}^*)\right],$$

and for any $\mathbf{u}$,

$$H_w^{\tilde{\boldsymbol{\vartheta}}} := a_N^{-2}\sum_{i=1}^N \frac{\delta_i^{\tilde{\boldsymbol{\vartheta}}}}{\pi_\varrho^{\mathrm{os}}(\mathbf{x}, y; \tilde{\boldsymbol{\vartheta}})}\phi(\mathbf{x}_i; \boldsymbol{\theta}^*)\dot{g}^{\otimes 2}(\mathbf{x}_i; \boldsymbol{\theta}^*) \longrightarrow \mathbf{M}_{\mathrm{f}}, \tag{S.32}$$

in probability, and $\Delta_{\ell_w}^{\tilde{\boldsymbol{\vartheta}}} = o_P(1)$ and $R_w^{\tilde{\boldsymbol{\vartheta}}} = o_P(1)$.

We prove (S.31) first. Let $\eta_i^{\tilde{\boldsymbol{\vartheta}}} = \frac{\delta_i^{\tilde{\boldsymbol{\vartheta}}}}{\pi_\varrho^{\mathrm{os}}(\mathbf{x},y;\tilde{\boldsymbol{\vartheta}})}\{y_i - p(\mathbf{x}_i;\boldsymbol{\theta}^*)\}\dot{g}(\mathbf{x}_i;\boldsymbol{\theta}^*)$. Given $\tilde{\boldsymbol{\vartheta}}$, $\eta_i^{\tilde{\boldsymbol{\vartheta}}}$, $i = 1, ..., N$, are i.i.d. with mean zero, and the variance satisfies that

$$\mathbb{V}(\eta_i^{\tilde{\boldsymbol{\vartheta}}} \mid \tilde{\boldsymbol{\vartheta}})$$

$$= \mathbb{E}\left[\frac{\{y_i - p(\mathbf{x}_i;\boldsymbol{\theta}^*)\}^2}{\{y_i + (1-y_i)\pi_\varrho^{\mathrm{os}}(\mathbf{x}_i;\tilde{\boldsymbol{\vartheta}})\}}\dot{g}^{\otimes 2}(\mathbf{x}_i;\boldsymbol{\theta}^*) \,\Big|\, \tilde{\boldsymbol{\vartheta}}\right]$$

$$= \mathbb{E}\left[p(\mathbf{x}_i;\boldsymbol{\theta}^*)\{1-p(\mathbf{x}_i;\boldsymbol{\theta}^*)\}^2\dot{g}^{\otimes 2}(\mathbf{x}_i;\boldsymbol{\theta}^*) + \{1-p(\mathbf{x}_i;\boldsymbol{\theta}^*)\}\frac{p^2(\mathbf{x}_i;\boldsymbol{\theta}^*)}{\pi_\varrho^{\mathrm{os}}(\mathbf{x}_i;\tilde{\boldsymbol{\vartheta}})}\dot{g}^{\otimes 2}(\mathbf{x}_i;\boldsymbol{\theta}^*) \,\Big|\, \tilde{\boldsymbol{\vartheta}}\right]$$

$$\leq \mathbb{E}\{p(\mathbf{x}_i;\boldsymbol{\theta}^*)\dot{g}^{\otimes 2}(\mathbf{x}_i;\boldsymbol{\theta}^*) + \varrho^{-1}p^2(\mathbf{x}_i;\boldsymbol{\theta}^*)\dot{g}^{\otimes 2}(\mathbf{x}_i;\boldsymbol{\theta}^*) \mid \tilde{\boldsymbol{\vartheta}}\}$$

$$\leq e^{\alpha^*}\mathbb{E}\{e^{f(\mathbf{x}_i;\boldsymbol{\beta}^*)}\dot{g}^{\otimes 2}(\mathbf{x}_i;\boldsymbol{\theta}^*) + (\varrho^{-1}e^{\alpha^*})e^{2f(\mathbf{x}_i;\boldsymbol{\beta}^*)}\dot{g}^{\otimes 2}(\mathbf{x}_i;\boldsymbol{\theta}^*)\}$$

Since $\rho^{-1}\varrho \to c_l > 0$, for large enough $N$,

$$\mathbb{V}\{a_N^{-1}\dot{\ell}_w^{\tilde{\boldsymbol{\vartheta}}}(\boldsymbol{\theta}^*) \mid \tilde{\boldsymbol{\vartheta}}\} \leq \mathbb{E}\{e^{f(\mathbf{x}_i;\boldsymbol{\beta}^*)}\dot{g}^{\otimes 2}(\mathbf{x}_i;\boldsymbol{\theta}^*) + (1 + c_l^{-1}c)e^{2f(\mathbf{x}_i;\boldsymbol{\beta}^*)}\dot{g}^{\otimes 2}(\mathbf{x}_i;\boldsymbol{\theta}^*)\}.$$

Thus, we know that

$$\mathbb{V}\{a_N^{-1}\dot{\ell}_w^{\tilde{\boldsymbol{\vartheta}}}(\boldsymbol{\theta}^*)\} \to \Lambda_w^{\mathrm{plt}}$$

Now we check the Lindeberg-Feller condition [Section $^*$2.8 of 4] given $\tilde{\boldsymbol{\vartheta}}$. For any $\epsilon > 0$,

$$\sum_{i=1}^N \mathbb{E}\{\|\eta_i^{\tilde{\boldsymbol{\vartheta}}}\|^2 I(\|\eta_i^{\tilde{\boldsymbol{\vartheta}}}\| > a_N\epsilon) \mid \tilde{\boldsymbol{\vartheta}}\}$$

$$= N\mathbb{E}\big[\|\{\pi_\varrho^{\mathrm{os}}(\mathbf{x},y;\tilde{\boldsymbol{\vartheta}})\}^{-1}\delta^{\tilde{\boldsymbol{\vartheta}}}\{y - p(\mathbf{x};\boldsymbol{\theta}^*)\}\dot{g}(\mathbf{x};\boldsymbol{\theta}^*)\|^2$$

$$\times I(\|\{\pi_\varrho^{\mathrm{os}}(\mathbf{x},y;\tilde{\boldsymbol{\vartheta}})\}^{-1}\delta^{\tilde{\boldsymbol{\vartheta}}}\{y - p(\mathbf{x};\boldsymbol{\theta}^*)\}\dot{g}(\mathbf{x};\boldsymbol{\theta}^*)\| > a_N\epsilon) \mid \tilde{\boldsymbol{\vartheta}}\big]$$

$$= N\mathbb{E}\big[\{\pi_\varrho^{\mathrm{os}}(\mathbf{x},y;\tilde{\boldsymbol{\vartheta}})\}^{-1}\|\{y - p(\mathbf{x};\boldsymbol{\theta}^*)\}\dot{g}(\mathbf{x};\boldsymbol{\theta}^*)\|^2$$

$$\times I(\|\{\pi_\varrho^{\mathrm{os}}(\mathbf{x},y;\tilde{\boldsymbol{\vartheta}})\}^{-1}\{y - p(\mathbf{x};\boldsymbol{\theta}^*)\}\dot{g}(\mathbf{x};\boldsymbol{\theta}^*)\| > a_N\epsilon) \mid \tilde{\boldsymbol{\vartheta}}\big]$$

$$+ N\mathbb{E}\big[\{1 - \pi_\varrho^{\mathrm{os}}(\mathbf{x},y;\tilde{\boldsymbol{\vartheta}})\}\|\{\pi_\varrho^{\mathrm{os}}(\mathbf{x},y;\tilde{\boldsymbol{\vartheta}})\}^{-1}y\{y - p(\mathbf{x};\boldsymbol{\theta}^*)\}\dot{g}(\mathbf{x};\boldsymbol{\theta}^*)\|^2$$

$$\times I(\|\{\pi_\varrho^{\mathrm{os}}(\mathbf{x},y;\tilde{\boldsymbol{\vartheta}})\}^{-1}y\{y - p(\mathbf{x};\boldsymbol{\theta}^*)\}\dot{g}(\mathbf{x};\boldsymbol{\theta}^*)\| > a_N\epsilon) \mid \tilde{\boldsymbol{\vartheta}}\big]$$

$$\leq N\mathbb{E}\big[p(\mathbf{x};\boldsymbol{\theta}^*)\|\dot{g}(\mathbf{x};\boldsymbol{\theta}^*)\|^2 I(\|\dot{g}(\mathbf{x};\boldsymbol{\theta}^*)\| > a_N\epsilon)\big]$$

$$+ N\mathbb{E}\big[\varrho^{-1}\|p(\mathbf{x};\boldsymbol{\theta}^*)\dot{g}(\mathbf{x};\boldsymbol{\theta}^*)\|^2 I(\|\varrho^{-1}p(\mathbf{x};\boldsymbol{\theta}^*)\dot{g}(\mathbf{x};\boldsymbol{\theta}^*)\| > a_N\epsilon)\big]$$

$$+ N\mathbb{E}\big[p(\mathbf{x};\boldsymbol{\theta}^*)\|\dot{g}(\mathbf{x};\boldsymbol{\theta}^*)\|^2 I(\|\dot{g}(\mathbf{x};\boldsymbol{\theta}^*)\| > a_N\epsilon)\big]$$

$$\leq 2a_N^2\mathbb{E}\big[e^{f(\mathbf{x}_i;\boldsymbol{\beta}^*)}\|\dot{g}(\mathbf{x};\boldsymbol{\theta}^*)\|^2 I(\|\dot{g}(\mathbf{x};\boldsymbol{\theta}^*)\| > a_N\epsilon)\big]$$

$$+ a_N^2 e^{\alpha^*}\varrho^{-1}\mathbb{E}\big[e^{2f(\mathbf{x}_i;\boldsymbol{\beta}^*)}\|\dot{g}(\mathbf{x};\boldsymbol{\theta}^*)\|^2 I(\varrho^{-1}e^{\alpha^*}\|e^{f(\mathbf{x}_i;\boldsymbol{\beta}^*)}\dot{g}(\mathbf{x};\boldsymbol{\theta}^*)\| > a_N\epsilon)\big]$$

$$= o(a_N^2),$$

where the last step is due to Assumptions 1, the dominated convergence theorem, and the facts that $a_N \to \infty$ and $\lim_{N\to\infty} e^\alpha/\varrho = c/c_l < \infty$. Thus, applying the Lindeberg-Feller central limit theorem [Section $^*$2.8 of 4] finishes the proof of (S.31).

Now we prove (S.32). We notice that

$$\mathbb{E}(H_w^{\tilde{\boldsymbol{\vartheta}}} \mid \tilde{\boldsymbol{\vartheta}}) = \mathbb{E}\left\{\frac{e^{f(\mathbf{x};\boldsymbol{\beta}^*)}}{(1 + e^{\alpha^* + f(\mathbf{x};\boldsymbol{\beta}^*)})^2}\dot{g}^{\otimes 2}(\mathbf{x};\boldsymbol{\theta}^*)\right\} = \mathbf{M}_{\mathrm{f}} + o(1), \tag{S.33}$$

where the last step is by the dominated convergence theorem. In addition, the conditional variance of each component of $H_w^{\tilde{\boldsymbol{\vartheta}}}$ is bounded by

$$\frac{1}{N}\mathbb{E}\left\{\frac{e^{2f(\mathbf{x};\boldsymbol{\beta}^*)}\|\dot{g}(\mathbf{x};\boldsymbol{\theta}^*)\|^4}{\pi_\varrho^{\mathrm{os}}(\mathbf{x},y;\tilde{\boldsymbol{\vartheta}})} \,\Big|\, \tilde{\boldsymbol{\vartheta}}\right\} \leq \frac{1}{N\varrho}\mathbb{E}\{e^{2f(\mathbf{x};\boldsymbol{\beta}^*)}\|\dot{g}(\mathbf{x};\boldsymbol{\theta}^*)\|^4\} = o(1). \tag{S.34}$$

where the last step is because of Assumption 1 and the fact that $Ne^{\alpha^*} \to \infty$, $e^{\alpha^*}/\rho \to c < \infty$, and $\rho^{-1}\varrho \to c_l > 0$ imply that $N\varrho \to \infty$. From (S.33) and (S.34), applying Chebyshev's inequality finishes the proof of (S.32).

In the following we finish the proof by showing that given $\tilde{\boldsymbol{\vartheta}}$, $\Delta_{\ell_w}^{\tilde{\boldsymbol{\vartheta}}} = o_P(1)$ and $R_w^{\tilde{\boldsymbol{\vartheta}}} = o_P(1)$.

For $\Delta_{\ell_w}^{\tilde{\boldsymbol{\vartheta}}}$, the conditional mean $\mathbb{E}(\Delta_{\ell_w}^{\tilde{\boldsymbol{\vartheta}}} \mid \tilde{\boldsymbol{\vartheta}}) = \mathbf{0}$ and the conditional variance satisfies that

$$
\begin{aligned}
\mathbb{V}(\Delta_{\ell_w}^{\tilde{\boldsymbol{\vartheta}}} \mid \tilde{\boldsymbol{\vartheta}}) &\leq \frac{\|\mathbf{u}\|^4 N}{4a_N^4} \mathbb{E}\left[ \frac{\{y - p(\mathbf{x};\boldsymbol{\theta}^*)\}^2 \|\ddot{g}(\mathbf{x};\boldsymbol{\theta}^*)\|^2}{y + (1-y)\pi_\varrho^{\mathrm{os}}(\mathbf{x},y;\tilde{\boldsymbol{\vartheta}})} \,\middle|\, \tilde{\boldsymbol{\vartheta}} \right] \\
&\leq \frac{\|\mathbf{u}\|^4 N}{4a_N^4} \mathbb{E}\left[ \{1 + \varrho^{-1} p(\mathbf{x};\boldsymbol{\theta}^*)\} p(\mathbf{x};\boldsymbol{\theta}^*) \|\ddot{g}(\mathbf{x};\boldsymbol{\theta}^*)\|^2 \right] \\
&\leq \frac{\|\mathbf{u}\|^4}{4a_N^2} \mathbb{E}\left[ \{1 + \varrho^{-1} e^{\alpha^*} e^{f(\mathbf{x};\boldsymbol{\beta}^*)}\} e^{f(\mathbf{x};\boldsymbol{\beta}^*)} \|\ddot{g}(\mathbf{x};\boldsymbol{\theta}^*)\|^2 \right] = o(1),
\end{aligned}
$$

so $\Delta_{\ell_w}^{\tilde{\boldsymbol{\vartheta}}} = o_P(1)$.

For the remainder term $R_w^{\tilde{\boldsymbol{\vartheta}}}$. By direct calculations similar to those used for the proof of Theorem 2, we know that

$$
\begin{aligned}
|R_w^{\tilde{\boldsymbol{\vartheta}}}| &\leq \frac{d^3 \|\mathbf{u}\|^3 e^{a_N^{-1}\|\mathbf{u}\|}}{6Na_N} \sum_{i=1}^N \frac{\delta_i^{\tilde{\boldsymbol{\vartheta}}}}{\pi_\varrho^{\mathrm{os}}(\mathbf{x},y;\tilde{\boldsymbol{\vartheta}})} B(\mathbf{x}_i) + \frac{d\|\mathbf{u}\|^3}{6a_N^3} \sum_{i=1}^N \frac{\delta_i^{\tilde{\boldsymbol{\vartheta}}}}{\pi_\varrho^{\mathrm{os}}(\mathbf{x},y;\tilde{\boldsymbol{\vartheta}})} y_i B(\mathbf{x}_i) \\
&\equiv \Delta_{R_w 1}^{\tilde{\boldsymbol{\vartheta}}} + \Delta_{R_w 2}^{\tilde{\boldsymbol{\vartheta}}}.
\end{aligned}
$$

From Assumption 1, $\mathbb{E}(\Delta_{R_w 1}^{\tilde{\boldsymbol{\vartheta}}}) \to 0$ and $\mathbb{E}(\Delta_{R_w 2}^{\tilde{\boldsymbol{\vartheta}}}) \to 0$. Since $\Delta_{R_w 1}^{\tilde{\boldsymbol{\vartheta}}}$ and $\Delta_{R_w 2}^{\tilde{\boldsymbol{\vartheta}}}$ are both positive, Markov's inequality shows that they are both $o_P(1)$ so $R_w^{\tilde{\boldsymbol{\vartheta}}} = o_P(1)$.

## S.9 Proof of Theorem 8

The outline of the proof is similar to that of the proof of Theorem 4. The estimator $\hat{\boldsymbol{\theta}}_{\mathrm{lik}}^{\tilde{\boldsymbol{\vartheta}}}$ is the maximizer of $\ell_{\mathrm{lik}}^{\tilde{\boldsymbol{\vartheta}}}(\boldsymbol{\theta})$, so $\mathbf{u}_N^{\tilde{\boldsymbol{\vartheta}}} = a_N^{-1}(\hat{\boldsymbol{\theta}}_{\mathrm{lik}}^{\tilde{\boldsymbol{\vartheta}}} - \boldsymbol{\theta})$ is the maximizer of

$$
\gamma_{\mathrm{lik}}^{\tilde{\boldsymbol{\vartheta}}}(\mathbf{u}) = \ell_{\mathrm{lik}}^{\tilde{\boldsymbol{\vartheta}}}(\boldsymbol{\theta} + a_N^{-1}\mathbf{u}) - \ell_{\mathrm{lik}}^{\tilde{\boldsymbol{\vartheta}}}(\boldsymbol{\theta}).
$$

By Taylor's expansion,

$$
\gamma_{\mathrm{lik}}^{\tilde{\boldsymbol{\vartheta}}}(\mathbf{u}) = a_N^{-1} \mathbf{u}^{\mathrm{T}} \dot{\ell}_{\mathrm{lik}}^{\tilde{\boldsymbol{\vartheta}}}(\boldsymbol{\theta}^*) + 0.5 a_N^{-2} \sum_{i=1}^N \phi_\pi(\mathbf{x}_i;\boldsymbol{\theta}^*) \{\mathbf{u}^{\mathrm{T}} \dot{g}(\mathbf{x}_i;\boldsymbol{\theta}^*)\}^2 + \Delta_{\ell_{\mathrm{lik}}}^{\tilde{\boldsymbol{\vartheta}}} + R_{\mathrm{lik}}^{\tilde{\boldsymbol{\vartheta}}}, \quad (\mathrm{S.35})
$$

where $\phi_\pi^{\tilde{\boldsymbol{\vartheta}}}(\mathbf{x};\boldsymbol{\theta}) = p_\pi^{\tilde{\boldsymbol{\vartheta}}}(\mathbf{x};\boldsymbol{\theta})\{1 - p_\pi^{\tilde{\boldsymbol{\vartheta}}}(\mathbf{x};\boldsymbol{\theta})\}$,

$$
p_\pi^{\tilde{\boldsymbol{\vartheta}}}(\mathbf{x};\boldsymbol{\theta}) = \frac{e^{g(\mathbf{x};\boldsymbol{\theta})+\tilde{l}}}{1 + e^{g(\mathbf{x};\boldsymbol{\theta})+\tilde{l}}}, \quad \text{with} \quad \tilde{l}_i = -\log\{\pi_\varrho^{\mathrm{os}}(\mathbf{x};\tilde{\boldsymbol{\vartheta}})\}, \quad (\mathrm{S.36})
$$

$$
\dot{\ell}_{\mathrm{lik}}^{\tilde{\boldsymbol{\vartheta}}}(\boldsymbol{\theta}) = \sum_{i=1}^N \delta_i^{\tilde{\boldsymbol{\vartheta}}} \{y_i - p_\pi^{\tilde{\boldsymbol{\vartheta}}}(\mathbf{x}_i;\boldsymbol{\theta})\} \dot{g}(\mathbf{x}_i;\boldsymbol{\theta}),
$$

$$
\Delta_{\ell_{\mathrm{lik}}}^{\tilde{\boldsymbol{\vartheta}}} = \frac{1}{2a_N^2} \sum_{i=1}^N \delta_i^{\tilde{\boldsymbol{\vartheta}}} \{y_i - p_\pi^{\tilde{\boldsymbol{\vartheta}}}(\mathbf{x}_i;\boldsymbol{\theta}^*)\} \mathbf{u}^{\mathrm{T}} \ddot{g}(\mathbf{x}_i;\boldsymbol{\theta}^*)\mathbf{u}, \quad (\mathrm{S.37})
$$

and $R_{\mathrm{lik}}^{\tilde{\boldsymbol{\vartheta}}}$ is the remainder term. By direct calculations similar to those used for the proof of Theorem 4, we know that

$$
|R_{\mathrm{lik}}^{\tilde{\boldsymbol{\vartheta}}}| \leq \frac{\|\mathbf{u}\|^3}{6a_N^3} \sum_{i=1}^N \delta_i^{\tilde{\boldsymbol{\vartheta}}} C(\mathbf{x}_i, \boldsymbol{\theta}^* + a_N^{-1}\acute{\mathbf{u}}) + \frac{d\|\mathbf{u}\|^3}{6a_N^3} \sum_{i=1}^N \delta_i^{\tilde{\boldsymbol{\vartheta}}} y_i B(\mathbf{x}_i) \leq \frac{(d+1)\|\mathbf{u}\|^3}{6a_N^3} \sum_{i=1}^N \delta_i^{\tilde{\boldsymbol{\vartheta}}} B(\mathbf{x}_i),
$$

where $\acute{\mathbf{u}}$ lies between $\mathbf{0}$ and $\mathbf{u}$.

Since $\varrho = o(\rho)$, we know that for large enough $N$,

$$\mathbb{E}\Big\{a_N^{-3}\sum_{i=1}^{N}\delta_i^{\tilde{\vartheta}}B(\mathbf{x}_i)\ \Big|\ \tilde{\vartheta}\Big\}$$

$$= Na_N^{-3}\mathbb{E}\Big\{[y+(1-y)\pi_\varrho^{\mathrm{os}}(\mathbf{x};\tilde{\vartheta})]B(\mathbf{x})\ \Big|\ \tilde{\vartheta}\Big\}$$

$$\leq Na_N^{-3}\mathbb{E}\Big\{[e^{\alpha^*}e^{f(\mathbf{x};\boldsymbol{\beta}^*)}+\tilde{\omega}^{-1}\rho t(\mathbf{x};\tilde{\boldsymbol{\theta}})]B(\mathbf{x})\ \Big|\ \tilde{\vartheta}\Big\}$$

$$\leq a_N^{-1}\mathbb{E}\big\{e^{f(\mathbf{x};\boldsymbol{\beta}^*)}B(\mathbf{x})\big\}+a_N^{-1}\tilde{\lambda}_{\min}^{-1}(e^{-\alpha^*}\rho)(e^{\tilde{\alpha}}\tilde{\omega})^{-1}\mathbb{E}\big\{e^{f(\mathbf{x};\tilde{\boldsymbol{\theta}})}\|\dot{g}(\mathbf{x};\tilde{\boldsymbol{\theta}})\|B(\mathbf{x})\ \big|\ \tilde{\vartheta}\big\}$$

$$\leq a_N^{-1}\mathbb{E}\big\{e^{f(\mathbf{x};\boldsymbol{\beta}^*)}B(\mathbf{x})\big\}+a_N^{-1}\tilde{\lambda}_{\min}^{-1}(e^{-\alpha^*}\rho)(e^{\tilde{\alpha}}\tilde{\omega})^{-1}\mathbb{E}\big\{B^2(\mathbf{x})\big\}=o_P(1),$$

where $\tilde{\lambda}_{\min}$ is the minimum eigenvalue of $\tilde{\mathbf{M}}_{\mathrm{f}}$. Thus, Markov's inequality shows that $R_{\mathrm{lik}}^{\tilde{\vartheta}}=o_P(1)$. For $\Delta_{\ell_{\mathrm{lik}}}^{\tilde{\vartheta}}$, the conditional mean $\mathbb{E}(\Delta_{\ell_{\mathrm{lik}}}^{\tilde{\vartheta}}\mid\tilde{\vartheta})=\mathbf{0}$ and the conditional variance satisfies that

$$\mathbb{V}(\Delta_{\ell_{\mathrm{lik}}}^{\tilde{\vartheta}}\mid\tilde{\vartheta})\leq\frac{\|\mathbf{u}\|^4}{4a_N^4}\sum_{i=1}^{N}\mathbb{E}\{\delta_i^{\tilde{\vartheta}}p_\pi^{\tilde{\vartheta}}(\mathbf{x}_i;\boldsymbol{\theta}^*)\|\ddot{g}(\mathbf{x}_i;\boldsymbol{\theta}^*)\|^2\mid\tilde{\vartheta}\}$$

$$\leq\frac{\|\mathbf{u}\|^4}{4a_N^2}\mathbb{E}[e^{f(\mathbf{x};\boldsymbol{\beta}^*)}\|\ddot{g}(\mathbf{x},\boldsymbol{\theta}^*)\|^2]\to 0,$$

so $\Delta_{\ell_{\mathrm{lik}}}^{\tilde{\vartheta}}=o_P(1)$.

Now we need to show that given $\tilde{\vartheta}$

$$a_N^{-1}\dot{\ell}_{\mathrm{lik}}^{\tilde{\vartheta}}(\boldsymbol{\theta}^*)\longrightarrow\mathbb{N}\big(\mathbf{0},\ \boldsymbol{\Lambda}_{\mathrm{lik}}^{\mathrm{plt}}\big),\tag{S.38}$$

in distribution, with

$$\boldsymbol{\Lambda}_{\mathrm{lik}}^{\mathrm{plt}}=\mathbb{E}\left[\frac{e^{f(\mathbf{x};\boldsymbol{\beta}^*)}\dot{g}^{\otimes 2}(\mathbf{x};\boldsymbol{\theta}^*)}{1+c\varphi_{\mathrm{plt}}^{-1}(\mathbf{x})e^{f(\mathbf{x};\boldsymbol{\beta}^*)}}\right],$$

and

$$H_{\mathrm{lik}}^{\tilde{\vartheta}}:=a_N^{-2}\sum_{i=1}^{N}\delta_i^{\tilde{\vartheta}}\phi_\pi^{\tilde{\vartheta}}(\mathbf{x}_i;\boldsymbol{\theta}^*)\dot{g}^{\otimes 2}(\mathbf{x}_i;\boldsymbol{\theta}^*)\longrightarrow\boldsymbol{\Lambda}_{\mathrm{lik}}^{\mathrm{plt}},\tag{S.39}$$

in probability, then from the Basic Corollary in page 2 of [3], we know that $a_N(\hat{\boldsymbol{\theta}}_{\mathrm{lik}}^{\tilde{\vartheta}}-\boldsymbol{\theta}^*)$, the maximizer of $\gamma_{\mathrm{lik}}^{\tilde{\vartheta}}(\mathbf{u})$, satisfies that

$$a_N(\hat{\boldsymbol{\theta}}_{\mathrm{lik}}^{\tilde{\vartheta}}-\boldsymbol{\theta}^*)=(\boldsymbol{\Lambda}_{\mathrm{lik}}^{\mathrm{plt}})^{-1}\times a_N^{-1}\dot{\ell}_{\mathrm{lik}}^{\tilde{\vartheta}}(\boldsymbol{\theta}^*)+o_P(1).\tag{S.40}$$

Slutsky's theorem together with (S.38) and (S.40) implies the result in Theorem 1. We prove (S.38) and (S.39) in the following.

Given $\tilde{\vartheta}$, $\dot{\ell}_{\mathrm{lik}}^{\tilde{\vartheta}}(\boldsymbol{\theta}^*)$ is a summation of i.i.d. quantities, $\delta_i^{\tilde{\vartheta}}\{y_i-p_\pi^{\tilde{\vartheta}}(\mathbf{x}_i;\boldsymbol{\theta}^*)\}\dot{g}(\mathbf{x}_i;\boldsymbol{\theta}^*)$'s. Using a similar approach to obtain (S.18) we know that $\mathbb{E}\{a_N^{-1}\dot{\ell}_{\mathrm{lik}}^{\tilde{\vartheta}}(\boldsymbol{\theta}^*)\mid\tilde{\vartheta}\}=\mathbf{0}$. The conditional variance of $a_N^{-1}\dot{\ell}_{\mathrm{lik}}^{\tilde{\vartheta}}(\boldsymbol{\theta}^*)$ satisfies that

$$\mathbb{V}\{a_N^{-1}\dot{\ell}_{\mathrm{lik}}^{\tilde{\vartheta}}(\boldsymbol{\theta}^*)\mid\tilde{\vartheta}\}$$

$$=e^{-\alpha^*}\mathbb{E}\big[\{y+(1-y)\pi_\varrho^{\mathrm{os}}(\mathbf{x};\tilde{\vartheta})\}\{y-p_\pi^{\tilde{\vartheta}}(\mathbf{x};\boldsymbol{\theta}^*)\}^2\dot{g}^{\otimes 2}(\mathbf{x};\boldsymbol{\theta}^*)\mid\tilde{\vartheta}\big]$$

$$=e^{-\alpha^*}\mathbb{E}\big([p(\mathbf{x};\boldsymbol{\theta}^*)\{1-p_\pi^{\tilde{\vartheta}}(\mathbf{x};\boldsymbol{\theta}^*)\}^2+\pi_\varrho^{\mathrm{os}}(\mathbf{x};\tilde{\vartheta})\{1-p(\mathbf{x};\boldsymbol{\theta}^*)\}p_\pi^2(\mathbf{x};\boldsymbol{\theta}^*)]\dot{g}^{\otimes 2}(\mathbf{x};\boldsymbol{\theta}^*)\ \big|\ \tilde{\vartheta}\big)$$

$$=e^{-\alpha^*}\mathbb{E}\big([\{1-p_\pi^{\tilde{\vartheta}}(\mathbf{x};\boldsymbol{\theta}^*)\}^2+\phi_\pi^{\tilde{\vartheta}}(\mathbf{x};\boldsymbol{\theta}^*)\{1-p(\mathbf{x};\boldsymbol{\theta}^*)\}^2]p(\mathbf{x};\boldsymbol{\theta}^*)\dot{g}^{\otimes 2}(\mathbf{x};\boldsymbol{\theta}^*)\mid\tilde{\vartheta}\big)$$

$$=\mathbb{E}\big([\{1-p_\pi^{\tilde{\vartheta}}(\mathbf{x};\boldsymbol{\theta}^*)\}^2+\phi_\pi^{\tilde{\vartheta}}(\mathbf{x};\boldsymbol{\theta}^*)\{1-p(\mathbf{x};\boldsymbol{\theta}^*)\}^2]\{1-p(\mathbf{x};\boldsymbol{\theta}^*)\}e^{f(\mathbf{x};\boldsymbol{\beta}^*)}\dot{g}^{\otimes 2}(\mathbf{x};\boldsymbol{\theta}^*)\mid\tilde{\vartheta}\big).$$

The elements of the term on the right hand side of the above equation are bounded by an integrable random variable $e^{f(\mathbf{x};\boldsymbol{\beta}^*)}\|\dot{g}(\mathbf{x};\boldsymbol{\theta}^*)\|^2$, so if $\varrho = o(\rho)$ then

$$\mathbb{V}\{a_N^{-1}\dot{\ell}_{\text{lik}}^{\tilde{\boldsymbol{\vartheta}}}(\boldsymbol{\theta}^*)\} \longrightarrow \boldsymbol{\Lambda}_{\text{lik}}^{\text{plt}}. \tag{S.41}$$

Using an similar approach as in checking the Lindeberg-Feller condition in the proof of Theorem 4, we obtain that for any $\epsilon > 0$,

$$\sum_{i=1}^{N}\mathbb{E}\Big[\|\delta_i^{\tilde{\boldsymbol{\vartheta}}}\{y_i - p_\pi^{\tilde{\boldsymbol{\vartheta}}}(\mathbf{x}_i;\boldsymbol{\theta}^*)\}\dot{g}(\mathbf{x}_i;\boldsymbol{\theta}^*)\|^2 I(\|\delta_i^{\tilde{\boldsymbol{\vartheta}}}\{y_i - p_\pi^{\tilde{\boldsymbol{\vartheta}}}(\mathbf{x}_i;\boldsymbol{\theta}^*)\}\dot{g}(\mathbf{x}_i;\boldsymbol{\theta}^*)\| > a_N\epsilon)\ \Big|\ \tilde{\boldsymbol{\vartheta}}\Big]$$

$$= N\mathbb{E}\Big[\delta^{\tilde{\boldsymbol{\vartheta}}}\|\{y - p_\pi^{\tilde{\boldsymbol{\vartheta}}}(\mathbf{x};\boldsymbol{\theta}^*)\}\dot{g}(\mathbf{x};\boldsymbol{\theta}^*)\|^2 I(\delta^{\tilde{\boldsymbol{\vartheta}}}\|\{y - p_\pi^{\tilde{\boldsymbol{\vartheta}}}(\mathbf{x};\boldsymbol{\theta}^*)\}\dot{g}(\mathbf{x};\boldsymbol{\theta}^*)\| > a_N\epsilon)\ \Big|\ \tilde{\boldsymbol{\vartheta}}\Big]$$

$$= N\mathbb{E}\Big[p(\mathbf{x};\boldsymbol{\theta}^*)\{1 - p_\pi^{\tilde{\boldsymbol{\vartheta}}}(\mathbf{x};\boldsymbol{\theta}^*)\}^2\|\dot{g}(\mathbf{x};\boldsymbol{\theta}^*)\|^2 I(\{1 - p_\pi^{\tilde{\boldsymbol{\vartheta}}}(\mathbf{x};\boldsymbol{\theta}^*)\}\|\dot{g}(\mathbf{x};\boldsymbol{\theta}^*)\| > a_N\epsilon)\ \Big|\ \tilde{\boldsymbol{\vartheta}}\Big]$$

$$\quad + N\mathbb{E}\Big[\{1 - p(\mathbf{x};\boldsymbol{\theta}^*)\}\pi_\varrho^{\text{os}}(\mathbf{x};\tilde{\boldsymbol{\vartheta}})\{p_\pi^{\tilde{\boldsymbol{\vartheta}}}(\mathbf{x};\boldsymbol{\theta}^*)\}^2\|\dot{g}(\mathbf{x};\boldsymbol{\theta}^*)\|^2$$

$$\qquad\qquad\qquad\qquad \times I(\pi_\varrho^{\text{os}}(\mathbf{x};\tilde{\boldsymbol{\vartheta}})p_\pi^{\tilde{\boldsymbol{\vartheta}}}(\mathbf{x};\boldsymbol{\theta}^*)\|\dot{g}(\mathbf{x};\boldsymbol{\theta}^*)\| > a_N\epsilon)\ \Big|\ \tilde{\boldsymbol{\vartheta}}\Big]$$

$$\leq N\mathbb{E}\Big[p(\mathbf{x};\boldsymbol{\theta}^*)\|\dot{g}(\mathbf{x};\boldsymbol{\theta}^*)\|^2 I(\|\dot{g}(\mathbf{x};\boldsymbol{\theta}^*)\| > a_N\epsilon)\ \Big|\ \tilde{\boldsymbol{\vartheta}}\Big]$$

$$\quad + N\mathbb{E}\Big[e^{\alpha^* + f(\mathbf{x};\boldsymbol{\beta}^*)}p_\pi^{\tilde{\boldsymbol{\vartheta}}}(\mathbf{x};\boldsymbol{\theta}^*)\|\dot{g}(\mathbf{x};\boldsymbol{\theta}^*)\|^2 I(e^{\alpha^* + f(\mathbf{x};\boldsymbol{\beta}^*)}\|\dot{g}(\mathbf{x};\boldsymbol{\theta}^*)\| > a_N\epsilon)\ \Big|\ \tilde{\boldsymbol{\vartheta}}\Big]$$

$$\leq a_N^2\mathbb{E}\{e^{f(\mathbf{x};\boldsymbol{\beta}^*)}\|\dot{g}(\mathbf{x};\boldsymbol{\theta}^*)\|^2 I(\|\dot{g}(\mathbf{x};\boldsymbol{\theta}^*)\| > a_N\epsilon)\}$$

$$\quad + a_N^2\mathbb{E}\big[e^{f(\mathbf{x};\boldsymbol{\beta}^*)}\|\dot{g}(\mathbf{x};\boldsymbol{\theta}^*)\|^2 I(e^{\alpha^* + f(\mathbf{x};\boldsymbol{\beta}^*)}\|\dot{g}(\mathbf{x};\boldsymbol{\theta}^*)\| > a_N\epsilon)\big]$$

$$= o(a_N^2),$$

where the last step is from the dominated convergence theorem. Thus, applying the Lindeberg-Feller central limit theorem [Section *2.8 of 4], we finish the proof of (S.38).

Now we prove (S.39). By similar derivations in (S.27), we know that

$$\mathbb{E}(H_{\text{lik}}^{\tilde{\boldsymbol{\vartheta}}}\mid\tilde{\boldsymbol{\vartheta}}) = \mathbb{V}\{a_N^{-1}\dot{\ell}_{\text{lik}}^{\tilde{\boldsymbol{\vartheta}}}(\boldsymbol{\theta}^*)\mid\tilde{\boldsymbol{\vartheta}}\} = \boldsymbol{\Lambda}_{\text{lik}}^{\text{plt}} + o_P(1). \tag{S.42}$$

where the second equality is from (S.41). In addition, the conditional variance of each component of $H_{\text{lik}}^{\tilde{\boldsymbol{\vartheta}}}$ is bounded by

$$a_N^{-4}N\mathbb{E}\{\delta^{\tilde{\boldsymbol{\vartheta}}}\{\phi_\pi^{\tilde{\boldsymbol{\vartheta}}}(\mathbf{x}_i;\boldsymbol{\theta}^*)\}^2\|\dot{g}(\mathbf{x}_i;\boldsymbol{\theta}^*)\|^4\mid\tilde{\boldsymbol{\vartheta}}\}$$

$$\leq a_N^{-4}N\mathbb{E}[\{y + (1 - y)\pi_\varrho^{\text{os}}(\mathbf{x};\tilde{\boldsymbol{\vartheta}})\}p_\pi^{\tilde{\boldsymbol{\vartheta}}}(\mathbf{x}_i;\boldsymbol{\theta}^*)\|\dot{g}(\mathbf{x}_i;\boldsymbol{\theta}^*)\|^4\mid\tilde{\boldsymbol{\vartheta}}]$$

$$\leq a_N^{-4}N\mathbb{E}[\{p(\mathbf{x};\boldsymbol{\theta}^*) + \pi_\varrho^{\text{os}}(\mathbf{x};\tilde{\boldsymbol{\vartheta}})\}p_\pi^{\tilde{\boldsymbol{\vartheta}}}(\mathbf{x}_i;\boldsymbol{\theta}^*)\|\dot{g}(\mathbf{x}_i;\boldsymbol{\theta}^*)\|^4\mid\tilde{\boldsymbol{\vartheta}}]$$

$$\leq 2a_N^{-4}Ne^{\alpha^*}\mathbb{E}\{e^{f(\mathbf{x};\boldsymbol{\beta}^*)}\|\dot{g}(\mathbf{x}_i;\boldsymbol{\theta}^*)\|^4\} = 2a_N^{-2}\mathbb{E}\{e^{f(\mathbf{x};\boldsymbol{\beta}^*)}\|\dot{g}(\mathbf{x}_i;\boldsymbol{\theta}^*)\|^4\} = o(1). \tag{S.43}$$

From (S.42) and (S.43), Chebyshev's inequality implies that $H_{\text{lik}}^{\tilde{\boldsymbol{\vartheta}}} \to \boldsymbol{\Lambda}_{\text{lik}}^{\text{plt}}$ in probability.

## S.10   Experiments on Pilot Mis-specification

This section complements Section 6.1 in the main text. We generate all data with a logistic regression model with $g(\mathbf{x};\boldsymbol{\theta}) = \alpha + \mathbf{x}^{\text{T}}\boldsymbol{\beta}$. The total number of data is $N = 5 \times 10^5$ with $N_0$ negative samples and $N_1$ positive samples. When generating data, we tune the bias $\alpha$ to force $N_0/N_1 \approx 400$. In order to test the robustness of our methods, we add an artificial biased to the pilot model, which will produce skewed sampling probabilities through the equation

$$\pi_\varrho^{\text{os}}(\mathbf{x};\tilde{\boldsymbol{\vartheta}}) = \min[\max\{\rho\tilde{\varphi}_{\text{os}}(\mathbf{x}),\varrho\},1], \quad \text{where} \quad \tilde{\varphi}_{\text{os}}(\mathbf{x}) = \tilde{\omega}^{-1}t(\mathbf{x};\tilde{\boldsymbol{\theta}}). \tag{S.44}$$

We try two types of biases: one on the intercept term $\alpha$ and the other on weights $\boldsymbol{\beta}$.

### S.10.1 Perturb the pilot intercept $\tilde{\alpha}$.

We perturb $\tilde{\alpha} \leftarrow \tilde{\alpha} + \xi \times \mathbb{U}(0,1) \times \log(N_0/N_1)$, where $\xi$ is a hyperparameter controlling the degree of perturbation. We try $\xi = \{0.1, 0.5, 1.0\}$. As the result will show, perturbing the intercept will not affect the relative performance for various types of sampling algorithms.

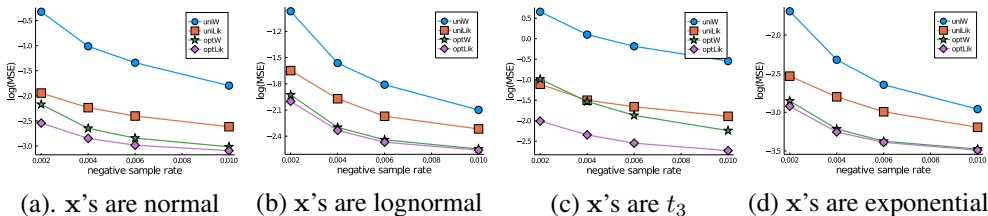

(a). **x**'s are normal     (b) **x**'s are lognormal     (c) **x**'s are $t_3$     (d) **x**'s are exponential

Figure S.1: Degree of perturbation $\xi = 0.1$. Log (MSEs) of subsample estimators for different sample sizes (the smaller the better).

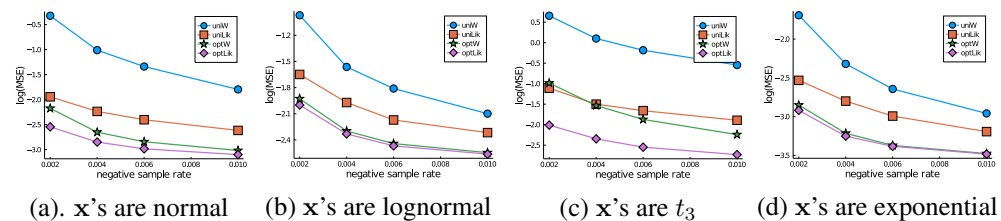

(a). **x**'s are normal     (b) **x**'s are lognormal     (c) **x**'s are $t_3$     (d) **x**'s are exponential

Figure S.2: Degree of perturbation $\xi = 0.5$. Log (MSEs) of subsample estimators for different sample sizes (the smaller the better).

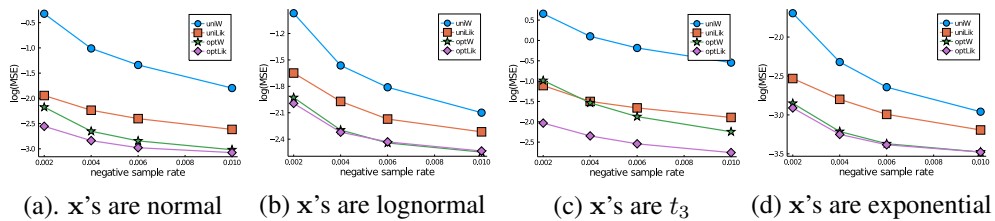

(a). **x**'s are normal     (b) **x**'s are lognormal     (c) **x**'s are $t_3$     (d) **x**'s are exponential

Figure S.3: Degree of perturbation $\xi = 1.0$. Log (MSEs) of subsample estimators for different sample sizes (the smaller the better).

### S.10.2 Perturb the pilot weights $\beta$.

We perturb $\tilde{\boldsymbol{\beta}} \leftarrow \tilde{\boldsymbol{\beta}} + \xi \times \mathcal{N}(0, I)$, where $\xi$ is a hyperparameter to control the degree of perturbation. We try $\xi = \{0.1, 0.5, 1.0\}$. As the result shows, the IPW estimator is pretty sensitive to the perturbation, and optW can be even worse than uniLik when the perturbation is significant (Figure S.6). uniLik may exceed optLik when the perturbation is too large such that the optimal sampling probability is significantly mis-calculated (Figure S.6(d)).

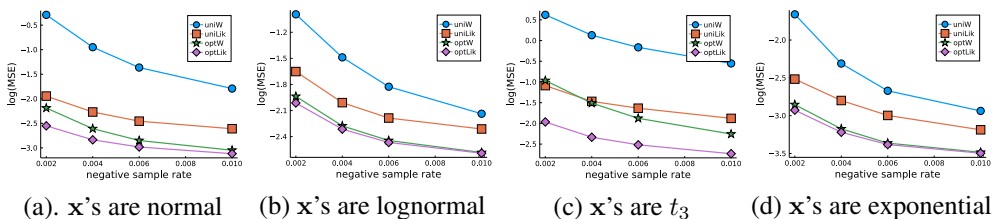

(a). **x**'s are normal    (b) **x**'s are lognormal    (c) **x**'s are $t_3$    (d) **x**'s are exponential

Figure S.4: Degree of perturbation $\xi = 0.1$. Log (MSEs) of subsample estimators for different sample sizes (the smaller the better).

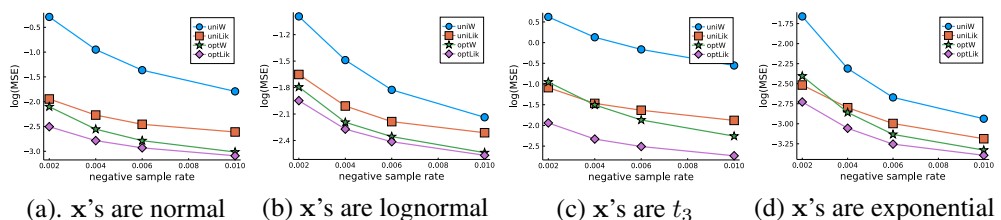

(a). **x**'s are normal    (b) **x**'s are lognormal    (c) **x**'s are $t_3$    (d) **x**'s are exponential

Figure S.5: Degree of perturbation $\xi = 0.5$. Log (MSEs) of subsample estimators for different sample sizes (the smaller the better).

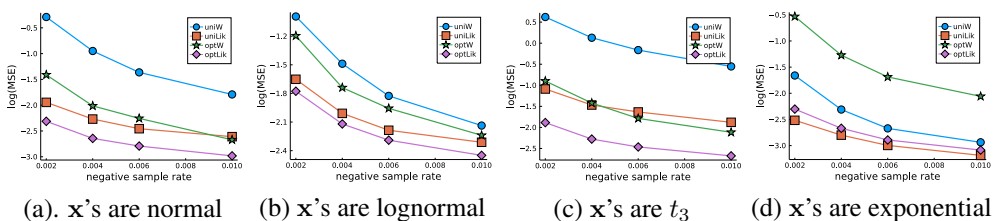

(a). **x**'s are normal    (b) **x**'s are lognormal    (c) **x**'s are $t_3$    (d) **x**'s are exponential

Figure S.6: Degree of perturbation $\xi = 1.0$. Log (MSEs) of subsample estimators for different sample sizes (the smaller the better).

## S.11    Experiments on Model Mis-specification

This section studies the case when model is mis-specified. That is, the ground truth model fall out of the domain of linear logistic regression model. We design the ground truth model as a two-layer neural network with non-linear activations:

$$g(\mathbf{x}; \boldsymbol{\theta}) = \alpha + \frac{1}{\xi} \tanh(\xi \cdot \mathbf{x}^{\mathrm{T}} \mathbf{W}) \boldsymbol{\beta}, \tag{S.45}$$

where $\xi$ is a constant to control the "cut-off" threshold of the non-linear activation $\tanh$. Note that as $\xi \to 0$, the scaled $\tanh$ function $\frac{1}{\xi} \cdot \tanh(\xi \cdot x) \approx x$ as shown in Figure S.7. Thus, by tuning $\xi$ we have the control of the deviation of scaled $\tanh$ function from a linear, identity transformation.

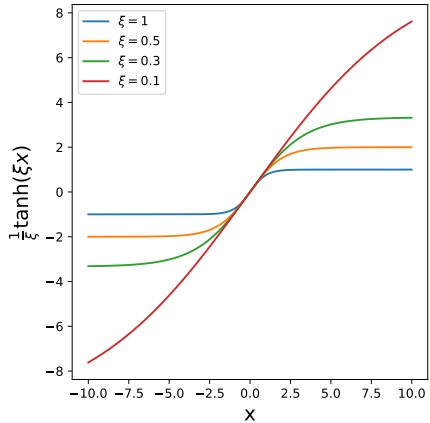

Figure S.7: Scaled $\tanh$ function approximates identity transformation when $\xi \to 0$.

We also generate $\mathbf{W}$ to be an approximately identity matrix, by setting the diagonal elements of $\mathbf{W}$ to be 1, and draw off diagonal terms to be i.i.d. Normal distribution scaled by a factor $\xi_w$. Clearly, the operator norm $\|\mathbf{W} - I\|_{2\to 2}$ is controlled by $\xi_w$. When $\xi_w = 0$, $\mathbf{W} = I$.

When perturbing the ground-truth model by setting $\xi$ and $\xi_w$ (thus $\mathbf{W}$), we need to reset the intercept $\alpha$ to match the pos/neg proportion to be $1/400$. For each experiment setup, we record the tuned $\alpha$ in the caption for reproducible research.

### S.11.1  Ablation study.

When $\xi$ and $\xi_w$ are extremely small, $g(\mathbf{x}; \boldsymbol{\theta}) \approx \alpha + \mathbf{x}^{\mathrm{T}}\boldsymbol{\beta}$. Empirical results shown in Figure S.8 match results without model mis-specification.

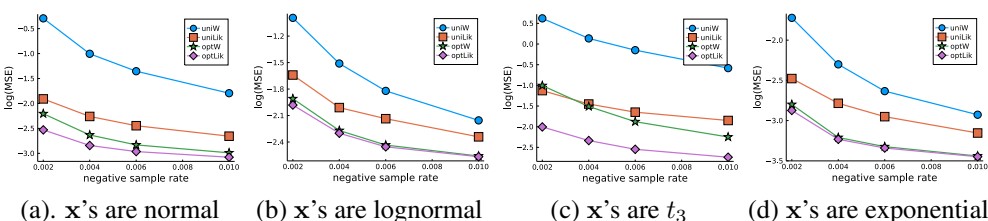

(a). $\mathbf{x}$'s are normal    (b) $\mathbf{x}$'s are lognormal    (c) $\mathbf{x}$'s are $t_3$    (d) $\mathbf{x}$'s are exponential

Figure S.8: Degree of non-linearity $\xi = 10^{-3}, \xi_w = 10^{-3}$. Ground-truth intercepts: (a) $\alpha = -7.65$, (b) $\alpha = -0.5$, (c) $\alpha = -7$, (d) $\alpha = -1.8$. Log (MSEs) of subsample estimators for different sample sizes (the smaller the better).

### S.11.2  Tune the degree of activation function non-linearity by trying different $\xi$.

We fix $\xi_w = 10^{-3}$ and tune $\xi$ to study the effect of non-linear activation. Note that when $\xi \to 0$, $\frac{1}{\xi}\tanh(\xi\mathbf{x}) \approx \mathbf{x}$ and when $\xi \to \infty$, $\frac{1}{\xi}\tanh(\xi\mathbf{x}) \approx 0$. As we shall see, the value $\xi$ significantly affect the non-linearity of the model. In Figure S.9, we choose a relatively small $\xi = 0.1$ and observe an almost indistinguishable loss in $\log(\text{MSE})$. In Figure S.10 we study the case when $\xi = 0.5$ and find that it significantly affect the model estimation when $\mathbf{x}$ has long tails (Figure S.10(b), (d)). When $\xi = 1$, all method crash since the true (non-linear) model deviate from the tractable domain (linear). In order to avoid this happen, we suggest to re-scale and centering the data before feeding them to the model. Note that when models are not crashed, optLik still achieves best performance among all methods.

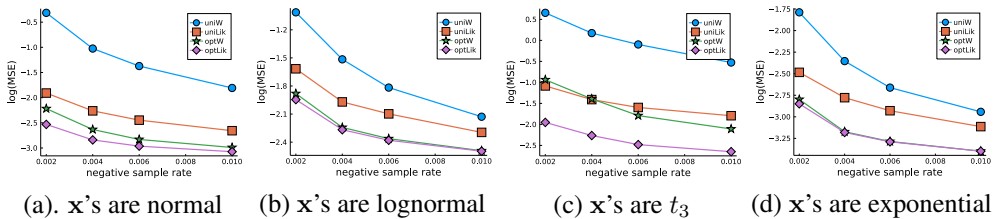

(a). **x**'s are normal  (b) **x**'s are lognormal  (c) **x**'s are $t_3$  (d) **x**'s are exponential

Figure S.9: Degree of non-linearity $\xi = 10^{-1}, \xi_w = 10^{-3}$. Ground-truth intercepts: (a) $\alpha = -7.7$, (b) $\alpha = -0.6$, (c) $\alpha = -6.9$, (d) $\alpha = -1.9$. Log (MSEs) of subsample estimators for different sample sizes (the smaller the better).

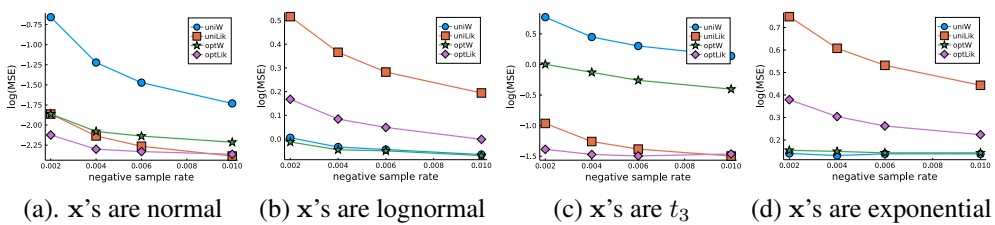

(a). **x**'s are normal  (b) **x**'s are lognormal  (c) **x**'s are $t_3$  (d) **x**'s are exponential

Figure S.10: Degree of non-linearity $\xi = 5{\cdot}10^{-1}, \xi_w = 10^{-3}$. Ground-truth intercepts: (a) $\alpha = -7.4$, (b) $\alpha = -0.95$, (c) $\alpha = -6.7$, (d) $\alpha = -2.15$. Log (MSEs) of subsample estimators for different sample sizes (the smaller the better).

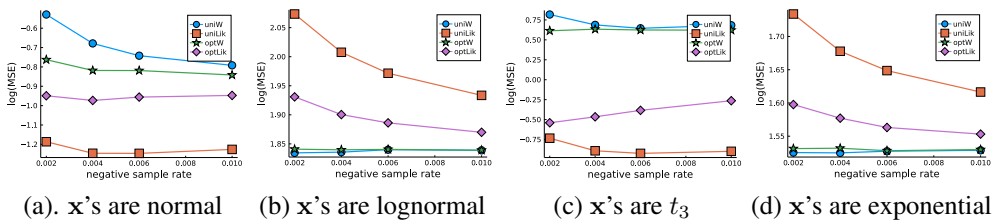

(a). **x**'s are normal  (b) **x**'s are lognormal  (c) **x**'s are $t_3$  (d) **x**'s are exponential

Figure S.11: Degree of non-linearity $\xi = 1.0, \xi_w = 10^{-3}$. Ground-truth intercepts: (a) $\alpha = -7.05$, (b) $\alpha = -1.75$, (c) $\alpha = -6.4$, (d) $\alpha = -2.85$. Log (MSEs) of subsample estimators for different sample sizes (the smaller the better).

### S.11.3  Tune $\|\mathbf{W} - I\|_{2\rightarrow 2}$ by trying different $\xi_w$.

We fix $\xi = 10^{-3}$ and focus on studying the effect of generative process of $\mathbf{W}$ to the estimation error. In Figure S.12, we set $\xi_w = 2{\cdot}10^{-2}$ to be a relatively small number such that $\|\mathbf{W} - I\|_{2\rightarrow 2} \approx 0.20 < 1$. In this case, each sampling method suffers from model mis-specification with higher $\log(\text{MSE})$. Still optLik is the best among all generative processes of $\mathbf{x}$. In Figure S.13, we set $\xi_w = 10^{-1}$ and $\|\mathbf{W} - I\|_{2\rightarrow 2} \approx 1$. In this case, the noise level is comparable to the ground-truth. All sampling models crash such that the estimated models deviate significantly from the ground-truth.

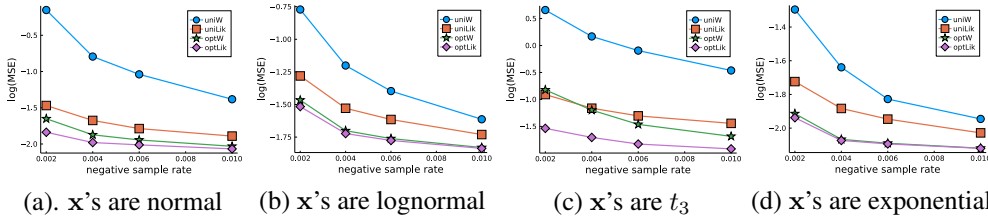

(a). **x**'s are normal  (b) **x**'s are lognormal  (c) **x**'s are $t_3$  (d) **x**'s are exponential

Figure S.12: Degree of non-linearity $\xi = 10^{-3}, \xi_w = 2{\cdot}10^{-2}$. Ground-truth intercepts: (a) $\alpha = -7.7$, (b) $\alpha = -0.5$, (c) $\alpha = -6.9$, (d) $\alpha = -1.9$. Log (MSEs) of subsample estimators for different sample sizes (the smaller the better).

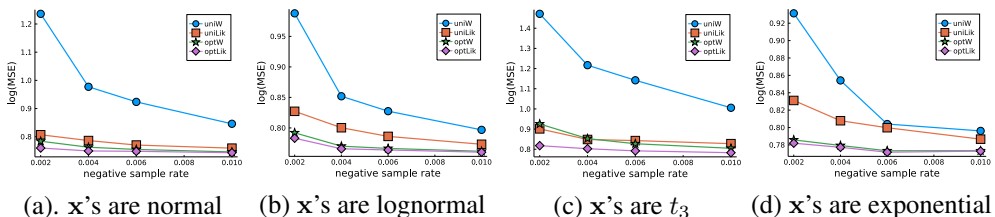

(a). **x**'s are normal    (b) **x**'s are lognormal    (c) **x**'s are $t_3$    (d) **x**'s are exponential

Figure S.13: Degree of non-linearity $\xi = 10^{-3}, \xi_w = 10^{-1}$. Ground-truth intercepts: (a) $\alpha = -8.25$, (b) $\alpha = -0.25$, (c) $\alpha = -6.6$, (d) $\alpha = -2.4$. Log (MSEs) of subsample estimators for different sample sizes (the smaller the better).