# OpenReview forum: "Nonuniform Negative Sampling and Log Odds Correction with Rare Events Data"
_NeurIPS.cc/2021/Conference — NeurIPS 2021 Poster_

### Official Review · Reviewer_VDHH · 2021-07-14

**Rating:** 6
**Confidence:** 2

**Summary:**

The paper discusses the sampling strategy for imbalanced binary-label datasets. For a binary-label dataset where the number of negative samples $N_0$ is much larger than that of positive samples $N_1$, we often "subsample" a part of negative samples in order to balance two labels.

The paper discusses the asymptotic convergence rate of the optimal model parameter against the number of samples in the dataset, and improves the rate from conventional model parameter estimation method.
-   Theorem 1 proved that the convergence rate is proportional to $N_1^{-1/2}$, that is, determined not by the number of all instances but of positive instances if $N_0\gg N_1$.
-   Theorems 2 and 3 composes a function that provides the best strategy to subsample negative samples.
-   Theorems 4 through 6 modifies the result of Theorems 2 and 3 for improving the efficiency by incorporating the sampling strategy.
-   Theorems 7 and 8 discusses the effect of the "pilot" model parameter and provides the construction of it for better convergence rate.

**Limitations And Societal Impact:**

The impact is mainly stated in Section 1 and 7, theoretical assumptions are stated before respective theorems, and limitations are stated in Section 7. The reviewer thinks that they are adequately stated.


**Main Review:**

Since we often encounter imbalanced datasets, subsampling strategy is very important for them. The proposed method is theoretically proved to be effective, including the setups for practical uses. Experimental results supported the results.

The reviewer felt that the paper was not well organized in the sense that the paper's contribution is not clear. The reviewer hopes the paper that the following points to be clearer: what is based on conventional methods, and what is new in the proposed method. (In the appendix there are references if existing analyses are employed for the proposed method, but the reviewer hopes references in the main paper in order to discuss the novelty of the paper.)

-   Section 1, the part "Rare events data and negative sampling are studied in [28], but it focused on linear logistic regression and only considered uniform sampling. We approach this problem under general binary response models from the perspective of optimal subsampling, which aims to minimize the asymptotic variance of the IPW estimator [30, 26, 29, 31]. This topic was not well investigated neither for imbalanced data nor for negative sampling."
    -   What is main updates of the proposed method from the negative sampling strategy in [28] and the IPW? Does the paper just combines them (as well as providing with some new analyses)?
-   Section 2.1
    -   The reviewer felt that the result of Theorem 1 for simpler case seem to be proved by someone else. Please refer to conventional related analyses and their relationships to Theorem 1 in the main paper (not only the appendix) if any: For example, balanced case with (without) Assumptions 1 or 2, or imbalanced case with different assumptions.
-   Section 2.1
    -   Please provide, if any, the intuitive interpretation of $M^{-1}_f$ since it is essential to discuss the performance of the proposed method.
-   Section 6
    -   Since main theorems discuss the convergence rate against the number of samples in Theorem 1, perhaps experiment on it will support the paper's analysis more.

**Time Spent Reviewing:**

3

---

> ### Author Response · Authors · 2021-08-08
> **Response to Reviewer VDHH**
>
> Thank you for your helpful review. We will more clearly distinguish our work from existing results in the paper. Please see the following for our response to your specific comments/questions.
>
> **Q1. The main updates of the proposed method from the negative sampling strategy in [28] and the IPW.**
>
> **A1**. In [28], only linear logistic regression is studied and only uniform negative sampling is considered. Compared with [28], we have 1) considered general binary response models (of course the linear logistic regression is a specific case of our model setup); 2) derived the optimal negative sampling function for IPW estimators; 3) developed the likelihood-based estimator through nonuniform log-odds correction, and 4) tested the proposed method on a real large data. Compared with existing researches on optimal sampling for IPW, our analysis is carried out in a new framework where the data distribution depends on $N$. More significantly, our paper gives the optimal sampling function without conditioning on the observed data. Existing literature about IPW derives optimal sampling probabilities conditional on observed data [e.g., 26, 29, 30, 31, among others] because their goal is to approximate the full data estimator, while our focus is to use negative sampled data to estimate the population parameter. If we set our model specifically to the linear logistic regression, then our result of Theorem 1 will reduce to that of Theorem 1 in [28], but our results in other Theorems will still be new additions to the literature.
>
> **Q2. Please refer to conventional related analyses and their relationships to Theorem 1.**
>
> **A2**. If we set our model specifically to the linear logistic regression, then our result of Theorem 1 will reduce to that of Theorem 1 in [28], but our results in other Theorems will still be new additions to the literature. For balanced cases, the asymptotic normality is a conventional result, but the convergence rate is expressed in terms of $N$ instead of $N_1$ and the asymptotic variance is different from that of our Theorem 1. We will point out the connections of our findings to existing results.
>
> **Q3. The intuitive interpretation of $M_f^{-1}$.**
>
> **A3**. For the conventional case when the data distribution does not depend on $N$, $M_f$ would be the Fisher information matrix, and $M_f^{-1}$ would be the Cramer-Rao lower bound for variances of unbiased estimators. However, for our case, although we conjecture this is true, a rigorous proof is required,  and we will investigate this in the future work.
>
> **Q4. Experiments on the convergence rate against the number of samples in Theorem 1.**
>
> **A4**. Please see the following tables for our results, in which we let the full data size $N$ increase much faster than the average number of positive instances $N_1$ so that the probability $\Pr(y=1)$ decreases towards zero. As $N$ increases, we fix $\beta$ and set decreasing values to $\alpha$, so this mimics our scaling regime. We generate covariates from multivariate normal and log-normal distributions and fit logistic models. A logistic model is a correct model (Table 1) when covariates are from normal distributions and it is a misspecified model (Table 2) when covariates are from log-normal distributions. We simulate for 100 times to calculate the empirical variance $\hat{V}(\hat{\theta}_f)$.
>
> Table 1 - Covariate distributions are normal (correctly specified model):
>
> | $N$  | $E(N_1)$  | $\Pr(y=1)$ | tr{$\hat{V}(\hat{\theta}_f)$} | $E(N_1)\times$tr{$\hat{V}(\hat{\theta}_f)$} | $N\times$tr{$\hat{V}(\hat{\theta}_f)$} |
> | :------: |:-------:| ------------:| :-------: |:--------:| ------------:|
> | $10^3$ | $2^5$ | $0.032$    | $0.169$ | $5.41$ | $169.17$   |
> | $10^4$ | $2^6$ | $0.0064$   | $0.097$ | $6.20$ | $969.29$   |
> | $10^5$ | $2^7$ | $0.00128$  | $0.045$ | $5.76$ | $4497.24$  |
> | $10^6$ | $2^8$ | $0.000256$ | $0.018$ | $4.62$ | $18048.40$ |
>
>
> Table 2 - Covariate distributions are log-normal (misspecified model):
>
> | $N$  | $E(N_1)$  | $\Pr(y=1)$ | tr{$\hat{V}(\hat{\theta}_f)$} | $E(N_1)\times$tr{$\hat{V}(\hat{\theta}_f)$} | $N\times$tr{$\hat{V}(\hat{\theta}_f)$} |
> | :------: |:-------:| ------------:| :-------: |:---------:| ------------:|
> | $10^3$ | $2^5$ | $0.032$    | $0.969$ | $30.99$ | $968.70$   |
> | $10^4$ | $2^6$ | $0.0064$   | $0.322$ | $20.59$ | $3217.12$  |
> | $10^5$ | $2^7$ | $0.00128$  | $0.135$ | $17.32$ | $13527.6$  |
> | $10^6$ | $2^8$ | $0.000256$ | $0.046$ | $11.74$ | $45847.4$  |
>
> It is seen that tr{$\hat{V}(\hat{\theta}_f)$} is decreasing towards zero in both tables. According to Theorem 1, the tr{$\hat{V}(\hat{\theta}_f)$} should converge in a rate of $1/N_1$, so $E(N_1)\times$tr{$\hat{V}(\hat{\theta}_f)$} should be relatively stable as $N$ increases. This is indeed the case as seen in the fifth columns of the above tables. On the other hand, $N\times$tr{$\hat{V}(\hat{\theta}_f)$} gets large dramatically as $N$ increases.
>
> The aforementioned observations confirms the theoretical result in Theorem 1. Furthermore, Table 2 is obtained from a misspecified model, showing that the convergence rate in Theorem 1 may also be true for some misspecified models.

---

> > ### Comment · Reviewer_VDHH · 2021-08-19
> > **Thank you for response**
> >
> > Thank you for response.
> > The reviewer understood the points of concern by the response, and the fixes in the comment will improve the paper.
> >
> > Note: Regarding Q3, what the reviewer misunderstood was the "MSE" in Section 6. It is not the MSE of predicted result (which appears much more often in the machine learning field), but the MSE of the estimated parameter $\theta$.

---

### Official Review · Reviewer_j78J · 2021-07-20

**Rating:** 7
**Confidence:** 4

**Summary:**

This paper considers the estimation of the parameters of a binary response model when the positive instances are rare. Rareness is modeled via a scaling regime on the logits, with the key restriction being that the scaling occurs _uniformly over features_. The paper first shows that the fraction of positive instances bottlenecks estimation and uses this to motivate subsampling the negative instances, to save on computation. The paper considers two negative subsampling approaches: importance weighting the original likelihood and maximizing it (IPW), or maximizing the log-likelihood under subsampling (Lik). Under correct model specification, asymptotic analysis is provided for both methods, with the latter enjoying many advantages, including stability and smaller asymptotic variance. The approach is tested on synthetic and click-through-rate data. The experiments support the theory and go beyond it, by showing that the methods work well even under model misspecification.

**Ethical Concerns:**

No ethical concerns. Some of the experiments are rather large and impossible to reproduce with small resources, but there are also synthetic experiments that are (likely, did not try) reproducible.

**Limitations And Societal Impact:**

The paper adequately discusses the limitations of the work. Societal impact is not discussed. While there may be some implications, considering that a big motivation is eliciting marketing response, which could negatively impact some sub-populations, it's not immediately apparent whether the proposed approaches could actively lead to discrimination.

**Main Review:**

### Strengths

+ The paper gives a very comprehensive modeling and treatment of rare positive instances in binary response: motivating sub-sampling, revisiting key sub-sampling methods, optimizing these methods to perform well, analyzing the presented variants, giving practical implementation tweaks, analyzing those tweaks, and finally testing on meaningful experiments. Each of these steps is a worthwhile contribution to share with the community. _(originality, significance, quality)_

### Weaknesses

- The scaling regime is somewhat limiting, because it requires the logits to decay uniformly over all features. This is definitely an acceptable and useful simplifying assumption for theoretical work, but it’s not clear why this is the right model to capture rare positives. Some discussion on this matter would be appreciated. _(significance)_

- The paper is theory-intense, which is not a problem, but this is presented in a dry and dense fashion which makes it a heavy read. More signpost paragraphs to maintain the high-level picture can make this a more pleasant read. _(clarity)_

- Some claims need a little care for nuance. For example, on line `line 86`, does the MLE remain the most efficient estimator (among asymptotically unbiased ones), even in this scaling regime? Saying the number of positive instances is what matters `line 98`, is obviously not true, because the result hinges on $N_1/N\to 0$. Of course, down the line with the subsampling results, it’s clear we need to include "enough" negative samples to get the same asymptotic behavior as the full sample.  On `line 117` saying that will happen at $c=0$ should be emphasized to be just conceptual, since at that point Theorem 2 no longer holds. _(quality)_

### Suggestions

* It’s worthwhile to do a grammatical pass on the text, since there are several incorrect English phrases. These don’t deter from the quality of the work, but the polish would be appreciated.

[Edit: Thank you authors for your response. My evaluation is unchanged.]

**Time Spent Reviewing:**

4.5

---

> ### Author Response · Authors · 2021-08-08
> **Response to Reviewer j78J**
>
> Thank you for your insightful review of our paper.
>
> **Q1. Limitations of the scaling regime.**
>
> **A1**. The scaling regime means that both the marginal and conditional probabilities for a positive instance are small and a change in the feature value does **NOT** make a rare event become a large probability event. In the following, we present three scenarios that the scaling regime may or may not fit in:
>
> Scenario 1 (phone call while driving). Car crashes occur with small probabilities, and making phone calls while driving significantly increases the probability of a car crash. However, the probability of car crashes among people making phone calls while driving is still small, so for these types of features, our scaling regime is appropriate to model the rare events.
>
> Scenario 2 (anti-causal learning). Anti-causal learning [r1, r2] assumes label ($Y$) causes observations ($X$). Thus $\Pr(X|Y)$ represents the causal mechanism and is fixed. One standard example is that diseases ($Y$) cause symptoms ($X$). Our scaling regime fits the framework of anti-causal learning. To see this, using Bayes Theorem we write the log odds as
>
> $$\alpha+f(x;\beta)=\log\frac{\Pr(Y=1)}{\Pr(Y=0)} + \log\frac{\Pr(x\mid Y=1)}{\Pr(x\mid Y=0)}.$$
>
> In anti-causal learning, the marginal distribution of $Y$ changes while the conditional distribution of $X$ given $Y$ is fixed. Thus, only the scale factor $\alpha$ changes. $f(x;\beta)$ is fixed.
>
> [r1] Schölkopf, et al. "On causal and anticausal learning." ICML. 2012.
>
> [r2] Kilbertus, et al. "Generalization in anti-causal learning." arXiv preprint arXiv:1812.00524 (2018).
>
> Scenario 3 (local transmission of the COVID-19). Of course, our scaling regime has some limitations and does not apply to all types of rare events data. For example, although the COVID-19 infection rate for the whole is low, the infection rate for people whose family members are in the same house with positive test results is high. This means the change of a family member's test result converts a small-probability-event to a large-probability-event, and our scaling regime would not be appropriate.
>
> **Q2. Does the MLE remain the most efficient estimator in the scaling regime?**
>
> **A2**. In the scaling regime the data distribution changes with $N$ and the standard result about the MLE being the most efficient estimator does not apply. We conjecture that the MLE is still the most efficient estimator, but a rigorous proof is required. We will change "the most efficient estimator" to "a commonly used estimator" before obtaining a rigorous proof.
>
> **Q3. Line 98 saying the number of positive instances is what matters, is not true.**
>
> **A3**. For the sentence in line 98, we mean to say that although $N\rightarrow\infty$ much faster than $N_1\rightarrow\infty$ (in terms of $N_1/N\rightarrow0$), $N$ does not explicitly show in the asymptotic normality result of Theorem 1. We will clarify this in the paper. We will also carefully proofread the paper to make our statements more precise and rigorous.
>
> **Q4. $c=0$ should be emphasized to be just conceptual, since at that point Theorem 2 no longer holds.**
>
> **A4**. For Theorem 2, it is valid if $c=0$; we require $c\in[0,\infty)$ in our Assumption 3 in line 110. We will rewrite the expression as "for a constant $c$ such that $0\le c<\infty$" to make this more explicit. Note that $c_N$ cannot be zero but its limit $c$ can be exactly zero.
>
> **Q5. Social impact.**
>
> **A5**. Thank you for your comment about the societal impact. Heuristically, our nonuniform negative sampling method gives preference to data points that are harder to observe, so in your example, it would give preference to sub-populations that are difficult to observe. We do not think this would bring any negative societal impacts. We also have also examined our real data application and did not notice discrimination against any sub-population. We will add some discussions on this in Section 7.

---

### Official Review · Reviewer_SFTU · 2021-07-21

**Rating:** 8
**Confidence:** 2

**Summary:**

This paper considers the question of parameter estimation in a binary response model (e.g., logistic regression), in the imbalanced case where the fraction of positive instances is small/vanishing.
This question is formulated by considering a well-specified model, and adding an intercept to the log-odds ratio which diverges as the sample size goes to infinity (with the other parameters kept fixed).

The main results are the following:

- First, it is shown (in Theorem 1) that the convergence rate of the MLE (computed on the full dataset) only depends on the number N_1 of positive instance, rather than on the sample size (which also includes negative instances). MLE also shown to be asymptotically normal, and the asymptotic covariance is provided.

- Since the convergence rate of MLE only depends on N_1 (but not on the total number N of samples), it can make sense to perform "negative sampling", namely to reduce N by only retaining a small fraction of negative instances, in order to reduce computations. Of course, removing negative instances creates a bias, so the authors considers inverse probability weighted (IPW) estimators, where negative samples are re-weighted (inversely proportionally to the deletion probability). An asymptotic normality result is obtained, with a covariance larger than MLE (as expected), by an additional covariance depending on the rejection probability function (Theorem 2). The optimal rejection function (minimizing asymptotic variance) is deduced (Theorem 3).

- Finally, the authors consider a different estimator, based on log-odds correction. This estimator corresponds to MLE on the selected data, based on adjusting the likelihood of the selected data by accounting for the deletion mechanism. This leads to a correction which is less sensitive to small probability instances than the IPW method.
The asymptotic normality and covariance of this estimator is provided (Theorem 4). Interestingly, it is shown (Theorem 5) that this scheme is always better than the IPW (with optimal rejection function), in that it leads to a smaller asymptotic covariance. An optimality result of this estimator among (asymptotically) unbiased estimators is then provided in Theorem 6.

**Limitations And Societal Impact:**

Yes

**Main Review:**

I enjoyed reading this paper. It is well-written, considers a practically meaningful question, and provides precise answers. It also does not contain any spurious or unsubstantiated claims or decorative results. I appreciate the interesting adjusted MLE estimator based on the likelihood of selected data, and in particular its comparison with the IPW. The results also provide a coherent picture and naturally complement each other.

I am therefore strongly in favor of acceptance of this paper, and would also suggest considering it for a spotlight or oral presentation, as it could potentially be of interest to many NeurIPS attendees. I would also encourage the authors to consider publishing a final version of this work to a suitable statistics journal.

I should mention, however, that I am not familiar with the literature on the topic, in the sense that I have read none of the papers listed in the references. I am therefore not in a very good position to gauge the level of originality (though I somehow doubt that related work could warrant rejection, unless the same results were obtained before). I might also be slightly biased by the fact that many of the ideas are new to me, although some might be more routine in this topic. (Due to a high reviewing load, I did not check the proofs in the supplementary material.)
I have accounted for this both in my low confidence score and in the overall score.

Below are some comments:

1. I would suggest (maybe in a final version) to put the results (asymptotic covariance matrices) in more explicit or simple form in the specific case of logistic regression, which is arguably one of the most standard and relevant ones.

2. It would perhaps be helpful to comment the results in connection with the asymptotic distribution of the MLE (given by the Fisher distribution) or the Cramer-Rao lower bound/asymptotic optimality of MLE. It actually seems that these standard result do not apply, as the data distribution changes with N (since the fraction of positive instances vanishes), which justifies a specific analysis as in this paper.

3. It seems that, as noted in the paper, the misspecified case could also be considered in future work. Asymptotic normality should also hold there (with a covariance depending on the distribution), with the true parameter replaced by the best KL approximation.

**Time Spent Reviewing:**

an afternoon

---

> ### Author Response · Authors · 2021-08-08
> **Response to Reviewer SFTU**
>
> Thank you for your positive opinion on our paper along with the insightful comments. We are excited to further pursue the topic and include more investigations to develop an extended final version in future work.
>
> Response to your comments:
> 1. For the logistic regression case, our result in Theorem 1 reduces to that of Theorem 1 in [28]. Our analysis goes beyond logistic regression and we present more novel results in Theorems 2 through Theorem 8.
> 2. You are totally correct. Since the data distribution changes with $N$, the standard Cramer-Rao result does not apply. But we conjecture that the MLE is still asymptotically optimal and reaches the efficiency bound asymptotically. We will work on a rigorous proof towards an extended future version.
> 3. We agree that asymptotic normality would hold for misspecified models as well. We will study this point in our future work towards the final extended version of the paper.

---

### Author Response · Authors · 2021-08-08
**Collective response to all reviewers**

We gratefully thank all reviewers for their time spent assessing our work and making helpful comments.

As Reviewers SFTU and j78J pointed out, our studying case considers data distribution changing with $N$, so standard results such as the Cramer-Rao lower bound do not apply. In response to each reviewer, we discuss the significant improvements of our work to existing works such as [28] in both theoretical analysis (relaxed model assumptions and better estimators) and practical considerations (experiment on massive real data).

Like all model assumptions, our scaling regime may not fit all possible scenarios. In response to Reviewer j78J, we discuss three different cases: dangerous driving, anti-causal learning, and COVID-19 infection. Our scaling regime fits the first two scenarios and is not appropriate in the third scenario.

As suggested by Reviewer VDHH, we include an experiment to justify Theorem 1 that the convergence rate depends on the number of positive samples instead of the full data sample sizes.

---

### Decision · Program_Chairs · 2021-09-27

**Decision:**

Accept (Poster)

**Comment:**

In this paper, an optimal negative sampling strategy from imbalanced data for rare event prediction is studied. The problem of class imbalance in rare event prediction is of considerable importance in various applied problems. The theoretical properties of the proposed method are clearly presented, and its advantages are clearly demonstrated by experimental validation on artificial and real data. All reviewers found the paper useful for the ML community. The authors are encouraged to consider updating the paper based on the reviewers' comments and suggestions.